# Interval Bound Interpolation for Few-shot Learning with Few Tasks

## Abstract

Few-shot learning aims to transfer the knowledge acquired from training on a diverse set of tasks to unseen tasks from the same task distribution, with a limited amount of labeled data. The underlying requirement for effective few-shot generalization is to learn a good representation of the task manifold. This becomes more difficult when only a limited number of tasks are available for training. In such a few-task few-shot setting, it is beneficial to explicitly preserve the local neighborhoods from the task manifold and exploit this to generate artificial tasks for training. To this end, we introduce the notion of *interval bounds* from the provably robust training literature to few-shot learning. The interval bounds are used to characterize neighborhoods around the training tasks. These neighborhoods can then be preserved by minimizing the distance between a task and its respective bounds. We then use a novel strategy to artificially form new tasks for training by interpolating between the available tasks and their respective interval bounds. We apply our framework to both model-agnostic meta-learning as well as prototype-based metric-learning paradigms. The efficacy of our proposed approach is evident from the improved performance on several datasets from diverse domains in comparison to recent methods.

## 1 Introduction

Few-shot learning problems deal with diverse tasks consisting of subsets of data drawn from the same underlying data manifold along with associated labels. The joint distribution of data and corresponding labels which governs the sampling of such tasks is often called the task distribution (Finn et al., 2017; Yao et al., 2022). Consequently, few-shot learning methods attempt to leverage the knowledge acquired by training on a large pool of such tasks to easily generalize to unseen tasks from the same distribution, using only a few labeled examples. We hereafter refer to the support of the task distribution as the task manifold which is distinct from but closely-related to the data manifold associated with the data distribution. Since the unseen tasks are sampled from the same underlying manifold governing the task distribution, we should ideally learn a good representation of the task manifold by preserving the neighborhoods from the high-dimensional manifold in the lower-dimensional feature embedding (Tenenbaum et al., 2000; Roweis & Saul, 2000; Van der Maaten & Hinton, 2008). However, the labels associated with a task can define any arbitrary partitioning of the data. Therefore, we can attempt to preserve the neighborhood for a task by simply conserving the neighborhoods for the corresponding subset of the data manifold in the feature embedding learned by the few-shot learner. This facilitates effective generalization to new tasks using a limited amount of labeled data by only updating the classifier as the learned feature embedding would likely require very little adaptation. However, existing few-shot learning methods lack an explicit mechanism for achieving this. Further, real-world few-shot learning scenarios like rare disease detection may not have the large number of training tasks required for effective learning, due to various constraints such as data collection costs, privacy concerns, and/or data availability in newer domains (Yao et al., 2022). In such scenarios, few-shot learning methods are prone to overfit the training tasks, thus limiting the ability to generalization to unseen tasks. Therefore, in this work, we develop a method to explicitly constrain the feature embedding in an attempt to preserve neighborhoods from the high-dimensional task manifold and to construct artificial tasks within these neighborhoods in the feature space, to improve the performance when a limited number of training tasks are available.

The proposed approach relies on characterizing the neighborhoods from the high-dimensional task manifold and propagating them through the network with the intent to preserve the task neighborhood in the feature space. We achieve this by employing the concept of interval bounds from the provably robust training literature (Gowal et al., 2019; Morawiecki et al., 2020), i.e. the axis-aligned bounds for the activations in each layer, obtained using interval arithmetic (Sunaga, 1958). Concretely, as shown in Figure 1, we first define a small $\epsilon$-neighborhood for each few-shot training task and then use Interval Bound Propagation (IBP; Gowal et al., 2019) to obtain the bounding box around the mapping of the corresponding neighborhood in the feature embedding space. We then explicitly attempt to preserve the $\epsilon$-neighborhoods by minimizing the distance between a task and its respective interval bounds in addition to optimizing the few-shot classification objective. We further devise a mechanism to construct the artificial tasks by interpolating between a task and its corresponding IBP bounds. It is important to notice that this setup is distinct from provably robust training for few-shot learning in that we do not attempt to minimize (or calculate for that matter) the worst-case classification loss.

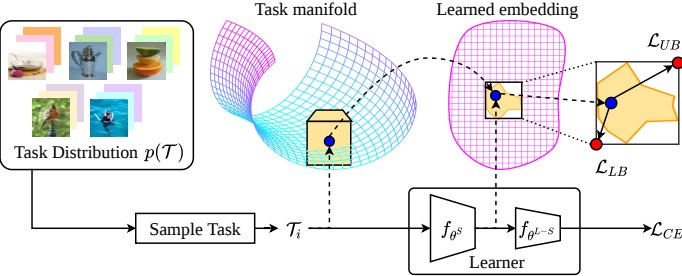

Figure 1: Illustration of the proposed interval bound propagation–aided few-shot learning setup (best viewed in color): We use interval arithmetic to define a small $\epsilon$-neighborhood around a training task $\mathcal{T}_i$ sampled from the task distribution $p(\mathcal{T})$. IBP is then used to obtain the bounding box around the mapping of the said neighborhood in the embedding space $f_{\theta^S}$ given by the first $S$ layers of the learner $f_\theta$. While training the learner $f_\theta$ to minimize the classification loss $\mathcal{L}_{CE}$ on the query set $\mathcal{D}_i^q$, we additionally attempt to minimize the losses $\mathcal{L}_{LB}$ and $\mathcal{L}_{UB}$, forcing the $\epsilon$-neighborhood to be compact in the embedding space as well.

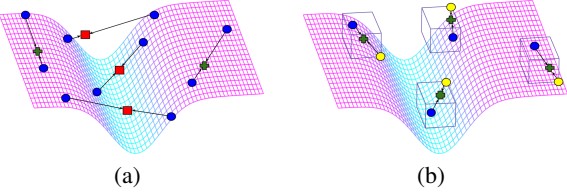

(a)        (b)

Figure 2: Interval bound–based task interpolation (best viewed in color): (a) Existing inter-task interpolation methods create new artificial tasks by combining pairs of original tasks (blue ball). However, depending on how flat the task-manifold embedding is at the layer where interpolation is performed, the artificial tasks may either be created close to the task-manifold (green cross) or away from the task-manifold (red box). (b) The proposed interval bound–based task interpolation creates artificial tasks by combining an original task with one of its interval bounds (yellow ball). Such artificial tasks are likely to be in the vicinity of the task manifold as the interval bounds are forced to be close to the task embedding by the losses $\mathcal{L}_{LB}$ and $\mathcal{L}_{UB}$.

Various methods have been proposed to mitigate the few-task few-shot problem using approaches such as explicit regularization (Jamal & Qi, 2019; Yin et al., 2019), intra-task augmentation (Lee et al., 2020; Ni et al., 2021; Yao et al., 2021), and inter-task interpolation to construct new artificial tasks (Yao et al., 2022). While inter-task interpolation has been shown to be the most effective among these existing approaches, it suffers from the limitation that the artificially created tasks may be generated away from the task manifold depending on the curvature of the feature embedding space, as there is no natural way to select pairs of task which are close to each other on the manifold (Figure 2(a)). The interval bounds obtained using IBP, on the other hand, are likely to be close to the original task embedding as we explicitly minimize the distance between a task and its interval bounds. Thus, using them for interpolation is likely to keep the generated tasks close to the manifold (Figure 2(b)).

In essence, the key contributions made in this article advance the existing literature in the following ways: **(1)** In Section 4.1, we present for the first time, a novel method to synergize few-shot learning with interval bound propagation (Gowal et al., 2019) so as to explicitly lend the ability to preserve task neighborhoods in the feature embedding space of the few-shot learner. **(2)** In Section 4.2, we propose the interval bound–based task interpolation technique which can create new tasks (as opposed to augmenting each individual task (Lee et al., 2020; Ni et al., 2021; Yao et al., 2021)), by interpolating between a task sampled from the task distribution and its interval bounds. **(3)** Unlike existing inter-task interpolation methods that require paired tasks for interpolation (Yao et al., 2022), our framework can generate new tasks from only a single task. This allows the proposed framework to be seamlessly integrated with existing few-shot learning paradigms.

In Section 5, we empirically demonstrate the effectiveness of our proposed approach on both gradient-based meta-learning and prototype-based metric-learning on few-task real-world datasets from multiple domains, in comparison to the recent prior methods. Finally, we make concluding remarks and also discuss future scopes of research in Section 6.

## 2   RELATED WORKS

**Few-shot learning:** The aim of few-shot learning is to generalize to new tasks using only a few examples (Wang et al., 2020) through three major strategies. First, one can augment the tasks at the data level (Hariharan & Girshick, 2017). Second, the hypothesis space can be constrained at the model level (Snell et al., 2017). Third, the hypothesis search strategy at the algorithm level can be improved (Finn et al., 2017). The problem of few-task learning can be even more difficult when there is a scarcity of training tasks in a few-task scenario. We take the route of Yao et al. (2022) to offer a novel task augmentation strategy that can work in conjunction with both algorithm-level meta-learning as well as model-level metric-learning methods.

**Provable robust training of neural networks:** A way to build robust neural networks is to find a differentiable upper bound on the verifiable violation of specifications. Such upper bounds can then be directly optimized alongside the original loss (Mirman et al., 2018; Raghunathan et al., 2018; Wong et al., 2018). IBP (Gowal et al., 2019) follows this direction by explicitly minimizing the worst-case loss inside the $\epsilon$-neighborhood of an input for an arbitrary network with some architectural constraints. However, in our work, instead of building robust networks, we repurpose IBP to characterize the $\epsilon$-neighborhood to learn better representation such that the generalization to new tasks by a few-shot learner becomes easier. Moreover, the bounds of the $\epsilon$-neighborhood obtained through IBP gives us a direct way to construct new artificial tasks when the number of available tasks is scarce.

**Manifold learning:** Traditional methods like ISOMAP (Tenenbaum et al., 2000), LLE (Roweis & Saul, 2000), t-SNE (Van der Maaten & Hinton, 2008), etc. aims to represent high-dimensional data in lower-dimensional space while preserving the local neighborhoods through manifold learning (Abukmeil et al., 2021). Recent manifold learning approaches mostly employ generative neural networks such as deep belief network (Lee et al., 2009), variational auto-encoders (Connor et al., 2021; Kumar & Poole, 2020), flow-based approaches (Brehmer & Cranmer, 2020; Caterini et al., 2021), etc. In a similar spirit, we repurpose IBP to define $\epsilon$-neighborhoods for few-shot learning tasks and constrain the learned feature embedding to preserve the said neighborhoods.

**Task augmentation:** To train on datasets with a limited number of tasks, some works directly impose regularization on the few-shot learner (Jamal & Qi, 2019; Yin et al., 2019). Another line of work performs data augmentation on the individual tasks (Lee et al., 2020; Ni et al., 2021; Yao et al., 2021). Finally, a third direction is to employ inter-task interpolation to mitigate task scarcity (Yao et al., 2022). Our approach is similar to the third category in that we directly create new artificial tasks. But, we also differs from all of the above-mentioned methods in that we neither undertake intra-task augmentation nor inter-task interpolation.

## 3   PRELIMINARIES

In a few-shot learning problem, we deal with tasks $\mathcal{T}_i \sim p(\mathcal{T})$. Each task $\mathcal{T}_i$ is associated with a dataset $\mathcal{D}_i = (X_i, Y_i)$, that we further subdivide into a support set $\mathcal{D}_i^s = (X_i^s, Y_i^s) = \{(\mathbf{x}_{i,r}^s, y_{i,r}^s)\}_{r=1}^{N_s}$ and a query set $\mathcal{D}_i^q = (X_i^q, Y_i^q) = \{(\mathbf{x}_{i,r}^q, y_{i,r}^q)\}_{r=1}^{N_q}$. Given a learning model $f_\theta$, where $\theta$ denotes the model parameters, few-shot learning algorithms attempt to learn $\theta$ to minimize the loss on the query

set $\mathcal{D}_i^q$ for each of the sampled tasks using the data-label pairs from the corresponding support set $\mathcal{D}_i^s$. Thereafter, the trained model $f_\theta$ and the support set $\mathcal{D}_j^s$ for new tasks $\mathcal{T}_j$ can be used to perform inference on the corresponding query set $\mathcal{D}_j^q$. In the following subsection, we discuss gradient-based meta-learning while the prototype-based metric-learning is detailed in Appendix A.

**Gradient-based meta-learning:** In gradient-based meta-learning, the aim is to learn initial parameters $\theta^*$ such that a typically small number of gradient update steps using the data-label pairs in the support set $\mathcal{D}_i^s$ results in a model $f_{\phi_i}$ that performs well on the query set of task $\mathcal{T}_i$. During the meta-training stage, first, a base learner is trained on multiple support sets $\mathcal{D}_i^s$, and the performance of the resulting models $f_{\phi_i}$ is evaluated on the corresponding query sets $\mathcal{D}_i^q$. The meta-learner parameters $\theta$ are then updated such that the expected loss of the base learner on query sets is minimized. In the meta-testing stage, the final meta-trained model $f_{\theta^*}$ is fine-tuned on the support set $\mathcal{D}_j^s$ for the given test task $\mathcal{T}_j$ to obtain the adapted model $f_{\phi_j}$ that can then be used for inference on the corresponding query set $\mathcal{D}_j^q$. Considering Model-Agnostic Meta-Learning (MAML) (Finn et al., 2017) as an example, the bi-level optimization of the gradient-based meta-learning is formulated as:

$$\theta^* \leftarrow \arg\min_\theta \mathbb{E}_{\mathcal{T}_i \sim p(\mathcal{T})}[\mathcal{L}(f_{\phi_i}; \mathcal{D}_i^q)], \text{ where } \phi_i = \theta - \eta_0 \nabla_\theta \mathcal{L}(f_\theta; \mathcal{D}_i^s), \qquad (1)$$

while $\eta_0$ denotes the inner-loop learning rate used by the base learner to train on $\mathcal{D}_i^s$ for task $\mathcal{T}_i$, and $\mathcal{L}$ is the loss function, which is usually the cross-entropy loss for classification problems:

$$\mathcal{L}_{CE} = \mathbb{E}_{\mathcal{T}_i \sim p(\mathcal{T})}[-\sum_r \log p(y_{i,r}^q | \mathbf{x}_{i,r}^q, f_{\phi_i})]. \qquad (2)$$

A key requirement for effective few-shot generalization to new tasks for both gradient-based meta-learning and prototype-based metric-learning is to learn a good embedding of the high-dimensional manifold characterizing the task distribution $p(\mathcal{T})$, i.e. the task manifold. Ideally, the learned embedding should conserve the neighborhoods from the high-dimensional task manifold (Tenenbaum et al., 2000; Roweis & Saul, 2000). Hence, in the following subsection, we discuss Interval Bound Propagation (IBP) (Gowal et al., 2019) that can be employed to define a neighborhood around a task.

**Interval bound propagation:** Let us consider a neural network $f_\theta$ consisting of a sequence of transformations $h_l, (l \in \{1, 2, \cdots, L\})$ for each of its $L$ layers. We start from an initial input $\mathbf{z}_0 = \mathbf{x}$ to the network along with lower bound $\underline{\mathbf{z}}_0(\epsilon) = \mathbf{x} - \mathbf{1}\epsilon$ and upper bound $\overline{\mathbf{z}}_0(\epsilon) = \mathbf{x} + \mathbf{1}\epsilon$ for an $\epsilon$-neighborhood around the input $\mathbf{x}$. In each of the subsequent layers $l \in \{1, 2, \cdots, L\}$ of the network, we get an activation $\mathbf{z}_l = h_l(\mathbf{z}_{l-1})$. IBP uses interval arithmetic to obtain the corresponding axis-aligned bounds of the form $\underline{\mathbf{z}}_l(\epsilon) \leq \mathbf{z}_l \leq \overline{\mathbf{z}}_l(\epsilon)$ on the activations for the $l$-th layer. Given the specific differentiable transformation $h_l$, interval arithmetic yields corresponding differentiable lower and upper bound transformations $\underline{\mathbf{z}}_l(\epsilon) = \underline{h}_l(\underline{\mathbf{z}}_{l-1}(\epsilon), \overline{\mathbf{z}}_{l-1}(\epsilon))$, and $\overline{\mathbf{z}}_l(\epsilon) = \overline{h}_l(\underline{\mathbf{z}}_{l-1}(\epsilon), \overline{\mathbf{z}}_{l-1}(\epsilon))$, as described in Appendix C. This ensures that each of the coordinates $\underline{z}_{l,c}(\epsilon)$ and $\overline{z}_{l,c}(\epsilon)$ of $\underline{\mathbf{z}}_l(\epsilon)$ and $\overline{\mathbf{z}}_l(\epsilon)$ respectively, satisfies the conditions:

$$\underline{z}_{l,c}(\epsilon) = \min_{\underline{\mathbf{z}}_{l-1}(\epsilon) \leq \mathbf{z}_{l-1} \leq \overline{\mathbf{z}}_{l-1}(\epsilon)} \mathbf{e}_c^\mathrm{T} h_l(\mathbf{z}_{l-1}) \text{ and } \overline{z}_{l,c}(\epsilon) = \max_{\underline{\mathbf{z}}_{l-1}(\epsilon) \leq \mathbf{z}_{l-1} \leq \overline{\mathbf{z}}_{l-1}(\epsilon)} \mathbf{e}_c^\mathrm{T} h_l(\mathbf{z}_{l-1}), \quad (3)$$

where $\mathbf{e}_c$ is the standard $c$-th basis vector. Further extending to multiple layers, such as $f_{\theta^S}$ having the first $S$ layers of $f_\theta$, the individual transformations $\underline{h}_l$ and $\overline{h}_l$ for $l \in \{1, 2, \cdots, S\}$ can be composed to obtain the corresponding functions $\underline{f}_{\theta^S}$ and $\overline{f}_{\theta^S}$, such that $\underline{\mathbf{z}}_S(\epsilon) = \underline{f}_{\theta^S}(\mathbf{z}_0, \epsilon)$, and $\overline{\mathbf{z}}_S(\epsilon) = \overline{f}_{\theta^S}(\mathbf{z}_0, \epsilon)$.

## 4 PROPOSED METHOD

Our proposed method aims to enable the learner $f_\theta$ to learn a feature embedding that attempts to preserve the $\epsilon$-neighborhoods in the task manifold. Therefore, in the following subsection, we describe the notion of an $\epsilon$-neighborhood for a training task $\mathcal{T}_i$ using IBP and show how they can be preserved to aid in few-shot learning problems.

### 4.1 FEW-SHOT LEARNING WITH INTERVAL BOUNDS

Consider the network $f_\theta = f_{\theta^{L-S}} \circ f_{\theta^S}$ where $S \ (\leq L)$ is a user-specified layer number that demarcates the boundary between the portion $f_{\theta^S}$ of the model that focuses on feature representation

and the subsequent portion $f_{\theta^{L-S}}$ responsible for the classification. For a given training task $\mathcal{T}_i$, the Euclidean distances between the embedding $f_{\theta^S}(\mathbf{x}^q_{i,r})$ for the query instances and their respective interval bounds $\underline{f}_{\theta^S}(\mathbf{x}^q_{i,r}, \epsilon)$ and $\overline{f}_{\theta^S}(\mathbf{x}^q_{i,r}, \epsilon)$ is a measure of how well the $\epsilon$-neighborhood is preserved in the learned feature embedding:

$$\mathcal{L}_{LB} = \frac{1}{N_q} \sum_{r=1}^{N_q} ||f_{\theta^S}(\mathbf{x}^q_{i,r}) - \underline{f}_{\theta^S}(\mathbf{x}^q_{i,r}, \epsilon)||_2^2 \text{ and} \tag{4}$$

$$\mathcal{L}_{UB} = \frac{1}{N_q} \sum_{r=1}^{N_q} ||f_{\theta^S}(\mathbf{x}^q_{i,r}) - \overline{f}_{\theta^S}(\mathbf{x}^q_{i,r}, \epsilon)||_2^2. \tag{5}$$

To ensure that the small $\epsilon$-neighborhoods get mapped to small interval bounds by the feature embedding $f_{\theta^S}$, we can minimize the losses $\mathcal{L}_{LB}$ and $\mathcal{L}_{UB}$ in addition to the classification loss $\mathcal{L}_{CE}$ in (2). Notice that the losses $\mathcal{L}_{LB}$ and $\mathcal{L}_{UB}$ are never used for the support instances $\mathbf{x}^s_{i,r}$.

Attempting to minimize a naïve sum of the three losses can cause some issues. For example, weighing the classification loss $\mathcal{L}_{CE}$ too high essentially reduces the proposed method to vanilla few-shot learning. On the other hand, assigning very high weights to the interval losses $\mathcal{L}_{LB}$ and/or $\mathcal{L}_{UB}$ may diminish learnability as the preservation of $\epsilon$-neighborhoods gets precedence over classification performance. Moreover, such static weighting approaches are not capable of adapting to (and consequently mitigating) situations where one of the losses comes to unduly dominate the others. Thus, we minimize a convex weighted sum $\mathcal{L}$ of the three losses:

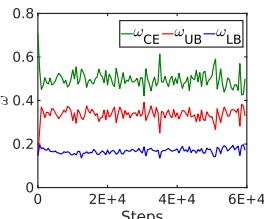

Figure 3: Dynamic weights for MAML+IBP on miniImageNet when $\gamma$ is set to 1 for ease of visualisation.

$$\mathcal{L}(t) = w_{CE}(t)\mathcal{L}_{CE}(t) + w_{LB}(t)\mathcal{L}_{LB}(t) + w_{UB}(t)\mathcal{L}_{UB}(t), \tag{6}$$

where $t$ denotes the current training step and $w_e(t)$ is the weight for the corresponding loss $\mathcal{L}_e$, $e \in \{CE, LB, UB\}$ at the $t$-th training step, which is dynamically calculated based on a softmax across the current values of the three losses:

$$w_e(t) = \frac{\exp(\mathcal{L}_e(t)/\gamma)}{\sum_{e' \in \{CE, LB, UB\}} \exp(\mathcal{L}_{e'}(t)/\gamma)}. \tag{7}$$

The hyperparameter $\gamma$ controls the relative importance of the losses. If any of the losses become too large, the dynamic weighing scheme strives to restore balance by assigning very high weightage to the concerned loss, thus prioritizing its minimization over that of the other losses. The changes in the dynamic weights over training steps for IBP-aided MAML (hereafter called MAML+IBP) using "4-CONV" network (Vinyals et al., 2016) on the miniImageNet dataset (Vinyals et al., 2016) is illustrated in Figure 3. We can observe that while there is a clear ordering to the magnitude of the weights (and therefore the corresponding losses) throughout the entire training run, the weights are in fact able to adapt to changes in loss values to maintain the status quo among the different losses.

**Motivating results:** In Table 1, we demonstrate the effect of employing IBP-aided training for MAML using "4-CONV" network. Apart from vanilla MAML, we consider three other regularized variants of MAML, (1) MAML+SN that applies Spectral Normalization (Miyato et al., 2018) to the feature embedding through the first $S$ layers, (2) MAML+GL that uses the distance between the original query set and its perturbed (by additive Gaussian noise) version as an extra loss, and (3) MAML+ULBL that considers the distance between the upper and lower interval bounds as an additional loss (Morawiecki et al., 2020) (further details in Appendix F.3). We see that MAML+IBP achieves higher 5-way 1-shot classification accuracy than the 5 contenders on the miniImageNet and tieredImageNet (Ren et al., 2018) datasets.

Table 1: Effect of IBP on MAML for miniImageNet and tieredImageNet datasets in terms of 5-way 1-shot Accuracy and intra-task compactness

| Algorithm | Accuracy | | 1-NN distance | |
|---|---|---|---|---|
| | miniImageNet | tieredImageNet | miniImageNet | tieredImageNet |
| MAML (Finn et al., 2017) | 48.70±1.75% | 51.67±1.81% | 0.97±0.02 | 0.98±0.02 |
| MAML+SN on $f_{\theta^S}$ | 44.90±1.12% | 45.26±1.05% | 1.38±0.04 | 1.41±0.04 |
| MAML+GL | 48.70± 0.97% | 51.90±0.98% | 0.96±0.02 | 0.98±0.02 |
| MAML+ULBL | 49.43±0.90% | 51.67±0.91% | 0.94±0.02 | 0.97±0.02 |
| MAML+IBP (ours) | **50.76±0.83%** | **54.36±0.80%** | **0.90±0.02** | **0.96±0.02** |

Moreover, we also illustrate that the feature embedding learned by IBP-aided training exhibits better intra-task compactness in terms of the mean Euclidean distances from the nearest neighbor in the same class for 100 query instances from 600 tasks, in the feature space characterized by $f_{\theta^S}$. Recent works (Ni et al., 2021; Yao et al., 2022) have shown that augmenting the training data with artificial tasks can improve performance in domains with a limited amount of tasks. Therefore, while IBP-aided training improves the performance of vanilla MAML (as well as other baselines, detailed in Appendix F.3), we are particularly interested in the added advantage that it lends by facilitating the generation of artificial tasks within the neighborhoods defined by the interval bounds.

## 4.2 Interval bound–based task interpolation

Since minimizing the additional losses $\mathcal{L}_{LB}$ and $\mathcal{L}_{UB}$ is expected to ensure that the $\epsilon$-neighborhood around a task is mapped to a small interval in the feature embedding space, artificial tasks formed within such intervals are naturally expected to be close to the task manifold. Therefore, we create additional artificial tasks by interpolating between an original task and its corresponding interval bounds (i.e., either the upper or the lower interval bound). In other words, for a training task $\mathcal{T}_i$, a corresponding artificial task $\mathcal{T}'_i$ is characterized by a support set $\mathcal{D}_i^{s'} = \{(\mathbf{H}_{i,r}^{s'}, \mathbf{y}_{i,r}^s)\}_{r=1}^{N_s}$ in the embedding space. The artificial support instances $\mathbf{H}_{i,r}^{s'}$ are created as:

$$\mathbf{H}_{i,r}^{s'} = (1 - \lambda_k) f_{\theta^S}(\mathbf{x}_{i,r}^s) + (1 - \nu_k)\lambda_k \underline{f}_{\theta^S}(\mathbf{x}_{i,r}^s, \epsilon) + \nu_k \lambda_k \overline{f}_{\theta^S}(\mathbf{x}_{i,r}^s, \epsilon), \tag{8}$$

where $k$ denotes the class to which $\mathbf{x}_{i,r}^s$ belongs, $\lambda_k \in [0, 1]$ is sampled from a Beta distribution $Beta(\alpha, \beta)$, and the random choice of $\nu_k \in \{0, 1\}$ dictates which of the bounds is chosen randomly for each class. The labels $\mathbf{y}_{i,r}^s$ for the artificial task remain identical to that of the original task. The query set $\mathcal{D}_i^{q'}$ for the artificial task is also constructed analogously. We then minimize the mean of the additional classification loss $\mathcal{L}'_{CE}$ for the artificial task $\mathcal{T}'_i$ and the classification loss $\mathcal{L}_{CE}$ for the original task $\mathcal{T}_i$ for query instances (also the support instances in case of meta-learning). As a reminder, the losses $\mathcal{L}_{LB}$ and $\mathcal{L}_{UB}$ are also additionally minimized for the query instances. The complete IBP-based task interpolation or Interval Bound Interpolation (IBI) training setup is illustrated in Figure 4 in Appendix B. Since IBI does not play any part during the testing phase, the testing recipe remains identical to that of vanilla few-shot learning. The detailed pseudocode of MAML+IBI (along with the IBI variant of ProtoNet) can be found in Appendix B.

**Theoretical analysis:** The data $X_i$ ($i = 1, 2, \cdots, N$) for tasks $\mathcal{T}_i$ can be thought of as i.i.d. observations from a marginal distribution $\mathbb{P}_X$ defined on a compact subset $\mathcal{X}$ of $\mathbb{R}^d$ ($d \geq 1$), paired with corresponding $Y_i$ drawn from the marginal distribution $\mathbb{P}_Y$. The map $f_{\theta^S}$ is bestowed with the task of producing a lower-dimensional representation of the input $X$. Let us denote the embedding space by $\mathcal{H} \subseteq \mathbb{R}^\kappa$, given that $\kappa \leq d$. The spaces $\mathcal{X}$ and $\mathcal{H}$ are endowed with $l_2$ norm for simplicity and conformity to our convention. One may observe that $f_{\theta^S} = h_1 \circ h_2 \circ \cdots \circ h_S$, where in general $h_l(\mathbf{z}) = \sigma(A_l \mathbf{z} + \mathbf{b}_l)$ given that $A_l \in \mathbb{R}^{d_{l+1} \times d_l}$ and $\mathbf{b}_l \in \mathbb{R}^{d_{l+1}}$, $l = 1, \cdots, S$. The function $\sigma$ denotes the activation (such as ReLU), applied component-wise. Evidently, in our notation $d_1 = d$ and $d_{S+1} = \kappa$. With this setup, we proceed to undertake the theoretical analysis of our approach. Please find the detailed proofs in Appendix C.

**Definition 1** (Perturbation). *Given any* $\mathbf{x}_1 \in \mathcal{X}$, *an* $\varepsilon$-*perturbation corresponding to* $\mathbf{x}_1$ *is the set of points* $\mathbf{x}_1(\varepsilon) \subset \mathcal{X}$ *such that* $\|\mathbf{x}_1 - \mathbf{x}_2\| = \varepsilon$, $\forall \mathbf{x}_2 \in \mathbf{x}_1(\varepsilon)$; $\varepsilon > 0$.

For the particular choice of the $l_2$ norm, Definition 1 characterizes $\varepsilon$-perturbation as a hollow ball of radius $\varepsilon = \epsilon\sqrt{d}$ around a given point.

**Lemma 1** (Lipschitz networks ensure bounded IBP). *Let* $\overline{\mathbf{x}}$ *and* $\underline{\mathbf{x}}$ *be* $\varepsilon$-*perturbations of* $\mathbf{x} \sim \mathbb{P}_X$ *for an* $\varepsilon > 0$ *(i.e.* $\overline{\mathbf{x}}, \underline{\mathbf{x}} \in \mathbf{x}(\varepsilon)$). *Given that the activation* $\sigma$ *is Lipschitz continuous (such as ReLU) with constant* $c_\sigma > 0$, *there exists a constant* $D = D(c_\sigma; A_1, A_2, \cdots, A_S; \varepsilon)$ *such that* $\underline{f}_{\theta^S}(\mathbf{x}, \varepsilon)$ *and* $\overline{f}_{\theta^S}(\mathbf{x}, \varepsilon))$ *will at most be an* $\hat{\varepsilon}$-*perturbed version of* $f_{\theta^S}(\mathbf{x})$, *where* $\hat{\varepsilon} = \varepsilon D$.

The minimization objective function of IBI can be rephrased as $\mathcal{L} = \mathcal{L}_{CE} + \omega_1 \mathcal{L}_{LB} + \omega_2 \mathcal{L}_{UB}$, where $\omega_1, \omega_2 \geq 0$ are Lagrangian multipliers. The forthcoming result, however, relies on the constrained formulation of the objective, given as $\min\{\mathcal{L}_{CE}\}$ subject to $\mathcal{L}_{LB} \leq t_1$ and $\mathcal{L}_{UB} \leq t_2$, where $t_1, t_2 \geq 0$. This is motivated by the fact that the constrained formulation yields solutions upper bounding the ones obtained using its Lagrangian counterpart [(Boyd & Vandenberghe, 2004), Chapter

5]. Lemma 1 implies that the two losses ($\mathcal{L}_{UB}$ and $\mathcal{L}_{LB}$) appearing in the constraints can always be made arbitrarily small, depending upon $\varepsilon$. As such, in the constrained regime, the remaining problem is to show that the multi-task sample classification loss can indeed be dealt with.

**Theorem 1** (Generalization bound). *Let $\tilde{\mathbb{P}}$ be the joint distribution of $(f_{\theta^S}(X), Y)$, supported on $\mathcal{H} \times \mathbb{R}$. In the multi-task regime, let $I$ denote the set of tasks, each consisting of $N$ samples. Define $\hat{\mathcal{R}}(N, |I|) = \mathbb{E}_{\mathcal{T}_i \sim \hat{p}(\mathcal{T})} \mathbb{E}_{(X_j, Y_j) \sim \hat{p}(\mathcal{T}_i)} [\mathcal{L}_{CE}(f_{\theta^{L-S}}(\mathbf{H}_j^*), Y_j)]$ and $\mathcal{R} = \mathbb{E}_{\mathcal{T}_i \sim p(\mathcal{T})} \mathbb{E}_{(X_j, Y_j) \sim \mathcal{T}_i} [\mathcal{L}_{CE}(f_{\theta^{L-S}}(f_{\theta^S}(X_j)), Y_j)]$. For a bounded loss function $\mathcal{L}_{CE} : \mathbb{R} \times \mathbb{R} \to [0, a](a \geq 0)$, if the neural network-induced map $f_{\theta^{L-S}}$ is such that $|\nabla f_{\theta^{L-S}}(\cdot)| < \infty$, we ensure:*

$$\left| \hat{\mathcal{R}}(N, |I|) - \mathcal{R} \right| - \tilde{\lambda} \precsim 2^{L-S+1} \sqrt{2 \log(2\kappa + 2)} \left\{ \frac{1}{\sqrt{N}} + \frac{1}{\sqrt{|I|}} \right\} + \sqrt{\frac{\log(\frac{2|I|}{\delta})}{N}} + \sqrt{\frac{\log(\frac{2}{\delta})}{|I|}}$$

*holds with probability at least $1 - \delta$, where $\tilde{\lambda} = \tilde{\lambda}(\hat{\varepsilon}, \lambda)$.*

## 5 EXPERIMENTS

**Experiment protocol:** The experiments are conducted on few-task few-shot image classification datasets, viz. a subset of the miniImageNet dataset called miniImageNet-S (Yao et al., 2022), and two medical images datasets namely DermNet-S (Yao et al., 2022), and ISIC (Codella et al., 2018; Li et al., 2020). We begin our experiments with a few analyses and ablations to better understand the properties of our proposed method. We then empirically demonstrate the effectiveness of our proposed IBI method on the gradient-based meta-learning method MAML (Finn et al., 2017) as well as the prototype-based metric-learner ProtoNet (Snell et al., 2017) to show that IBI can be seamlessly integrated with multiple few-shot learning paradigms. For our experiments, we employ the commonly used "4-CONV" network (Vinyals et al., 2016) as well as the larger ResNet-12 network (Lee et al., 2019) to demonstrate the scalability of the proposed method (further details on scalability in Appendix E). We perform 5-way 1-shot and 5-way 5-shot classification on all the above datasets (except ISIC where we use 2-way classification problems, similar to Yao et al. (2021), due to the lack of sufficient training classes). Further discussion on the datasets and implementation details of IBI along with the choice of hyperparameters can be found in the Appendix.

**Ablation studies on task interpolation:** We undertake an ablation study to highlight the importance of generating artificial tasks using IBP bound–based interpolation by comparing IBI with (1) inter-task interpolation on images, (2) inter-task interpolation in the feature embedding learned by $f_\theta^S$, (3)

Table 2: Ablation on task interpolation strategies in terms of mean Accuracy and average median distance between original and interpolated tasks over 600 tasks on miniImageNet-S (mIS), ISIC and DermNet-S (DS).

| Algorithm | Accuracy | | | Average median distance | | |
|---|---|---|---|---|---|---|
| | mIS | ISIC | DS | mIS | ISIC | DS |
| MAML+Inter-task interpolation in image space | 40.90% | 55.25% | 48.30% | N/A | N/A | N/A |
| MAML+Inter-task interpolation after $f_{\theta^S}$ | 41.00% | 61.33% | 47.43% | 3.08 | 1.23 | 2.99 |
| MAML+IBP+WCL | 41.56% | 64.75% | 48.90% | NA | NA | NA |
| MAML+ULBL+Inter-task interpolation after $f_{\theta^S}$ | 40.37% | 64.91% | 48.23% | 3.10 | 0.97 | 2.83 |
| MAML+IBP+GA (Image Space) | 41.83% | 62.67% | 48.83% | NA | NA | NA |
| MAML+IBP+GA (after $f_{\theta^S}$) | 41.66% | 63.75% | 47.60% | NA | NA | NA |
| MAML+MLTI (Yao et al., 2022) | 41.58% | 61.79% | 48.03% | 3.24 | 1.36 | 3.05 |
| MAML+IBI without $\mathcal{L}_{UB}$ and $\mathcal{L}_{LB}$ losses | 35.26% | 48.94% | 41.30% | N/A | N/A | N/A |
| MAML+IBI (ours) | **42.20%** | **68.58%** | **49.13%** | **2.74** | **0.60** | **2.65** |

The Average median distance is calculated with features after the third block for all cases.

Worst-Case Loss (WCL) on the $\epsilon$-neighbourhood (Gowal et al., 2019) along with IBP losses, (4) inter-task interpolation while minimizing ULBL (Morawiecki et al., 2020), (5) Gaussian noise–based perturbation (GA) in the image space with IBP losses, (6) Gaussian noise–based perturbation in the feature embedding space $f_{\theta^S}$ with IBP losses, (7) MLTI (Yao et al., 2022), which performs MixUp (Zhang et al., 2017) at randomly chosen layers of the learner, and (8) IBP bound–based interpolation without minimizing the $\mathcal{L}_{UB}$ and $\mathcal{L}_{LB}$ while only optimizing $\mathcal{L}_{CE}$ (more results in Appendix F.3). We perform the ablation study on 5-way 1-shot classification with the "4-CONV" network on miniImageNet-S, ISIC, and DermNet-S. From Table 2, we observe that IBI performs best in all cases. Moreover, inter-class interpolation at the same fixed layer $S$ as IBI and at randomly selected task-specific layers in MLTI shows worse performance, demonstrating the superiority of the proposed interval bound–based interpolation mechanism. Further, it is interesting to observe that IBI,

when performed without minimizing the $\mathcal{L}_{UB}$ and $\mathcal{L}_{LB}$, performs the worst. This behavior is not unexpected as the neighborhoods are no longer guaranteed to be preserved by the learned embedding in this case, thus potentially resulting in the generation of out-of-manifold artificial tasks.

To further check whether the tasks generated by IBI indeed follow the distribution, we undertake an additional comparison based on the similarity of the artificial tasks with the corresponding original tasks. Concretely, we define the distance between a task and its artificial counterpart as the median of the pairwise distances between the corresponding data instances in the two tasks. If an artificial task is created by combining two tasks, *a la* MLTI (Yao et al., 2022), we consider the smaller of the two median distances. We observe from Table 2, that the average median distance over 600 tasks is smaller for the proposed method compared to MLTI, as well as inter-task interpolation in the feature embedding learned by $f_\theta^S$. This indicates that the tasks generated by IBI are more likely to lie close to the original task distribution.

**Importance of dynamic loss weighting:** To validate the usefulness of softmax-based dynamic weighting of the three losses for both IBP and IBI, we first find the average weights for each loss in a dynamic weight run and then plug in the respective values as static weights for new runs. All experiments in Table 3 are conducted on the miniImageNet-S dataset. From the upper half of Table 3, we can see that the three average weights are always distinct with a definite trend in that $\mathcal{L}_{CE}$ gets maximum importance followed by $\mathcal{L}_{UB}$ while $\mathcal{L}_{LB}$ contributes very little to the total loss $\mathcal{L}$.

Table 3: Average loss weights for the two proposed methods, and a comparison of the static weighting and dynamic weighting versions, including transferability of static weight values across variants.

| | MAML+IBP | MAML+IBI |
|---|---|---|
| Average of dynamic loss weights calculated for IBP and IBI. | | |
| $w_{CE}$ | 0.8600 | 0.8658 |
| $w_{UB}$ | 0.1369 | 0.1314 |
| $w_{LB}$ | 0.0029 | 0.0027 |
| Accuracy of algorithms with different weight choices. | | |
| Dynamic weighting | **41.30±0.79%** | **42.20±0.82%** |
| Static average weights for MAML+IBP | 40.55±0.81% | N/A |
| Static average weights for MAML+IBI | N/A | 40.72±0.79% |

This may be due to the particular "4-CONV" architecture used in this study which employs ReLU activations, thus implicitly limiting the spread of the lower bound (Gowal et al., 2019). Further, the average weights of IBP and IBI are similar for a particular learner highlighting their commonalities, while they are distinct over different learners stressing their learner-dependent behavior. Further, in the lower half of Table 3, we explore the effect of using static weights as well as the transferability of the loss weights across learners. In all cases, the softmax-based dynamic weighting outperforms static weighting, thus demonstrating the importance of dynamic weighting.

**Results on few-task few-shot classification problems:** For evaluating the few-shot classification performance of IBI in few-task situations, we compare against the regularization-based meta-learning methods TAML (Jamal & Qi, 2019), Meta-Reg (Yin et al., 2019), and Meta-Dropout (Lee et al., 2020) for MAML. We also compare against data augmentation–based methods like MetaMix (Yao et al., 2021), Meta-Maxup (Ni et al., 2021), and MLTI (Yao et al., 2022) for both MAML and ProtoNet. The results in Table 4 show that in keeping with the observation in Table 1, IBP without task interpolation can improve upon the corresponding baselines. The incorporation of IBP-based task interpolation in IBI generally improves the results even further. Overall, we observe that both IBP and IBI outperform the other competitors, with the largest gains being observed for the ISIC dataset.

**Cross-domain transferability analysis:** The miniImageNet-S and DermNet-S datasets both allow 5-way 1-shot classification. Moreover, miniImageNet-S contains images from natural scenes while DermNet-S consists of medical images. Therefore, we undertake a cross-domain transferability study in Table 5. We summarize the Accuracy values obtained by a source model trained on DermNet-S but tested on miniImageNet-S and vice-versa (denoted DS $\rightarrow$ mIS and mIS $\rightarrow$ DS, respectively). We can see that in most cases the IBP variant is able to improve upon the corresponding baseline. Further, the interpolation-based methods, i.e. MLTI

Table 5: Transferability comparison of MAML and ProtoNet, with their MLTI, IBP, and IBI variants in terms of Accuracy over 600 tasks.

| Algorithms | Accuracy | |
|---|---|---|
| | DS $\rightarrow$ mIS | mIS $\rightarrow$ DS |
| MAML | 25.06% | 33.40% |
| MAML+MLTI (Yao et al., 2022) | 30.03% | **36.74%** |
| MAML+IBP (ours) | 27.06% | 33.90% |
| MAML+IBI (ours) | **30.23%** | 36.21% |
| ProtoNet | 28.76% | 34.03% |
| ProtoNet*+MLTI (Yao et al., 2022) | 30.06% | 35.46% |
| ProtoNet+IBP (ours) | 29.60% | 34.13% |
| ProtoNet+IBI (ours) | **30.32%** | **35.63%** |

*: ProtoNet implementation as per Yao et al. (2022).

Table 4: Performance comparison of the proposed method with baselines and contending algorithms in terms of mean Accuracy over 600 tasks.

| Backbone Network | Algorithm | miniImageNet-S | | ISIC | | DermNet-S | |
|---|---|---|---|---|---|---|---|
| | | 1-shot | 5-shot | 1-shot | 5-shot | 1-shot | 5-shot |
| 4-CONV | MAML (Finn et al., 2017; Yao et al., 2022) | 38.27% | 52.14% | 57.59% | 65.24% | 43.47% | 60.56% |
| | MAML+Meta-Reg (Yin et al., 2019; Yao et al., 2022) | 38.35% | 51.74% | 58.57% | 68.45% | 45.01% | 60.92% |
| | TAML (Jamal & Qi, 2019; Yao et al., 2022) | 38.70% | 52.75% | 58.39% | 66.09% | 45.73% | 61.14% |
| | MAML+Meta-Dropout (Lee et al., 2020; Yao et al., 2022) | 38.32% | 52.53% | 58.40% | 67.32% | 44.30% | 60.86% |
| | MAML+MetaMix (Yao et al., 2021; 2022) | 39.43% | 54.14% | 60.34% | 69.47% | 46.81% | 63.52% |
| | MAML+Meta-Maxup (Ni et al., 2021; Yao et al., 2022) | 39.28% | 53.02% | 58.68% | 69.16% | 46.10% | 62.64% |
| | MAML+MLTI (Yao et al., 2022) | 41.58% | 55.22% | 61.79% | 70.69% | 48.03% | 64.55% |
| | MAML+IBP (ours) | 41.30% | 54.36% | 64.91% | 78.75% | 48.33% | 63.33% |
| | MAML+IBI (ours) | **42.20%** | **55.23%** | **68.58%** | **79.75%** | **49.13%** | **65.43%** |
| | ProtoNet* (Snell et al., 2017; Yao et al., 2022) | 36.26% | 50.72% | 58.56% | 66.25% | 44.21% | 60.33% |
| | ProtoNet (Snell et al., 2017) | 40.70% | 53.16% | 65.58% | 75.25% | 46.86% | 62.03% |
| | ProtoNet*+MetaMix (Yao et al., 2021; 2022) | 39.67% | 53.10% | 60.58% | 70.12% | 47.71% | 62.68% |
| | ProtoNet*+Meta-Maxup (Ni et al., 2021; Yao et al., 2022) | 39.80% | 53.35% | 59.66% | 68.97% | 46.06% | 62.97% |
| | ProtoNet*+MLTI (Yao et al., 2022) | 41.36% | 55.34% | 62.82% | 71.52% | 49.38% | 65.19% |
| | ProtoNet+IBP (ours) | 41.46% | 55.00% | **70.75%** | 81.01% | 48.66% | **67.26%** |
| | ProtoNet+IBI (ours) | **43.30%** | **55.73%** | 70.25% | **81.16%** | **51.13%** | 65.93% |
| ResNet-12 | MAML (Finn et al., 2017; Yao et al., 2022) | 40.02% | 52.56% | 59.41% | 67.66% | 47.58% | 63.13% |
| | MAML+MetaMix (Yao et al., 2021; 2022) | 42.26% | 54.65% | 62.06% | 72.18% | 51.40% | 64.82% |
| | MAML+MetaMaxup (Ni et al., 2021; Yao et al., 2022) | 41.97% | 53.92% | 61.64% | 72.04% | 50.82% | 64.24% |
| | MAML+MLTI (Yao et al., 2022) | 43.35% | 54.89% | 62.16% | 73.56% | 52.03% | 65.12% |
| | MAML+IBP (ours) | 43.50% | 55.13% | **64.50%** | 73.91% | 50.40% | 65.40% |
| | MAML+IBI (ours) | **43.90%** | **57.00%** | 63.25% | **75.66%** | **52.10%** | **66.50%** |
| | ProtoNet* (Snell et al., 2017; Yao et al., 2022) | 40.96% | 53.77% | 61.91% | 72.97% | 48.65% | 64.61% |
| | ProtoNet (Snell et al., 2017) | 42.60% | 55.00% | 63.01% | 75.91% | 50.66% | 65.40% |
| | ProtoNet*+MetaMix (Yao et al., 2021; 2022) | 42.95% | 56.95% | 65.55% | 78.33% | 51.18% | 66.80% |
| | ProtoNet*+MetaMaxup (Ni et al., 2021; Yao et al., 2022) | 42.68% | 56.07% | 64.17% | 77.62% | 50.96% | 66.38% |
| | ProtoNet*+MLTI (Yao et al., 2022) | 44.08% | 57.14% | 66.02% | 79.15% | 52.01% | 67.28% |
| | ProtoNet+IBP (ours) | 43.33% | 57.40% | 66.66% | 81.00% | 51.33% | 67.57% |
| | ProtoNet+IBI (ours) | **45.33%** | **58.23%** | **66.75%** | **81.83%** | **52.53%** | **68.00%** |

* ProtoNet implementation as per Yao et al. (2022).

and IBI are able to further improve performance, with IBI achieving the best performance in most cases, thus showing that IBI training can improve cross-domain transferability.

# 6 CONCLUSION AND FUTURE WORKS

In this paper, we attempt to explore the utility of IBP beyond its originally-intended usage for building and verifying classifiers that are provably robust against adversarial attacks. In summary, we identify the potential of IBP to conserve a neighborhood from the input image space to the learned feature space through the layers of a deep neural network by minimizing the distances of the feature embedding from the two bounds. This can be effective in few-shot classification problems to obtain feature embeddings where task neighborhoods are preserved, thus enabling easy adaptability to unseen tasks. Further, since interpolating between training tasks and their corresponding IBP bounds can yield artificial tasks with a higher chance of lying on the task manifold, we exploit this property of IBP to prevent overfitting to seen tasks in the few-task scenario. The resulting IBI training scheme is shown to be effective in both the meta-learning and metric-learning paradigms of few-shot learning.

We demonstrate in our results that IBI can be effectively scaled to relatively large networks like ResNet-12 as we typically only need to apply IBP to a few initial layers (see Appendix E). However, this still adds extra computational cost (see Appendix E for a comparative study) which scales linearly with the number of layers subjected to IBP. Therefore, to limit the additional complexity and computational cost, a future direction of research may be to investigate the applicability of more advanced provably robust training methods that yield more efficient and tighter bounds (Lyu et al., 2021). Moreover, few-shot learners can also be improved with adaptive hyperparameters (Baik et al., 2020), feature reconstruction (Lee & Chung, 2021), knowledge distillation (Tian et al., 2020), embedding propagation (Rodríguez et al., 2020), etc. Thus, it may be interesting to observe the performance gains from these orthogonal techniques when coupled with IBI. However, this may not be a straightforward endeavor, given the complex dynamic nature of such frameworks.

## REPRODUCIBILITY STATEMENT

We have included the pseudo-codes and PyTorch based Python implementation for the proposed method in Appendices B and G, respectively. The description of all datasets used in this study along with other key implementation details is available in Appendices D, E. The hyperparameter settings for different algorithms along with their tuning strategy are listed in Appendix F. For the theoretical analyses, complete proofs are provided in Appendix C. A copy of the code is available in the Appendix G while the same can also be found at `https://anonymous.4open.science/r/maml-ibp-ibi-D072/`.

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

## A  PROTOTYPE-BASED METRIC-LEARNING:

Metric-based few-shot learning aims to obtain a feature embedding of the task manifold suitable for non-parametric classification. Prototype-based metric-learning, specifically Prototypical Network (ProtoNet) (Snell et al., 2017), assigns a query point to the class having the nearest (in terms of Euclidean distance) prototype in the learned embedding space. Given the model $f_\theta$ and a task $\mathcal{T}_i$, we first compute class prototypes $\{\mathbf{c}_k\}_{k=1}^K$ as the mean of $f_\theta(\mathbf{x}_{i,r}^s)$ for the instances $\mathbf{x}_{i,r}^s$ belonging to class $k$:

$$\mathbf{c}_k = \frac{1}{N_s} \sum\nolimits_{(\mathbf{x}_{i,r}^s, y_{i,r}^s) \in \mathcal{D}_i^{s,k}} f_\theta(\mathbf{x}_{i,r}^s), \tag{9}$$

where $\mathcal{D}_i^{s,k} \subset \mathcal{D}_i^s$ represents the subset of $N_s$ support samples from class $k$. Given a sample $\mathbf{x}_{i,r}^q$ from the query set, the probability $p(y_{i,r}^q = k|\mathbf{x}_{i,r}^q)$ of assigning it to the $k$-th class is calculated using the distance function $d(.,.)$ between the representation $f_\theta(\mathbf{x}_{i,r}^q)$ and the prototype $\mathbf{c}_k$:

$$p(y_{i,r}^q = k|\mathbf{x}_{i,r}^q, f_\theta) = \frac{\exp(-d(f_\theta(\mathbf{x}_{i,r}^q), \mathbf{c}_k))}{\sum_{k'} \exp(-d(f_\theta(\mathbf{x}_{i,r}^q), \mathbf{c}_{k'}))}. \tag{10}$$

Thereafter, the parameters $\theta$ for the model $f_\theta$ can be trained by minimizing cross-entropy loss (2). In the testing stage, each query sample $\mathbf{x}_{j,r}^q$ is assigned to the class having the maximal probability, i.e., $\mathbf{y}_{j,r}^q = \arg\max_k p(y_{j,r}^q = k|\mathbf{x}_{j,r}^q)$.

## B  ALGORITHMS OF MAML AND PROTONET COUPLED WITH IBP AND IBI

The following Figure 4 illustrates a schematic diagram for the training of IBP and IBI variants.

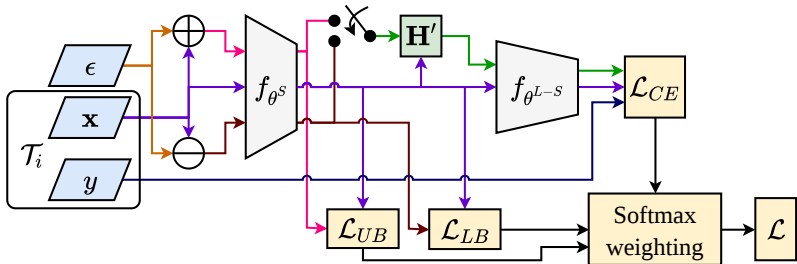

Figure 4: Interval bound propagation–based few-shot training (best viewed in color): For each query data-label pair $(\mathbf{x}, y)$ in a given training task $\mathcal{T}_i$, we start by defining a $\epsilon$-neighborhood $[\mathbf{x} - \mathbf{1}\epsilon, \mathbf{x} + \mathbf{1}\epsilon]$ around $\mathbf{x}$. The bounding box $[\underline{f}_{\theta^s}(\mathbf{x}, \epsilon), \overline{f}_{\theta^s}(\mathbf{x}, \epsilon)]$ around the embedding $f_{\theta^s}(\mathbf{x})$ after the first $S$ layers of the learner is found using IBP. In addition to the classification loss $\mathcal{L}_{CE}$, we also minimize the losses $\mathcal{L}_{LB}$ and $\mathcal{L}_{UB}$ which respectively measure the distances of $f_{\theta^s}(\mathbf{x})$ to $\underline{f}_{\theta^s}(\mathbf{x}, \epsilon)$ and $\overline{f}_{\theta^s}(\mathbf{x}, \epsilon)$. A softmax across the three loss values is used to dynamically calculate the convex weights for the losses, so as to prioritize the minimization of the dominant loss(es) at any given training step. For IBP-based interpolation, artificial tasks $\mathcal{T}_i'$ are created with instances $\mathbf{H}'$ formed by interpolating both the support and query instances with their corresponding lower or upper bounds. The mean of the classification loss $\mathcal{L}_{CE}$ for the $\mathcal{T}_i$ and the corresponding extra loss $\mathcal{L}_{CE}'$ for $\mathcal{T}_i'$ is minimized.

The steps for MAML+IBP/IBI and ProtoNet+IBP/IBI are respectively presented in Algorithm 1 and 2. Please consult the main paper for various notations and equations used in the algorithms.

**Remark 1.** The way in which the training support set $\mathcal{D}_i^s$ informs the loss calculation on the corresponding query set $\mathcal{D}_i^q$ differs between the MAML and ProtoNet variants. While a limited number of training steps on the support set is undertaken to obtain the model $f_{\phi_i}$ where the loss is calculated on the query set for MAML, the support set is used to calculate the prototypes $\{\mathbf{c}_k\}_{k=1}^K$ for the loss calculation on the query set for ProtoNet.

---

**Algorithm 1** IBP/IBI for MAML training

---

**Requires:** Task distribution $p(\mathcal{T})$, batch size $B$, learning rates $\eta_0$ and $\eta_1$, interval coefficient $\epsilon$.

---

1: Randomly initialize the meta-learner parameters $\theta$.
2: **while** not converged **do**
3:     Sample a batch of $B$ tasks from the distribution $\rho(\mathcal{T})$.
4:     For IBI, randomly sample an index $1 \leq m \leq B$ to perform the interpolation.
5:     **for all** $i \in \{1, 2, \cdots, B\}$ **do**
6:         Initialize base learner to meta-learner state.
7:         Sample a support set $\mathcal{D}_i^s$ of data-label pairs $\{(\mathbf{x}_{i,r}^s, \mathbf{y}_{i,r}^s)\}_{r=1}^{N_s}$ from task $\mathcal{T}_i$.
8:         Calculate the classification loss $\mathcal{L}_{CE}$ using $f_\theta(\mathbf{x}_{i,r}^s)$ and $\mathbf{y}_{i,r}^s$.
9:         **if** $i = m$ **then**
10:             Generate interpolated support and query instances $\mathbf{H}_{i,r}^{s'}$ and $\mathbf{H}_{i,r}^{s'}$ using (8).
11:             Calculate classification loss $\mathcal{L}'_{CE}$ using $f_{\theta^{L-S}}(\mathbf{H}_{i,r}^{s'})$ and $\mathbf{y}_{i,r}^s$.
12:             Set $\mathcal{L}_{CE} = \frac{1}{2}(\mathcal{L}_{CE} + \mathcal{L}'_{CE})$.
13:         **end if**
14:         Update base learner parameters to $\phi_i = \theta - \eta_0 \nabla_\theta \mathcal{L}_{CE}$.
15:         Sample a query set $\mathcal{D}_i^q$ of data-label pairs $\{(\mathbf{x}_{i,r}^q, \mathbf{y}_{i,r}^q)\}_{r=1}^{N_q}$ from task $\mathcal{T}_i$.
16:         Calculate the classification loss $\mathcal{L}_{CE}$ with $f_{\phi_i}(\mathbf{x}_{i,r}^q)$ and $\mathbf{y}_{i,r}^q$.
17:         Calculate $\mathcal{L}_{LB}$ and $\mathcal{L}_{UB}$ respectively using (4) and (5).
18:         **if** $i = m$ **then**
19:             Calculate classification loss $\mathcal{L}'_{CE}$ using $f_{\phi^{L-S}}(\mathbf{H}_{i,r}^{q'})$ and $\mathbf{y}_{i,r}^q$.
20:             Set $\mathcal{L}_{CE} = \frac{1}{2}(\mathcal{L}_{CE} + \mathcal{L}'_{CE})$.
21:         **end if**
22:         Calculate $\mathcal{L}$ by accumulating $\mathcal{L}_{CE}$, $\mathcal{L}_{LB}$ and $\mathcal{L}_{UB}$ using (6).
23:     **end for**
24:     Update meta-learner parameters $\theta = \theta - \eta_1 \frac{1}{B} \sum_{i=1}^{B} \nabla_\theta \mathcal{L}$.
25: **end while**

---

**Algorithm 2** IBP/IBI for ProtoNet training

---

**Requires:** Task distribution $p(\mathcal{T})$, learning rate $\eta$, interval coefficient $\epsilon$.

---

1: Randomly initialize the learner parameters $\theta$.
2: **while** not converged **do**
3:     For IBI, randomly select if interpolation is to be performed.
4:     Sample a support set $\mathcal{D}_i^s$ of data-label pairs $\{(\mathbf{x}_{i,r}^s, \mathbf{y}_{i,r}^s)\}_{r=1}^{N_s}$ from task $\mathcal{T}_i$.
5:     Calculate the features $f_{\theta^L}(\mathbf{x}_{i,r}^s)$ and find the prototypes $\{\mathbf{c}_k\}_{k=1}^{K}$ using (9).
6:     **if** interpolation to be performed **then**
7:         Generate interpolated support and query instances $\mathbf{H}_{i,r}^{s'}$ and $\mathbf{H}_{i,r}^{s'}$ using (8).
8:         Calculate features $f_{\theta^{L-S}}(\mathbf{H}_{i,r}^{s'})$ and find prototypes $\{\mathbf{c}'_k\}_{k=1}^{K}$.
9:     **end if**
10:     Sample a query set $\mathcal{D}_i^q$ of data-label pairs $\{(\mathbf{x}_{i,r}^q, \mathbf{y}_{i,r}^q)\}_{r=1}^{N_q}$ from task $\mathcal{T}_i$.
11:     Calculate the loss $\mathcal{L}_{CE}$ using (10) and (2).
12:     Calculate $\mathcal{L}_{LB}$ and $\mathcal{L}_{UB}$ using (4) and (5).
13:     **if** interpolation to be performed **then**
14:         Calculate classification loss $\mathcal{L}'_{CE}$ with $f_{\theta^{L-S}}(\mathbf{H}_{i,r}^{q'})$, $\{\mathbf{c}'_k\}_{k=1}^{K}$ and $\mathbf{y}_{i,r}^q$ by (10) and (2).
15:         Set $\mathcal{L}_{CE} = \frac{1}{2}(\mathcal{L}_{CE} + \mathcal{L}'_{CE})$.
16:     **end if**
17:     Calculate $\mathcal{L}$ by accumulating $\mathcal{L}_{CE}$, $\mathcal{L}_{LB}$ and $\mathcal{L}_{UB}$ using (6).
18:     Update learner parameters $\theta = \theta - \eta \nabla_\theta \mathcal{L}$.
19: **end while**

---

## C   DETAILED THEORETICAL ANALYSIS

**Interval bound propagation for networks with affine layer:** Let us assume a network $f$ with $L$ layers where the 0-th layer denotes the initial input. Let us also consider a layer $l \leq L$ that is not the 0-th input layer. The 0-th layer of $f$ takes the input along with its perturbed counterparts as shown in Section 3 in the main paper. If at the end of $l-1$-th layer the activation, upper bound, and lower bound are respectively $\mathbf{z}_{l-1}, \overline{\mathbf{z}}_{l-1}$ and $\underline{\mathbf{z}}_{l-1}$. If the $l$-th layer performs an affine transformation (such as a convolutional, fully connected, batch normalization, etc.) followed by a monotonic activation function (such as ReLU, sigmoid, tanh, etc.) i.e. $\mathbf{z}_l = \sigma(A_l \mathbf{z}_{l-1} + \mathbf{b}_l)$, then as per Gowal et al. (2019), we can calculate the interval bounds for the subsequent $l$-th layer as follows:

$$\underline{\mathbf{z}}_l = \sigma(\mu_l - \psi_l), \tag{11}$$

$$\overline{\mathbf{z}}_l = \sigma(\mu_l + \psi_l), \tag{12}$$

where $\psi_l = |A_l|\psi_{l-1}$ and $\mu_l = A_l\mu_{l-1} + \mathbf{b}_l$ given $\mu_{l-1} = \frac{\underline{\mathbf{z}}_{l-1}+\overline{\mathbf{z}}_{l-1}}{2}$ and $\psi_{l-1} = \frac{\underline{\mathbf{z}}_{l-1}-\overline{\mathbf{z}}_{l-1}}{2}$.

**Lemma 1** (Lipschitz networks ensure bounded IBP). *Let $\overline{\mathbf{x}}$ and $\underline{\mathbf{x}}$ be $\varepsilon$-perturbations of $\mathbf{x} \sim \mathbb{P}_X$ for an $\varepsilon > 0$ (i.e. $\overline{\mathbf{x}}, \underline{\mathbf{x}} \in \mathbf{x}(\varepsilon)$). Given that the activation $\sigma$ is Lipschitz continuous (such as ReLU) with constant $c_\sigma > 0$, there exists a constant $D = D(c_\sigma; A_1, A_2, \cdots, A_S; \varepsilon)$ such that $\underline{f}_{\theta^S}(\mathbf{x}, \varepsilon)$ and $\overline{f}_{\theta^S}(\mathbf{x}, \varepsilon))$ will at most be an $\hat{\varepsilon}$-perturbed version of $f_{\theta^S}(\mathbf{x})$, where $\hat{\varepsilon} = \varepsilon D$.*

*Proof.* Given that $\mathbf{x}_1, \mathbf{x}_2 \in \mathcal{X}$

$$\begin{aligned}
\big\|h_1(\mathbf{x}_1) - h_1(\mathbf{x}_2)\big\| &= \big\|\sigma(A_1\mathbf{x}_1 + \mathbf{b}_1) - \sigma(A_1\mathbf{x}_2 + \mathbf{b}_1)\big\| \\
&\leq c_\sigma\big\|A_1(\mathbf{x}_1 - \mathbf{x}_2)\big\| \\
&\leq c_\sigma\|A_1\|\|\mathbf{x}_1 - \mathbf{x}_2\|
\end{aligned} \tag{13}$$

where $\|A_1\| = \sup_{\|\mathbf{x}\|=1}\|A_1\mathbf{x}\|$. The inequality 13 is due to the Lipschitz continuity of $\sigma$. Commonly used activation functions, such as ReLU, tend to satisfy this condition. In particular, for ReLU, $c_\sigma = 1$. As such, the map $h_1$ also turns out to be Lipschitz continuous. A similar argument also proves that $h_l$, $l = 2, \cdots, S$ all follow the same trait. As a result, $f_{\theta^S}$ also becomes Lipschitz continuous with accompanying constant $(c_\sigma A)^S$, where $A = \max\{\|A_l\|\}$.

The recurrence relation of extremities in IBP, as suggested by Gowal et al. (2019), can be written as:

$$\overline{f}_{\theta^l}(\mathbf{x}, \varepsilon) = \sigma\left\{\frac{(A_l + |A_l|)}{2}\overline{f}_{\theta^{l-1}}(\mathbf{x}, \varepsilon) + \frac{(A_l - |A_l|)}{2}\underline{f}_{\theta^{l-1}}(\mathbf{x}, \varepsilon) + \mathbf{b}_l\right\},$$

$$\text{and } \underline{f}_{\theta^l}(\mathbf{x}, \varepsilon) = \sigma\left\{\frac{(A_l - |A_l|)}{2}\overline{f}_{\theta^{l-1}}(\mathbf{x}, \varepsilon) + \frac{\{A_l + |A_l|\}}{2}\underline{f}_{\theta^{l-1}}(\mathbf{x}, \varepsilon) + \mathbf{b}_l\right\},$$

where the $|\cdot|$ operator results in a matrix with all elements replaced by their corresponding absolute values, and $l = 1, 2, ..., S$. Thus,

$$\begin{aligned}
&\left\|\overline{f}_{\theta^l}(\mathbf{x}, \varepsilon) - f_{\theta^l}(\mathbf{x})\right\| \\
=&\left\|\sigma\left\{\frac{(A_l + |A_l|)}{2}\overline{f}_{\theta^{l-1}}(\mathbf{x}, \varepsilon) + \frac{(A_l - |A_l|)}{2}\underline{f}_{\theta^{l-1}}(\mathbf{x}, \varepsilon) + \mathbf{b}_l\right\} - \sigma\left(A_l f_{\theta^{l-1}}(\mathbf{x}) + \mathbf{b}_l\right)\right\| \\
\leq& c_\sigma\left\|\frac{(A_l + |A_l|)}{2}\overline{f}_{\theta^{l-1}}(\mathbf{x}, \varepsilon) + \frac{(A_l - |A_l|)}{2}\underline{f}_{\theta^{l-1}}(\mathbf{x}, \varepsilon) - A_l f_{\theta^{l-1}}(\mathbf{x})\right\| \\
=& c_\sigma\left\|\frac{(A_l + |A_l|)}{2}\left(\overline{f}_{\theta^{l-1}}(\mathbf{x}, \varepsilon) - f_{\theta^{l-1}}(\mathbf{x})\right) + \frac{(A_l - |A_l|)}{2}\left(\underline{f}_{\theta^{l-1}}(\mathbf{x}, \varepsilon) - f_{\theta^{l-1}}(\mathbf{x})\right)\right\| \\
\leq& c_\sigma\left\{\left\|\frac{(A_l + |A_l|)}{2}\left(\overline{f}_{1(\theta^{l-1}}(\mathbf{x}, \varepsilon) - f_{\theta^{l-1}}(\mathbf{x})\right)\right\| + \left\|\frac{(|A_l| - A_l)}{2}\left(f_{\theta^{l-1}}(\mathbf{x}) - \underline{f}_{\theta^{l-1}}(\mathbf{x}, \varepsilon)\right)\right\|\right\} \\
\leq& c_\sigma\left\{\left\|\frac{A_l + |A_l|}{2}\right\|\left\|\overline{f}_{\theta^{l-1}}(\mathbf{x}, \varepsilon) - f_{\theta^{l-1}}(\mathbf{x})\right\| + \left\|\frac{|A_l| - A_l}{2}\right\|\left\|f_{\theta^{l-1}}(\mathbf{x}) - \underline{f}_{\theta^{l-1}}(\mathbf{x}, \varepsilon)\right\|\right\}.
\end{aligned}$$

Observe that, in particular for $l = 1$

$$\left\|\overline{f}_{\theta^1}(\mathbf{x}, \varepsilon) - f_{\theta^1}(\mathbf{x})\right\| \leq c_\sigma \left\{ \left\|\frac{A_1 + |A_1|}{2}\right\| \|\overline{\mathbf{x}} - \mathbf{x}\| + \left\|\frac{|A_1| - A_1}{2}\right\| \|\mathbf{x} - \underline{\mathbf{x}}\| \right\}$$

$$= c_\sigma \varepsilon \left\{ \left\|\frac{A_1 + |A_1|}{2}\right\| + \left\|\frac{|A_1| - A_1}{2}\right\| \right\} = \varepsilon_1 \text{ say,}$$

i.e., the deviation in the first layer can be made arbitrarily small based on $\varepsilon$. The quantity $\left\|f_{\theta^1}(\mathbf{x}) - \underline{f}_{\theta^1}(\mathbf{x}, \varepsilon)\right\|$ can be shown to be upper bounded using a similar argument. In other words, both $\overline{f}_{\theta^1}(\mathbf{x}, \varepsilon)$ and $\underline{f}_{\theta^1}(\mathbf{x}, \varepsilon)$ are at most $\varepsilon_1$-perturbed from $f_{\theta^1}(\mathbf{x})$. By the method of induction we eventually get a $D = D(c_\sigma; A_1, A_2, \cdots, A_S; \varepsilon) > 0$ for which the lemma holds. $\square$

**Theorem 1** (Generalization bound). *Let $\tilde{\mathbb{P}}$ be the joint distribution of $(f_{\theta^S}(X), Y)$, supported on $\mathcal{H} \times \mathbb{R}$. In the multi-task regime, let $I$ denote the set of tasks, each consisting of $N$ samples. Define $\hat{\mathcal{R}}(N, |I|) = \mathbb{E}_{\mathcal{T}_i \sim \hat{p}(\mathcal{T})} \mathbb{E}_{(X_j, Y_j) \sim \hat{p}(\mathcal{T}_i)}[\mathcal{L}_{CE}(f_{\theta^{L-S}}(\mathbf{H}_j^*), Y_j)]$ and $\mathcal{R} = \mathbb{E}_{\mathcal{T}_i \sim p(\mathcal{T})} \mathbb{E}_{(X_j, Y_j) \sim \mathcal{T}_i}[\mathcal{L}_{CE}(f_{\theta^{L-S}}(f_{\theta^S}(X_j)), Y_j)]$. For a bounded loss function $\mathcal{L}_{CE} : \mathbb{R} \times \mathbb{R} \to [0, a](a \geq 0)$, if the neural network-induced map $f_{\theta^{L-S}}$ is such that $\left|\nabla f_{\theta^{L-S}}(\cdot)\right| < \infty$, we ensure:*

$$\left|\hat{\mathcal{R}}(N, |I|) - \mathcal{R}\right| - \tilde{\lambda} \precsim 2^{L-S+1} \sqrt{2\log(2\kappa + 2)} \left\{ \frac{1}{\sqrt{N}} + \frac{1}{\sqrt{|I|}} \right\} + \sqrt{\frac{\log(\frac{2|I|}{\delta})}{N}} + \sqrt{\frac{\log(\frac{2}{\delta})}{|I|}}$$

*holds with probability at least $1 - \delta$, where $\tilde{\lambda} = \tilde{\lambda}(\hat{\varepsilon}, \lambda)$.*

*Proof.* Before beginning with the proof we point out that, based on Definition 1, given $\varepsilon > 0$ and $\mathbf{x} \in \mathcal{X}$, any $\mathbf{x}' \in \mathbf{x}(\varepsilon)$ can be written as $\mathbf{x}' = \mathbf{x} + \eta(\varepsilon)$. For example, in the simplest case, $\eta(\varepsilon)$ can be a function in the family $\pm\epsilon\mathbf{1}$. Thus, in case of IBI, the $\underline{f}_{\theta^S}(\mathbf{x}_i, \varepsilon)$ and $\overline{f}_{\theta^S}(\mathbf{x}_i, \varepsilon)$ can both be expressed as $f_{\theta^S}(\mathbf{x}_i) + \eta(\hat{\varepsilon})$ with corresponding $\eta(\hat{\varepsilon})$. In essence $\mathbf{H}_i^* = (1 - \lambda)f_{\theta^S}(\mathbf{x}_i) + \lambda(f_{\theta^S}(\mathbf{x}_i) + \eta(\hat{\varepsilon}))$, where $\lambda \in [0, 1]$. Now, we can observe that,

$$f_{\theta^{L-S}}(\mathbf{H}_i^*) = f_{\theta^{L-S}}\left((1-\lambda)f_{\theta^S}(\mathbf{x}_i) + \lambda\left[f_{\theta^S}(\mathbf{x}_i) + \eta(\hat{\varepsilon})\right]\right)$$

$$= f_{\theta^{L-S}}\left(f_{\theta^S}(\mathbf{x}_i) + \lambda\eta(\hat{\varepsilon})\right)$$

$$= f_{\theta^{L-S}}\left(f_{\theta^S}(\mathbf{x}_i)\right) + \lambda\nabla f_{\theta^{L-S}}\left(f_{\theta^S}(\mathbf{x_i})\right)\eta(\hat{\varepsilon}), \tag{14}$$

where $\eta(\hat{\varepsilon}) \in \mathbb{R}^\kappa$, $\hat{\varepsilon}$ being as mentioned in lemma 1. We obtain (14) by using the Taylor expansion of $f_{\theta^{L-S}}$ up to the first order. Given that $\left|\nabla f_{\theta^{L-S}}(\cdot)\right| < \infty$, the second term $\lambda\nabla f_{\theta^{L-S}}\left(f_{\theta^S}(\mathbf{x}_i)\right)\eta(\hat{\varepsilon})$ can be made arbitrarily small. The higher-order terms in the expansion all follow suit, which justifies their omission. Now,

$$\left|\frac{1}{N}\sum_{i=1}^N \mathcal{L}_{CE}(f_{\theta^{L-S}}(\mathbf{H}_i^*), y_i) - \int_{\mathcal{H} \times \mathbb{R}} \mathcal{L}_{CE}(f_{\theta^{L-S}}(\mathbf{x}), y)d\tilde{\mathbb{P}}(\mathbf{x}, y)\right|$$

$$= \left|\frac{1}{N}\sum_{i=1}^N \left[\mathcal{L}_{CE}(f_{\theta^{L-S}}(\mathbf{H}_i^*), y_i) - \mathcal{L}_{CE}(f_{\theta^{L-S}}(f_{\theta^S}(\mathbf{x}_i)), y_i)\right] \right.$$

$$\left. + \frac{1}{N}\sum_{i=1}^N \mathcal{L}_{CE}(f_{\theta^{L-S}}(f_{\theta^S}(\mathbf{x}_i)), y_i) - \int_{\mathcal{H} \times \mathbb{R}} \mathcal{L}_{CE}(f_{\theta^{L-S}}(\mathbf{x}), y)d\tilde{\mathbb{P}}(\mathbf{x}, y)\right|$$

$$\leq \frac{1}{N}\sum_{i=1}^N \left|\mathcal{L}_{CE}(f_{\theta^{L-S}}(\mathbf{H}_i^*), y_i) - \mathcal{L}_{CE}(f_{\theta^{L-S}}(f_{\theta^S}(\mathbf{x}_i)), y_i)\right|$$

$$+ \left|\frac{1}{N}\sum_{i=1}^N \mathcal{L}_{CE}(f_{\theta^{L-S}}(f_{\theta^S}(\mathbf{x}_i)), y_i) - \int_{\mathcal{H} \times \mathbb{R}} \mathcal{L}_{CE}(f_{\theta^{L-S}}(\mathbf{x}), y)d\tilde{\mathbb{P}}(\mathbf{x}, y)\right|. \tag{15}$$

Since our networks use ReLU activation, the map induced by $f_{\theta^{L-S}}$ can be shown to be continuous. Given $\mathcal{H}$ is compact the output space also becomes compact. Restricted to such a space, the cross-entropy loss $\mathcal{L}_{CE}$ (similarly, regularized cross-entropy loss) turns out to be Lipschitz continuous. Consequently,

$$|\mathcal{L}_{CE}(f_{\theta^{L-S}}(\mathbf{H}_i^*), y_i) - \mathcal{L}_{CE}(f_{\theta^{L-S}}(f_{\theta^S}(\mathbf{x}_i)), y_i)|$$
$$\leq c_L \left\| f_{\theta^{L-S}}(\mathbf{H}_i^*) - f_{\theta^{L-S}}\left(f_{\theta^S}(\mathbf{x}_i)\right) \right\| = \tilde{\lambda}(\hat{\varepsilon}, \lambda), \qquad (16)$$

where $c_L > 0$ is the Lipschitz constant associated with $\mathcal{L}_{CE}$. Without loss of generality we can construct the map $f_{\theta^{L-S}}$ such that $\|f_{\theta^{L-S}}\| \leq 1$. Now, in case there are $|I|$ tasks involved, namely $\{\mathcal{T}_i\}_{i=1}^{|I|}$ (i.e., the multi-task regime), the population risk turns out to be

$$\mathcal{R} = \mathbb{E}_{\mathcal{T}_i \sim p(\mathcal{T})} \mathbb{E}_{(X_j, Y_j) \sim \mathcal{T}_i} \left[ \mathcal{L}_{CE}\left(f_{\theta^{L-S}}(f_{\theta^S}(X_j)), Y_j\right) \right]$$
$$= \mathbb{E}_{\mathcal{T}_i \sim p(\mathcal{T})} \mathbb{E}_{(X_j, Y_j) \sim \mathcal{T}_i} \left[ \mathcal{L}_{CE}\left(f_{\theta^{L-S}}(\mathbf{H}_j), Y_j\right) \right].$$

We are interested in observing the deviation of the same from the realized risk. In other words,

$$\left| \hat{\mathcal{R}}(N, |I|) - \mathcal{R} \right| \leq \underbrace{\left| \hat{\mathcal{R}}(N, |I|) - \mathcal{J} \right|}_{(i)} + \underbrace{\left| \mathcal{J} - \mathcal{R} \right|}_{(ii)}, \qquad (17)$$

where $\mathcal{J} = \mathbb{E}_{\mathcal{T}_i \sim \hat{p}(\mathcal{T})} \mathbb{E}_{(X_j, Y_j) \sim \mathcal{T}_i} \left[ \mathcal{L}_{CE}\left(f_{\theta^{L-S}}(\mathbf{H}_j), Y_j\right) \right]$ and $\hat{p}$ is the empirical counterpart of the task distribution. Using Jensen's inequality, $(i)$ can be upper bounded by

$$\mathbb{E}_{\mathcal{T}_i \sim \hat{p}(\mathcal{T})} \left| \mathbb{E}_{(X_j, Y_j) \sim \hat{p}(\mathcal{T}_i)} \left[ \mathcal{L}_{CE}\left(f_{\theta^{L-S}}(\mathbf{H}_j^*), Y_j\right) \right] - \mathbb{E}_{(X_j, Y_j) \sim \mathcal{T}_i} \left[ \mathcal{L}_{CE}\left(f_{\theta^{L-S}}(\mathbf{H}_j), Y_j\right) \right] \right|$$

$$\leq \tilde{\lambda} + \mathbb{E}_{\mathcal{T}_i \sim \hat{p}(\mathcal{T})} \left| \mathbb{E}_{(X_j, Y_j) \sim \hat{p}(\mathcal{T}_i)} \left[ \mathcal{L}_{CE}\left(f_{\theta^{L-S}}(\mathbf{H}_j), Y_j\right) \right] - \mathbb{E}_{(X_j, Y_j) \sim \mathcal{T}_i} \left[ \mathcal{L}_{CE}\left(f_{\theta^{L-S}}(\mathbf{H}_j), Y_j\right) \right] \right|, \qquad (18)$$

where we utilize arguments (15) and (16) to reach (18). Using the union bound based on $|I|$ tasks on top of Corollary 3.14 of Wojtowytsch & E (2020) we can show that the second term in the right-hand side of (18) becomes $\precsim 2^{L-S+1}\sqrt{\frac{2\log(2\kappa+2)}{N}} + a\sqrt{\frac{2\log(\frac{2|I|}{\delta})}{N}}$, with probability at least $1 - \delta$.

To put a deterministic upper bound on $(ii)$ let us first define the class of functions

$$\mathcal{G} = \left\{ g : g(\mathcal{T}) = \mathbb{E}_{(f_{\theta^S}(X), Y) \sim \tilde{\mathbb{P}}} \left[ \mathcal{L}_{CE}\left(f_{\theta^{L-S}}(\mathbf{H}), Y\right) \right]; f_{\theta^{L-S}} \in W^{L-S} \right\},$$

where $W^{L-S}$ is the function space induced by networks with $L - S$ hidden layers (Wojtowytsch & E, 2020). Let us now calculate the Rademacher complexity of the class functions $\mathcal{G}$:

$$Rad\left(\mathcal{G}, \{\mathcal{T}_i\}_{i=1}^{|I|}\right) = \mathbb{E}_\xi \sup_{g \in \mathcal{G}} \frac{1}{|I|} \left| \sum_{i=1}^{|I|} \xi_i g(\mathcal{T}_i) \right| = \mathbb{E}_\xi \sup_{g \in \mathcal{G}} \frac{1}{|I|} \left| \sum_{i=1}^{|I|} \xi_i \mathbb{E}_{\mathcal{T}_i} \left[ \mathcal{L}_{CE}\left(f_{\theta^{L-S}}(\mathbf{H}), Y\right) \right] \right|$$

$$\leq \mathbb{E}_{\mathcal{T}_i} \mathbb{E}_\xi \sup_{g \in \mathcal{G}} \frac{1}{|I|} \left| \sum_{i=1}^{|I|} \xi_i \mathcal{L}_{CE}\left(f_{\theta^{L-S}}(\mathbf{H}), Y\right) \right| \qquad (19)$$

$$\leq c_L \mathbb{E}_{\mathcal{T}_i} \mathbb{E}_\xi \sup_{f_{\theta^{L-S}} \in W^{L-S}} \frac{1}{|I|} \left| \sum_{i=1}^{|I|} \xi_i f_{\theta^{L-S}}(\mathbf{H}) \right| \qquad (20)$$

$$\leq c_L 2^{L-S+1} \sqrt{\frac{2\log(2\kappa + 2)}{|I|}}, \qquad (21)$$

where (20) is due to the Lipschitz property of $\mathcal{L}_{CE}(\cdot, y)$ [Lemma 26.9 of Shalev-Shwartz & Ben-David (2014) or Theorem 7 of Meir & Zhang (2003)]. We arrive at (21) using lemma 3.13 of Wojtowytsch

& E (2020). The inequality (19) is based on the fact that $\sup_{u \in \mathcal{U}} \left| \mathbb{E}[u(X)] \right| \leq \mathbb{E}[\sup_{u \in \mathcal{U}} |u(X)|]$, given the expectation exists for the class of functions $\mathcal{U}$ and random variable $X$.

Thus we obtain the deterministic bound on $(ii)$ given by

$$\left| \mathbb{E}_{\mathcal{T}_i \sim \hat{p}(\mathcal{T})} \mathbb{E}_{(X_j, Y_j) \sim \mathcal{T}_i} \left[ \mathcal{L}_{CE} \left( f_{\theta^{L-S}}(\mathbf{H}_j), Y_j \right) \right] - \mathbb{E}_{\mathcal{T}_i \sim p(\mathcal{T})} \mathbb{E}_{(X_j, Y_j) \sim \mathcal{T}_i} \left[ \mathcal{L}_{CE} \left( f_{\theta^{L-S}}(\mathbf{H}_j), Y_j \right) \right] \right|$$

$$\precsim \sqrt{\frac{2 \log(2\kappa + 2)}{|I|}} + \sqrt{\frac{\log(\frac{2}{\delta})}{|I|}},$$

that holds with probability at least $1 - \delta$. The bounds on (i) and (ii) together prove the theorem. $\square$

## D DETAILS OF DATASETS USED IN THIS STUDY

**miniImageNet:** The miniImageNet dataset (Vinyals et al., 2016) is a commonly used subset of ImageNet (Deng et al., 2009) for evaluating few-shot classifiers. The dataset contains a total of 100 classes each containing 600 images of resolution $84 \times 84 \times 3$. Following the directives of Vinyals et al. (2016) from the total 100 classes 64 are kept in the Training set, 16 are retained for Validation, and the rest of the 20 classes are used for testing.

**tieredImageNet:** In (Ren et al., 2018) the authors proposed a new larger subset of ImageNet (Deng et al., 2009) for addressing the limitations of miniImageNet. In miniImageNet it is not ensured that the classes used for training are distinct from those contained in the Test set. Evidently, this contains the risk of information leakage and may not provide a fair evaluation of the few-shot classifier. As a remedy Ren et al. (2018) proposed to go higher in the class hierarchy in ImageNet. This enables tieredImageNet to use higher-level categories in the Training, Validation, and Test sets maintaining significant diversity between the three. In essence, a total of 608 ImageNet leaf-level classes are considered that can be categorized into 34 groups. Among these 34 higher-level groups, 20 are used for training, 6 are kept for validation, and the rest 8 are included in the Test set.

**miniImageNet-S:** This dataset is created by only using a subset of the original miniImageNet Training set for training the few-shot learner in a few-task scenario Yao et al. (2022).

```
Training Classes:  n03017168, n07697537, n02108915, n02113712,
n02120079, n04509417, n02089867, n03888605, n04258138, n03347037,
n02606052, n06794110
```

Validation and Test sets are kept as same as those used in miniImageNet.

**DermNet-S:** DermNet-S (Yao et al., 2022) is a subset of the "Dermnet Skin Disease Atlas" publicly available at `http://www.dermnet.com/`. The dataset after discarding the duplicates contains more than 22,000 medical images spread across 625 classes of dermatological diseases. Following the preprocessing suggested by Prabhu et al. (2019) the authors of (Yao et al., 2022) created DermNet-S by first extracting the 203 classes containing more than 30 images. Then from the long-tailed data distribution of the 203 disease classes, the top 30 larger classes are kept for training while the smaller 53 bottom classes are considered for meta-testing. The images are resized to $84 \times 84 \times 3$ to match the resolution of miniImageNet. We follow the same dataset construction strategy in our case. Moreover, we use random classes not included in the Training or Test set as the Validation set. The complete list of classes in the Training and Test sets are listed as follows:

```
Training Classes:  Seborrheic Keratoses Ruff, Herpes Zoster,
Atopic Dermatitis Adult Phase, Psoriasis Chronic Plaque, Eczema
Hand, Seborrheic Dermatitis, Keratoacanthoma, Lichen Planus,
Epidermal Cyst, Eczema Nummular, Tinea (Ringworm) Versicolor,
Tinea (Ringworm) Body, Lichen Simplex Chronicus, Scabies,
Psoriasis Palms Soles, Malignant Melanoma, Candidiasis large
Skin Folds, Pityriasis Rosea, Granuloma Annulare, Erythema
Multiforme, Seborrheic Keratosis Irritated, Stasis Dermatitis
and Ulcers, Distal Subungual Onychomycosis, Allergic Contact
Dermatitis, Psoriasis, Molluscum Contagiosum, Acne Cystic,
Perioral Dermatitis, Vasculitis, Eczema Fingertip
```

```
Testing Classes:  Warts, Ichthyosis Sex Linked, Atypical Nevi,
Venous Lake, Erythema Nodosum, Granulation Tissue, Basal Cell
Carcinoma Face, Acne Closed Comedo, Scleroderma, Crest Syndrome,
Ichthyosis Other Forms, Psoriasis Inversus, Kaposi Sarcoma,
Trauma, Polymorphous Light Eruption, Dermagraphism, Lichen
Sclerosis Vulva, Pseudomonas, Cutaneous Larva Migrans, Psoriasis
Nails, Corns, Lichen Sclerosus Penis, Staphylococcal Folliculitis,
Chilblains Perniosis, Psoriasis Erythrodermic, Squamous Cell
Carcinoma Ear, Basal Cell Carcinoma Ear, Ichthyosis Dominant,
Erythema Infectiosum, Actinic Keratosis Hand, Basal Cell Carcinoma
Lid, Amyloidosis, Spiders, Erosio Interdigitalis Blastomycetica,
Scarlet Fever, Pompholyx, Melasma, Eczema Trunk Generalized,
Metastasis, Warts Cryotherapy, Nevus Spilus, Basal Cell Carcinoma
Lip, Enterovirus, Pseudomonas Cellulitis, Benign Familial Chronic
Pemphigus, Pressure Urticaria, Halo Nevus, Pityriasis Alba,
Pemphigus Foliaceous, Cherry Angioma, Chapped Fissured Feet,
Herpes Buttocks, Ridging Beading
```

**ISIC:** Following Yao et al. (2022) for "ISIC 2018: Skin Lesion Analysis Towards Melanoma Detection" (Codella et al., 2018; Li et al., 2020) we select the third task where 10,015 medical images are categorized into seven classes based on lesion types. We first resize the images to $84 \times 84 \times 3$ to match the miniImageNet resolution. Then among the seven classes in the ISIC dataset, we select the 4 classes containing a higher number of samples for training while considering the rest for meta-testing as per the directives of Yao et al. (2022). Since there are only 4 classes in the Training set setting the number of ways to 2 results in six possible class combinations in a task. This in consequence offers an extreme few-task scenario. For hyper-parameter tuning random classes are used as a Validation set following the cross-validation-based approach employed in (Yao et al., 2022). The list of classes in the Training and the Test sets are listed as follows:

```
Training Set:  Nevus, Melanoma, Benign Keratoses, Basal Cell
Carcinoma
```

```
Testing Set:  Dermatofibroma, Pigmented Bowen's, Vascular
```

## E  IMPLEMENTATION DETAILS

**Scheduling of $\epsilon$:** In their paper Gowal et al. (2019) recommended starting with an initial perturbation $\epsilon_0 = 0$ and gradually increasing it to the intended perturbation $\epsilon$ over the training steps. In our case, we follow a similar approach for scheduling the value of perturbation $\epsilon_t$ at the $t$-th training step. We have observed that a fast increase in perturbation usually slows down training while a very slow increment fails to aid the learner. From extensive experimentation with various scheduling techniques such as linear, cosine, etc., we have found that the following strategy works well in practice. If the maximum allowed number of training steps is set to $T$ then for the step $t$ the perturbation $\epsilon_t$ is calculated as:

$$\epsilon_t = \begin{cases} \epsilon \text{ if } t > \lceil 0.9T \rceil \\ \frac{t}{0.9T}\epsilon \end{cases} . \tag{22}$$

In essence, we linearly increase $\epsilon_t$ starting from 0 up to $\epsilon$ over 90% of the maximum training steps $T$ and keep it fixed at $\epsilon$ for the remainder of the training.

**Frequency of interpolation for IBI variants:** Performing IBP bound–based interpolation for every task during training may not be beneficial and may instead mislead the learner. For MAML, we have seen that performing interpolation once in every batch of $B$ tasks aids the training process. In the case of ProtoNet, we have found that performing IBP bound–based interpolation with a 25% probability results in the best outcome.

**Modifications to network architecture:** We have used two networks for our experiments namely "4-CONV" and "ResNet-12". The "4-CONV" network can be seamlessly integrated with IBI for both MAML and ProtoNet. This network consists of 4 blocks, each having a convolution, batch normalization, max pooling, and ReLU in sequential order. IBI can be performed after any one of the blocks. The ResNet-12 network also consists of 4 blocks where a block (except the first one)

receives inputs from (1) the output of the preceding block, and (2) the input of the preceding block through a skip connection. While the idea of applying IBI after any of the blocks seems appealing, the presence of skip connections may hinder a straightforward integration of IBI in this case. To understand how ResNet-12 can be customized to accommodate MAML+IBP (and consequently MAML+IBI) we undertake an ablation study on the miniImageNet-S dataset in a 5-way 1-shot classification problem as described in Table 6. We can observe that in our initial hyperparameter tuning experiment, MAML+IBP can not match the performance of vanilla MAML on ResNet-12. Moreover, the performance gap increases as IBP is applied deeper into the network. This may be explained by the fact that the interval bounds become gradually loose as they progress through the network. Thus, with increasing depth, the magnitude of the bound losses (especially $\mathcal{L}_{UB}$ as the ReLU activations prevent $\mathcal{L}_{LB}$ from becoming too large) will largely outscale the classification loss and consequently affect convergence (see Remark 2). Applying IBP after only the first block still fails to achieve parity with the baseline because IBP induces a distortion in the feature space, due to its regularization effect. While the sequential part of the blocks after IBP can adapt to this distortion due to their complexity, the simpler skip paths can not do so. Hence, the effect of the distortion keeps propagating to the deeper blocks via skip connections. To aid the network in such a situation, we investigate three approaches to modify the skip connection immediately after the block(s) subjected to IBP, viz. (1) remove the skip connection for the subsequent block (2) introduce additional layers in the skip connection for the subsequent block to make it deeper and more complex (3) use a skip after one or more of the initial sub-block(s) (consisting sequentially of one convolution, one batch normalization, and one ReLU layer) of the next block. Among the three approaches, we empirically found that MAML+IBP (consequently MAML+IBI) performs best when the skip connection starts after the second sub-block in block 2. Due to the comparatively powerful learning strategy of ProtoNet, no such modifications to ResNet-12 are necessary for ProtoNet+IBI.

Table 6: Ablation study of ResNet-12 modifications for MAML+IBP on miniImageNet-S in terms of mean Accuracy over 600 tasks with 95% confidence interval.

| Algorithm | IBP position | Accuracy |
|---|---|---|
| MAML | None (Baseline) | 40.02±0.78% |
| MAML+IBP | after block 4 | 21.24±0.54% |
| MAML+IBP | after block 3 | 23.77±0.59% |
| MAML+IBP | after block 2 | 29.62±0.62% |
| MAML+IBP | after block 1 | 37.95±0.83% |
| MAML+IBP | after block 1 with no-skip at block 2 | 37.81±0.85% |
| MAML+IBP | after block 1 with deeper skip at block 2 | 38.54±0.81% |
| MAML+IBP | after block 1 with skip and output combination at block 2 | 40.50±0.83% |
| MAML+IBP | after block 1 with skip after 1 sub-block in block 2 | 42.18±0.82% |
| MAML+IBP | after block 1 with skip after 2 sub-blocks in block 2 | **43.50±0.86%** |

**Remark 2.** [Scalability of IBI] IBP (and consequently IBI) requires the propagation of the two interval bounds along with the input data. This introduces a computational overhead, especially in deeper networks. However, in practice even in a deeper network, we may not need to perform IBP except in the initial few layers, as the bound losses will otherwise overwhelm the classification loss and consequently impact convergence. To demonstrate this, we plot the losses (up to 5000 training steps for the ease of visualization) in the following Figure 5 for MAML+IBI using a ResNet-12 network for 5-way 1-shot miniImageNet-S classification, when IBP is applied up to blocks 1-4. We can see that the three losses have comparable scales only when IBP is applied after block 1. In all other cases, $\mathcal{L}_{UB}$ heavily dominates the total loss. But, due to its sheer magnitude, the optimizer is unable to minimize it. Thus, in practice, IBP should only be limited to a few initial layers in deeper networks. Consequently, IBI easily scales to deeper networks despite the computational overhead.

**Remark 3.** To show that IBP and IBI variants are well-scalable as their vanilla counterpart we list the actual training costs in the following Table 7 in terms of the average time in seconds to execute a single training step of the algorithm. All the experiments are performed in the same environment using a RTX 3090 GPU. From Table 7 we can observe that, in the case of MAML, the IBI and IBP variants only takes about 40%-70% additional time when "4-CONV" is used. The difference in cost reduces further if ResNet-12 is used as the backbone. This is expected as we only need to apply

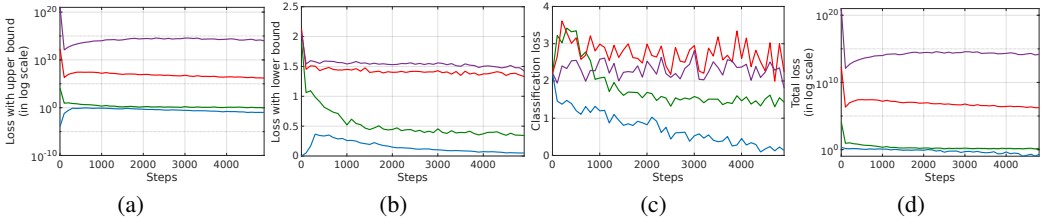

Figure 5: In the four plots above of losses against training steps the Blue, Green, Red, and Magenta lines respectively denote IBI applied after blocks 1,2,3, and 4 in ResNet-12 without any additional modifications. (a) Plot of $\mathcal{L}_{UB}$ in log scale for ease of visualization. (b) Plot of $\mathcal{L}_{LB}$. (c) Plot of $\mathcal{L}_{CE}$. (d) Plot of $\mathcal{L}$ in log scale for ease of visualization.

IBP in the first few layers of ResNet-12 to gain its full advantage. For ProtoNet, the increment in computational cost for the proposed techniques is observed to be slightly more compared to that of MAML.

Table 7: Actual computational cost in seconds for IBP and IBI variants of MAML and ProtoNet with "4-CONV" and ResNet-12 backbone.

| Algorithm | 4-CONV | | ResNet-12 | |
|---|---|---|---|---|
| | 1-shot | 5-shot | 1-shot | 5-shot |
| MAML | 0.244 | 0.432 | 1.408 | 3.742 |
| MAML + IBP (ours) | 0.407 | 0.615 | 1.994 | 4.324 |
| MAML + IBI (ours) | 0.412 | 0.616 | 2.001 | 4.326 |
| ProtoNet | 0.067 | 0.073 | 0.075 | 0.091 |
| ProtoNet + IBP (ours) | 0.129 | 0.144 | 0.196 | 0.221 |
| ProtoNet + IBI (ours) | 0.133 | 0.156 | 0.202 | 0.233 |

## F  HYPERPARAMETERS USED IN IBP AND IBI

### F.1  NAMES AND FUNCTIONS OF HYPERPARAMETERS

The following Table 8 describes the hyperparameters used in the vanilla MAML, MAML+IBP, and MAML+IBI.

The following Table 9 describes the hyperparameters used in the vanilla ProtoNet, ProtoNet+IBP, and ProtoNet+IBI.

### F.2  HYPERPARAMETER SEARCH SPACE AND TUNING

For hyperparameter tuning, we employ a grid search. In Table 10, we list the search spaces for each of the hyperparameters used in MAML+IBP and MAML+IBI. Moreover, in Table 11, we also detail the search spaces for each of the hyperparameters used in ProtoNet+IBP and ProtoNet+IBI. For all other learners used in Tables 1 and 2 in the main paper, the results are either taken from the corresponding article or reproduced using the originally recommended hyperparameter settings.

In Tables 12 and 13, we report the optimal dataset-specific hyperparameters for MAML+IBP and MAML+IBI. Similarly, Tables 14 and 15 detail the optimal dataset-specific hyperparameter choices for ProtoNet+IBP and ProtoNet+IBI.

For Table 3 in the main paper, the methods using static weights share the same hyperparameter settings with their dynamic weighted counterpart except for $\gamma$, which is not used for the static weight runs. For Table 4 in the main paper, all the MAML variants use the same settings as vanilla MAML. Further, for all the different interpolation strategies $Beta$ distribution is used with the choices of $\alpha$ and $\beta$ matching those of the MAML+IBI settings.

Table 8: Descriptions of hyperparameters used in vanilla MAML, MAML+IBP, MAML+IBI

| Hyperarameter name | Hyperparameter description |
|---|---|
| **Hyperparameters used in MAML** | |
| Meta-shots | Number of shots in the query set in the training phase. |
| Inner loop iterations | Number of iterations of the inner loop during training on support set. |
| Inner loop learning rate $\eta_0$ | Learning rate for SGD in the inner loop during training on support set. |
| Meta-step size $\eta_1$ | Learning rate for ADAM in the meta-learner update during training. |
| Meta-batch $B$ | Batch size of task during training. |
| Meta-iterations $T$ | Number of training steps. |
| Evaluation iterations | Number of fine-tuning steps on support set during meta-testing. |
| **Additional hyperparameters introduced in MAML+IBP** | |
| Interval coefficient $\epsilon$ | Perturbation required for IBP. |
| Softmax coefficient $\gamma$ | Controls the relative importance of the three losses used in MAML+IBP during softmax-based weighting in training phase. |
| Layer $S$ | A layer in the network where IBP losses will be calculated. |
| **Additional hyperparameters introduced in MAML+IBI** | |
| $\alpha$ and $\beta$ | Hyperparameters associated with the $Beta$ distribution required for performing IBP bounds-based interpolation. |

Table 9: Descriptions of hyperparameters used in vanilla ProtoNet, ProtoNet+IBP, ProtoNet+IBI

| Hyperarameter name | Hyperparameter description |
|---|---|
| **Hyperparameters used in MAML** | |
| Number of ways in training | Traditional ProtoNet Snell et al. (2017) usually considers a higher number of ways during training. |
| Meta-shots | Number of shots in the query set in training phase. |
| Meta-step size $\eta$ | Learning rate for ADAM in the learner update during training. |
| Meta-iterations $T$ | Number of training steps. |
| Distance metric | Choice of distance measure, Euclidean or Cosine. |
| **Additional hyperparameters introduced in ProtoNet+IBP** | |
| Interval coefficient $\epsilon$ | Perturbation required for IBP. |
| Softmax coefficient $\gamma$ | Controls the relative importance of the three losses used in ProtoNet+IBP during softmax-based weighting in training phase. |
| Layer $S$ | A layer in the network where IBP losses will be calculated. |
| **Additional hyperparameters introduced in ProtoNet+IBI** | |
| $\alpha$ and $\beta$ | Hyperparameters associated with the $Beta$ distribution required for performing IBP bounds-based interpolation. |

## F.3 FULL RESULTS

**Contenders in Motivating Example:** For the contenders in Table 1 the settings are as follows:

1. MAML+SN on $f_{\theta^S}$: This variant of MAML applies Spectral Normalization (Miyato et al., 2018) up to the $S$-th layer of the "4-CONV" network. Here similar to the MAML+IBP the value of $S$ is set to 3.

2. MAML+SN on $f_\theta$: Here Spectral Normalization is applied on the full network.

3. MAML+GL: In this variant, we calculate a Gaussian regularization loss instead of IBP. Here we send the query set along with its perturbed version and attempt to minimize their norm after the $S$-th layer alongside $\mathcal{L}_{CE}$. The extra loss $\mathcal{L}_{GL}$ can be expressed as follows:

$$\mathcal{L}_{GL} = \frac{1}{N_q} \sum_{r=1}^{N_q} ||f_{\theta^S}(\mathbf{x}_{i,r}^q) - f_{\theta^S}(\mathbf{x}_{i,r}^q + \zeta)||_2^2,$$

Table 10: Grid search space of hyperparameters used in vanilla MAML, MAML+IBP, MAML+IBI

| Hyperaramter name | Hyperparameter search space |
|---|---|
| Hyperparameters used in MAML | |
| Meta-shots | Set to 15 following Finn et al. (2017). |
| Inner loop iterations | Set to 5 following Finn et al. (2017). |
| Inner loop learning rate $\eta_0$ | Set to 0.01 following Finn et al. (2017). |
| Meta-step size $\eta_1$ | Set to 0.001 following Finn et al. (2017). |
| Meta-batch $B$ | Set to 4 following Finn et al. (2017). |
| Meta-iterations $T$ | Set to 60000 for miniImageNet and tieredImageNet following Finn et al. (2017). Set to 50000 for miniImageNet-S, DermNet-S, and ISIC following Yao et al. (2022). |
| Evaluation iterations | Set to 10 following Finn et al. (2017). |
| Additional hyperparameters introduced in MAML+IBP | |
| Interval coefficient $\epsilon$ | Searched in the set $\{0.05, 0.1, 0.2\}$. |
| Softmax coefficient $\gamma$ | Searched in the set $\{0.01, 1, 10\}$. |
| Layer $S$ | For the "4-CONV" learner containing 4 blocks of Convolution, Batch normalization, Max pooling, and ReLU, S is searched at the block level in the set $\{1, 2, 3, 4\}$. For example, $S = 2$ means IBP losses are calculated after the second block. For the "ResNet-12" network the ablation study in Appendix E provides the optimum choice of $S$. |
| Additional hyperparameters introduced in MAML+IBI | |
| $\alpha$ and $\beta$ | Search space contains three pairs of choices $(0.1, 1)$, $(0.25, 1)$, and $(0.5, 0.5)$ where a tuple contains the value of $\alpha$ and $\beta$ in order. |

Table 11: Grid search space of hyperparameters used in vanilla ProtoNet, ProtoNet+IBP, ProtoNet+IBI

| Hyperaramter name | Hyperparameter search space |
|---|---|
| Hyperparameters used in ProtoNet | |
| Number of ways in training | Set to 30 for miniImageNet and tieredImageNet following Snell et al. (2017). Set to 5 for miniImageNet-S and DermNet-S, and 2 for ISIC as the benefit of training using higher ways cannot be leveraged in the few-task scenario Yao et al. (2022). |
| Meta-shots | Set to 15 following Snell et al. (2017). |
| Meta-step size $\eta$ | Set to 0.001 following Snell et al. (2017). |
| Meta-iterations $T$ | Set to 20000 for miniImageNet and tieredImageNet following Snell et al. (2017). Our implementation of ProtoNet, unlike Yao et al. (2022), does not require an additional hyperparameter $B$, analogous to MAML, for IBP or IBI training. Thus, for miniImageNet-S, DermNet-S, and ISIC also we set $T$ to 20000. |
| Distance metric | Set to Euclidean following Snell et al. (2017). |
| Additional hyperparameters introduced in ProtoNet+IBP | |
| Interval coefficient $\epsilon$ | Searched in the set $\{0.05, 0.1, 0.2\}$. |
| Softmax coefficient $\gamma$ | Searched in the set $\{0.01, 1, 10\}$. |
| Layer $S$ | For the "4-CONV" learner containing 4 blocks of Convolution, Batch normalization, Max pooling, and ReLU, S is searched at the block level in the set $\{1, 2, 3, 4\}$. For example, $S = 2$ means IBP losses are calculated after the second block. For the "ResNet-12" network the ablation study in Appendix E provides the optimum choice of $S$. |
| Additional hyperparameters introduced in ProtoNet+IBI | |
| $\alpha$ and $\beta$ | Search space contains three pairs of choices $(0.1, 1)$, $(0.25, 1)$, and $(0.5, 0.5)$ where a tuple contains the value of $\alpha$ and $\beta$ in order. |

Table 12: Optimal hyperparamter setting for MAML+IBP, MAML+IBI in 1-shot settings when "4-CONV" network is used.

| Hyperarameter | Hyperparameter settings for datasets | | | | |
|---|---|---|---|---|---|
| | miniImageNet | tieredImageNet | miniImageNet-S | DermNet-S | ISIC |
| Additional hyperparameters introduced in MAML+IBP | | | | | |
| $\epsilon$ | 0.1 | 0.05 | 0.1 | 0.2 | 0.05 |
| $\gamma$ | 0.1 | 0.1 | 0.1 | 0.1 | 0.1 |
| $S$ | 3 | 3 | 3 | 3 | 3 |
| Additional hyperparameters introduced in MAML+IBI | | | | | |
| $\alpha$ and $\beta$ | (0.25, 1) | (0.25, 1) | (0.5, 0.5) | (0.5, 0.5) | (0.25, 1) |

Table 13: Optimal hyperparamter setting for MAML+IBP, MAML+IBI in 5-shot settings when "4-CONV" network is used.

| Hyperarameter | Hyperparameter settings for datasets | | | | |
|---|---|---|---|---|---|
| | miniImageNet | tieredImageNet | miniImageNet-S | DermNet-S | ISIC |
| Additional hyperparameters introduced in MAML+IBP | | | | | |
| $\epsilon$ | 0.1 | 0.05 | 0.1 | 0.2 | 0.05 |
| $\gamma$ | 0.1 | 0.1 | 0.1 | 0.1 | 0.1 |
| $S$ | 3 | 3 | 3 | 3 | 3 |
| Additional hyperparameters introduced in MAML+IBI | | | | | |
| $\alpha$ and $\beta$ | (0.1, 1) | (0.1, 1) | (0.5, 0.5) | (0.5, 0.5) | (0.25, 1) |

Table 14: Optimal hyperparamter setting for ProtoNet+IBP, ProtoNet+IBI in 1-shot settings when "4-CONV" network is used.

| Hyperarameter | Hyperparameter settings for datasets | | | | |
|---|---|---|---|---|---|
| | miniImageNet | tieredImageNet | miniImageNet-S | DermNet-S | ISIC |
| Additional hyperparameters introduced in ProtoNet+IBP | | | | | |
| $\epsilon$ | 0.05 | 0.05 | 0.1 | 0.1 | 0.05 |
| $\gamma$ | 1 | 1 | 1 | 1 | 1 |
| $S$ | 1 | 1 | 1 | 1 | 1 |
| Additional hyperparameters introduced in ProtoNet+IBI | | | | | |
| $\alpha$ and $\beta$ | (0.1, 1) | (0.25, 1) | (0.5, 0.5) | (0.25, 1) | (0.1, 1) |

where $\zeta \sim \mathcal{N}(0, \sigma)$, and the standard deviation $\sigma$ is scheduled similar to $\epsilon$ with starting from 0 and slowly increasing to $\epsilon/2$.

4. MAML+ULBL: Following Morawiecki et al. (2020) we replace the two bound losses with a single one that calculates the distance between the upper and lower interval bounds. The loss $\mathcal{L}_{ULBL}$ in this case can be written as:

$$\mathcal{L}_{ULBL} = \frac{1}{N_q} \sum_{r=1}^{N_q} ||\overline{f}_{\theta^S}(\mathbf{x}_{i,r}^q, \epsilon) - \underline{f}_{\theta^S}(\mathbf{x}_{i,r}^q, \epsilon)||_2^2$$

The full version of Table 1 is provided in the following Table 17.

**Comparison of IBP with other few-shot learners:** As contending meta-learning algorithms, we choose the vanilla MAML along with notable meta-learners such as Meta-SGD Li et al. (2017), Reptile Nichol et al. (2018), LLAMA Grant et al. (2018), R2-D2 Bertinetto et al. (2019), and BOIL

Table 15: Optimal hyperparamter setting for ProtoNet+IBP, ProtoNet+IBI in 5-shot settings when "4-CONV" network is used.

| Hyperarameter | Hyperparameter settings for datasets | | | | |
|---|---|---|---|---|---|
| | miniImageNet | tieredImageNet | miniImageNet-S | DermNet-S | ISIC |
| Additional hyperparameters introduced in ProtoNet+IBP | | | | | |
| $\epsilon$ | 0.05 | 0.05 | 0.1 | 0.1 | 0.05 |
| $\gamma$ | 1 | 1 | 1 | 1 | 1 |
| $S$ | 1 | 1 | 1 | 1 | 1 |
| Additional hyperparameters introduced in ProtoNet+IBI | | | | | |
| $\alpha$ and $\beta$ | (0.1, 1) | (0.1, 1) | (0.5, 0.5) | (0.5, 0.5) | (0.25, 1) |

Table 16: Optimal hyperparameter settings for MAML+IBP/IBI and ProtoNet+IBP/IBI when "ResNet-12" is used as the network.

| | | | Datasets | | |
|---|---|---|---|---|---|
| Learner | Shots | Parameter | miniImageNet-S | DermNet-S | ISIC |
| MAML+IBP | 1 and 5 | $\epsilon$ | 0.1 | 0.1 | 0.1 |
| MAML+IBP | 1 and 5 | $\gamma$ | 0.1 | 0.1 | 0.1 |
| MAML+IBP | 1 and 5 | $S^*$ | 1 | 1 | 1 |
| MAML+IBI (Additional) | 1 and 5 | $\alpha$ and $\beta$ | (0.1, 1) | (0.1, 1) | (0.1, 1) |
| ProtoNet+IBP | 1 and 5 | $\epsilon$ | 0.05 | 0.05 | 0.05 |
| ProtoNet+IBP | 1 and 5 | $\gamma$ | 0.1 | 0.1 | 0.1 |
| ProtoNet+IBP | 1 and 5 | $S^*$ | 1 | 1 | 1 |
| ProtoNet+IBI (Additional) | 1 and 5 | $\alpha$ and $\beta$ | (0.1, 1) | (0.1, 1) | (0.1, 1) |

$^*$: Set as per Appendix E with necessary modifications.

Table 17: Effect of IBP on MAML for miniImageNet and tieredImageNet datasets in terms of 5-way 1-shot Accuracy and intra-task compactness. This is the full version of Table 1.

| Algorithm | Accuracy | | 1-NN distance | |
|---|---|---|---|---|
| | miniImageNet | tieredImageNet | miniImageNet | tieredImageNet |
| MAML (Finn et al., 2017) | 48.70±1.75% | 51.67±1.81% | 0.97±0.02 | 0.98±0.02 |
| MAML+SN on $f_\theta s$ | 44.90±1.12% | 45.26±1.05% | 1.38±0.04 | 1.41±0.04 |
| MAML+SN on $f_\theta$ | 42.83±0.94% | 43.06±0.96% | 1.52±0.04 | 1.53±0.04 |
| MAML+GL | 48.70± 0.97% | 51.90±0.98% | 0.96±0.02 | 0.98±0.02 |
| MAML+ULBL | 49.43±0.90% | 51.67±0.91% | 0.94±0.02 | 0.97±0.02 |
| MAML+IBP (ours) | **50.76±0.83%** | **54.36±0.80%** | **0.90±0.02** | **0.96±0.02** |

Oh et al. (2021). Moreover, considering the regularizing effect of IBP and IBI, we also include meta-learners such as TAML Jamal & Qi (2019), Meta-Reg Yin et al. (2019), and Meta-Dropout Lee et al. (2020) which employ explicit regularization. We further include data augmentation–reliant learners such as MetaMix Yao et al. (2021), Meta-Maxup Ni et al. (2021), as well as the inter-task interpolation method MLTI Yao et al. (2022). In case of metric-learners, we compare against the vanilla ProtoNet in addition to other notable methods like MatchingNet Vinyals et al. (2016), RelationNet Sung et al. (2018), IMP Allen et al. (2019), and GNN Satorras & Estrach (2018). We also compare against ProtoNet coupled with data augmentation methods such as MetaMix, Meta-Maxup, and MLTI, as done in Yao et al. (2022). While Yao et al. (2022) had to modify the training strategy of the canonical ProtoNet to accommodate the changes introduced by MetaMix, Meta-Maxup, and MLTI, the flexibility of IBP and IBI imposes no such requirements. We summarize the findings in Table 18. We can observe that either IBP or IBI or both achieve better Accuracy than the competitors in all cases. The slightly better performance of IBP with ProtoNet seems to imply that IBP-based task interpolation is often unnecessary for ProtoNet when a large number of tasks is available.

Table 18: Performance comparison of the two proposed methods with baselines and competing algorithms on miniImageNet and tieredImageNet datasets. The results are reported in terms of mean Accuracy over 600 tasks with 95% confidence interval.

| Dataset | Learner type | Algorithm | 1-shot | 5-shot |
|---|---|---|---|---|
| miniImageNet | Meta-learners | MAML (Finn et al., 2017) | 48.70±1.75% | 63.11±0.91% |
| | | Meta-SGD (Li et al., 2017) | 50.47±1.87% | 64.03±0.94% |
| | | Reptile (Nichol et al., 2018) | 49.97±0.32% | 65.99±0.58% |
| | | LLAMA (Grant et al., 2018) | 49.40±0.84% | - |
| | | R2-D2 (Bertinetto et al., 2019) | 49.50±0.20% | 65.40±0.20% |
| | | TAML (Jamal & Qi, 2019; Yao et al., 2022) | 46.40±0.82% | 63.26±0.68% |
| | | BOIL (Oh et al., 2021) | 49.61±0.16% | 66.45±0.37% |
| | | MAML+Meta-Reg (Yin et al., 2019; Yao et al., 2022) | 47.02±0.77% | 63.19±0.69% |
| | | MAML+Meta-Dropout (Lee et al., 2020; Yao et al., 2022) | 47.47±0.81% | 64.11±0.71% |
| | | MAML+MetaMix (Yao et al., 2021; 2022) | 47.81±0.78% | 64.22±0.68% |
| | | MAML+Meta-Maxup (Ni et al., 2021; Yao et al., 2022) | 47.68±0.79% | 63.51±0.75% |
| | | MAML+MLTI (Yao et al., 2022) | 48.62±0.76% | 64.65±0.70% |
| | | MAML+IBP (ours) | 50.76±0.83% | 67.13±0.81% |
| | | MAML+IBI (ours) | **52.16±0.84%** | **67.56±0.86%** |
| | Metric-learners | MatchingNet Vinyals et al. (2016) | 43.44±0.77% | 55.31±0.73% |
| | | RelationNet (Sung et al., 2018) | 50.44±0.82% | 65.32±0.70% |
| | | IMP (Allen et al., 2019) | 49.60±0.80% | 68.10±0.80% |
| | | GNN (Satorras & Estrach, 2018) | 49.02±0.98% | 63.50±0.84% |
| | | ProtoNet (Snell et al., 2017) | 49.42±0.78% | 68.20±0.66% |
| | | ProtoNet*+MetaMix (Yao et al., 2021; 2022) | 47.21±0.76% | 64.38±0.67% |
| | | ProtoNet*+Meta-Maxup (Ni et al., 2021; Yao et al., 2022) | 47.33±0.79% | 64.43±0.69% |
| | | ProtoNet*+MLTI (Yao et al., 2022) | 48.11±0.81% | 65.22±0.70% |
| | | ProtoNet+IBP (ours) | 50.48±0.83% | 68.33±0.79% |
| | | ProtoNet+IBI (ours) | **51.79±0.81%** | **68.46±0.79%** |
| tieredImageNet | Meta-learners | MAML (Finn et al., 2017) | 51.67±1.81% | 70.30±0.08% |
| | | Meta-SGD (Li et al., 2017) | 48.97±0.21% | 66.47±0.21% |
| | | Reptile (Nichol et al., 2018) | 49.97±0.32% | 65.99±0.58% |
| | | BOIL (Oh et al., 2021) | 49.35±0.26% | 69.37±0.12% |
| | | MAML+IBP (ours) | **54.36±0.80%** | **71.30±0.77%** |
| | | MAML+IBI (ours) | 54.16±0.79% | 71.00±0.84% |
| | Metric-learners | MatchingNet (Vinyals et al., 2016) | 54.02±0.79% | 70.11±0.82% |
| | | RelationNet (Sung et al., 2018) | 54.48±0.93% | 71.32±0.78% |
| | | ProtoNet (Snell et al., 2017) | 53.31±0.20% | 72.69±0.74% |
| | | ProtoNet+IBP (ours) | 53.83±0.81% | **75.26±0.83%** |
| | | ProtoNet+IBI (ours) | **55.16±0.77%** | 74.96±0.82% |

\* ProtoNet implementation as per Yao et al. (2022).

**Notes on contenders used in Table 2:** The extra parameter settings required for the contenders in Table 2 are as follows:

1. MAML+WCL: Here given a task its worst-case loss in the $\epsilon$-neighborhood (Gowal et al., 2019) is added with the original loss. In essence, this acts similar to augmentation with the worst-case logits. We tune the relative contribution of the original task and the worst-case task to the final $\mathcal{L}_{CE}$ following the recommendations made by (Gowal et al., 2019).

2. MAML+GA (image space): Here the original task is perturbed with Gaussian noise to form the augmented task in the image space. The noise is sampled from a Gaussian with mean 0 and standard deviation $\sigma = \epsilon/2$. The value of $\sigma$ is scheduled similarly to $\epsilon$.

3. MAML+GA (at $f_{\theta^s}$ feature space): Here the embedding of the original task after $f_{\theta^s}$ is perturbed with Gaussian noise. Similar to the image space, the mean of the normal distribution used for sampling noise can be set to 0. However, finding a good $\sigma$ may not be straightforward as the $f_{\theta^s}$ feature space is continuously updating. In our implementation, we take $\sigma$ as half of the median distance between the original task and its bounds over a MAML+IBI run.

The full version of Table 2 is detailed in Table 19. The full version of Table 4 in the main paper is provided here across Tables 20 and 21. Moreover, the full version of Table 5 in the main paper is presented in Table 22.

Table 19: Full version of Table 2 for performance comparison of MAML+IBI against 11 augmentation strategies, in the 5-way 1-shot setting. The results are reported in terms of mean Accuracy over 600 tasks along with the 95% confidence intervals.

| Algorithm | mIS | ISIC | DS |
|---|---|---|---|
| MAML+Inter-task interpolation in image space | 40.90±0.86% | 55.25±1.58% | 48.30±0.81% |
| MAML+Inter-task interpolation after $f_{\theta S}$ | 41.00±0.83% | 61.33±1.52% | 47.43±0.78% |
| MAML+WCL | 41.56±0.88% | 66.83±1.64% | 48.20±0.81% |
| MAML+ULBL+WCL | 41.27±0.84% | 63.50±1.48% | 48.43±0.80% |
| MAML+IBP+WCL | 41.56±0.85% | 64.75±1.61% | 48.90±0.83% |
| MAML+ULBL+Intra-task Interpolation | 40.37±0.80% | 64.91±1.45% | 48.23±0.77% |
| MAML+GA (Image Space) | 41.33±0.85% | 63.25±1.68% | 47.67±0.86% |
| MAML+IBP+GA (Image Space) | 41.83±0.82% | 62.67±1.59% | 48.83±0.82% |
| MAML+IBP+GA (after $f_{\theta S}$) | 41.66±0.84% | 63.75±1.63% | 47.60±0.82% |
| MAML+MLTI Yao et al. (2022) | 41.58±0.72% | 61.79±1.00% | 48.03±0.80% |
| MAML+IBI without $\mathcal{L}_{UB}$ and $\mathcal{L}_{LB}$ losses | 35.26±0.79% | 48.94±1.36% | 41.30±0.81% |
| MAML+IBI (Ours) | **42.20±0.82%** | **68.58±0.93%** | **49.13±0.80%** |

Table 20: Full results for MAML variants on miniImageNet-S, DermNet-S, and ISIC in Table 4 of the main paper. All results are reported in terms of Accuracy over 600 tasks along with 95% confidence level.

| Algorithm | 4-CONV | | ResNet-12 | |
|---|---|---|---|---|
| | 5-way 1-shot | 5-way 5-shot | 5-way 1-shot | 5-way 5-shot |
| miniImageNet-S | | | | |
| MAML (Finn et al., 2017) | 38.27±0.74% | 52.14±0.65% | 40.02±0.78% | 52.56±0.85% |
| MAML+Meta-Reg (Yin et al., 2019; Yao et al., 2022) | 38.35±0.76% | 51.74±0.68% | - | - |
| TAML (Jamal & Qi, 2019; Yao et al., 2022) | 38.70±0.77% | 52.75±0.70% | - | - |
| MAML+Meta-Dropout (Lee et al., 2020; Yao et al., 2022) | 38.32±0.75% | 52.53±0.69% | - | - |
| MAML+MetaMix (Yao et al., 2021; 2022) | 39.43±0.77% | 54.14±0.73% | 42.26±0.75% | 54.65±0.87% |
| MAML+Meta-Maxup (Ni et al., 2021; Yao et al., 2022) | 39.28±0.77% | 53.02±0.72% | 41.97±0.78% | 53.92±0.85% |
| MAML+MLTI (Yao et al., 2022) | 41.58±0.72% | 55.22±0.76% | 43.35±0.90% | 54.89±0.88% |
| MAML+IBP (ours) | 41.30±0.79% | 54.36±0.81% | 43.50±0.86% | 55.13±0.90% |
| MAML+IBI (ours) | **42.20±0.82%** | **55.23±0.81%** | **43.90±0.90%** | **57.00±0.88%** |
| ISIC | | | | |
| MAML (Finn et al., 2017) | 57.59±0.79% | 68.24±0.77% | 59.41±1.98% | 67.66±1.92% |
| MAML+Meta-Reg (Yin et al., 2019; Yao et al., 2022) | 58.57±0.94% | 68.45±0.81% | - | - |
| TAML (Jamal & Qi, 2019; Yao et al., 2022) | 58.39±1.00% | 66.09±0.71% | - | - |
| MAML+Meta-Dropout (Lee et al., 2020; Yao et al., 2022) | 58.40±1.02% | 67.32±0.92% | - | - |
| MAML+MetaMix (Yao et al., 2021; 2022) | 60.34±1.03% | 69.47±0.60% | 62.06±1.77% | 72.18±1.75% |
| MAML+Meta-Maxup (Ni et al., 2021; Yao et al., 2022) | 58.68±0.86% | 69.16±0.61% | 61.64±1.81% | 72.04±1.79% |
| MAML+MLTI (Yao et al., 2022) | 61.79±1.00% | 70.69±0.68% | 62.16±1.88% | 73.56±1.82% |
| MAML+IBP (ours) | 64.91±0.92% | 78.75±0.94% | **64.50±1.48%** | 73.91±1.42% |
| MAML+IBI (ours) | **68.58±0.93%** | **79.75±0.91%** | 63.25±1.51% | **75.66±1.56%** |
| DermNet-S | | | | |
| MAML (Finn et al., 2017) | 43.47±0.83% | 60.56±0.74% | 47.58±0.93% | 63.13±0.85% |
| MAML+Meta-Reg (Yin et al., 2019; Yao et al., 2022) | 45.01±0.83% | 60.92±0.69% | - | - |
| TAML (Jamal & Qi, 2019; Yao et al., 2022) | 45.73±0.84% | 61.14±0.72% | - | - |
| MAML+Meta-Dropout (Lee et al., 2020; Yao et al., 2022) | 44.30±0.84% | 60.86±0.73% | - | - |
| MAML+MetaMix (Yao et al., 2021; 2022) | 46.81±0.81% | 63.52±0.73% | 51.40±0.89% | 64.82±0.87% |
| MAML+Meta-Maxup (Ni et al., 2021; Yao et al., 2022) | 46.10±0.82% | 62.64±0.72% | 50.82±0.85% | 64.24±0.86% |
| MAML+MLTI (Yao et al., 2022) | 48.03±0.79% | 64.55±0.74% | 52.03±0.90% | 65.12±0.88% |
| MAML+IBP (ours) | 48.33±0.83% | 63.33±0.84% | 50.40±0.88% | 65.40±0.89% |
| MAML+IBI (ours) | **49.13±0.80%** | **65.43±0.79%** | **52.10±0.87%** | **66.50±0.92%** |

Table 21: Full results for ProtoNet variants on miniImageNet-S, ISIC, and DermNet-S in Table 4 of the main paper. All results are reported in terms of Accuracy over 600 tasks along with 95% confidence level.

| Algorithm | 4-CONV | | ResNet-12 | |
|---|---|---|---|---|
| | 5-way 1-shot | 5-way 5-shot | 5-way 1-shot | 5-way 5-shot |
| miniImageNet-S | | | | |
| ProtoNet* (Snell et al., 2017; Yao et al., 2022) | 36.26±0.70% | 50.72±0.70% | - | - |
| ProtoNet (Snell et al., 2017) | 40.70±0.79% | 53.16±0.77% | 40.96±0.75% | 55.00±0.86% |
| ProtoNet*+MetaMix (Yao et al., 2021; 2022) | 39.67±0.71% | 53.10±0.74% | 42.95±0.87% | 56.95±0.89% |
| ProtoNet*+Meta-Maxup (Ni et al., 2021; Yao et al., 2022) | 39.80±0.73% | 53.35±0.68% | 42.68±0.78% | 56.07±0.85% |
| ProtoNet*+MLTI (Yao et al., 2022) | 41.36±0.75% | 55.34±0.74% | 44.08±0.83% | 57.14±0.90% |
| ProtoNet+IBP (ours) | 41.46±0.79% | 55.00±0.81% | 43.33±0.82% | 57.40±0.90% |
| ProtoNet+IBI (ours) | **43.30±0.81%** | **55.73±0.80%** | **45.33±0.85%** | **58.23±0.92%** |
| ISIC | | | | |
| ProtoNet* (Snell et al., 2017; Yao et al., 2022) | 58.56±1.01% | 66.25±0.96% | - | - |
| ProtoNet (Snell et al., 2017) | 65.58±0.91% | 75.25±0.90% | 61.91±1.94% | 75.91±1.92% |
| ProtoNet*+MetaMix (Yao et al., 2021; 2022) | 60.58±1.17% | 70.12±0.94% | 65.55±1.80% | 78.33±1.76% |
| ProtoNet*+Meta-Maxup (Ni et al., 2021; Yao et al., 2022) | 59.66±1.13% | 68.97±0.83% | 64.17±1.85% | 77.62±1.86% |
| ProtoNet*+MLTI (Yao et al., 2022) | 62.82±1.13% | 71.52±0.89% | 66.02±1.88% | 79.15±1.87% |
| ProtoNet+IBP (ours) | **70.75±0.95%** | 81.01±0.93% | 66.66±1.52% | 81.00±1.49% |
| ProtoNet+IBI (ours) | 70.25±0.91% | **81.16±0.94%** | **66.75±1.63%** | **81.83±1.58%** |
| DermNet-S | | | | |
| ProtoNet* (Snell et al., 2017; Yao et al., 2022) | 44.21±0.75% | 60.33±0.70% | - | - |
| ProtoNet (Snell et al., 2017) | 46.86±0.77% | 62.03±0.79% | 48.65±0.85% | 65.40±0.81% |
| ProtoNet*+MetaMix (Yao et al., 2021; 2022) | 47.71±0.83% | 62.68±0.71% | 51.18±0.90% | 66.80±0.83% |
| ProtoNet*+Meta-Maxup (Ni et al., 2021; Yao et al., 2022) | 46.06±0.78% | 62.97±0.74% | 50.96±0.88% | 66.38±0.85% |
| ProtoNet*+MLTI (Yao et al., 2022) | 49.38±0.85% | 65.19±0.73% | 52.01±0.93% | 67.28±0.87% |
| ProtoNet+IBP (ours) | 48.06±0.81% | **67.26±0.84%** | 51.33±0.91% | 67.57±0.88% |
| ProtoNet+IBI (ours) | **51.13±0.80%** | 65.93±0.82% | **52.53±0.94%** | **68.00±0.88%** |

\* ProtoNet implementation as per Yao et al. (2022).

Table 22: Full result for Table 5 describing transferability comparison of MAML and ProtoNet, with their MLTI, IBP and IBI variants. All results are reported in terms of Accuracy over 600 tasks along with the 95% confidence intervals. Here, $A \to B$ indicates the model trained on dataset $A$ is tested on dataset $B$.

| Algorithms | Accuracy | |
|---|---|---|
| | DermNet-S → miniImageNet-S | miniImageNet-S → DermNet-S |
| MAML | 25.06±0.79% | 33.40±0.77% |
| MAML+MLTI | 30.03±0.58% | **36.74±0.64%** |
| MAML+IBP (ours) | 27.06±0.78% | 33.90±0.81% |
| MAML+IBI (ours) | **30.23±0.82%** | 36.21±0.84% |
| ProtoNet | 28.76±0.82% | 34.03±0.80% |
| ProtoNet*+MLTI | 30.06±0.56% | 35.46±0.63% |
| ProtoNet+IBP (ours) | 29.60±0.81% | 34.13±0.82% |
| ProtoNet+IBI (ours) | **30.32±0.84%** | **35.63±0.83%** |

\*: ProtoNet implementation as per Yao et al. (2022).

# G CODES

The following code can be used to run MAML, MAML+IBP, and MAML+IBI. Please put the data in a folder with the dataset name. The data should be in ".jpg" image format typically of resolution $84 \times 84$. The data is expected to be arranged in train, validation, and test folders each containing folders for individual classes. The main code is kept inside the "src" folder while the helper code can reside outside. The "main" function resides in the "run_learner.py".

**Helper code: image_data_process.py**

```python
import os
import random

from PIL import Image
import numpy as np
import torch

def read_dataset(data_dir, val_presence=True):

    # Read the image dataset.

    if val_presence is True:
        return tuple(_read_classes(os.path.join(data_dir, x)) for x in ['train', 'val', 'test'])
    else:
        return tuple(_read_classes(os.path.join(data_dir, x)) for x in ['train', 'test', 'test'])

def _read_classes(dir_path):

    # Read the class directories in a train/val/test directory.
    # Images should be in ".jpg" format.

    return [ImageProcessClass(os.path.join(dir_path, f)) for f in os.listdir(dir_path)]

class ImageProcessClass:

    # Loading and using the image dataset.
    # To use these APIs, you should prepare a directory that
    # contains three sub-directories: train, test, and val.

    def __init__(self, dir_path):
        self.dir_path = dir_path
        self._cache = {}

    def sample(self, num_images):

        # Sample images (as pytorch tensor) from the class.

        names = [f for f in os.listdir(self.dir_path) if f.endswith('.jpg')]
        random.shuffle(names)
        images = []
        for name in names[:num_images]:
            images.append(self._read_image(name))
        return images

    def _read_image(self, name):

        # For reading images and transformations as necessary.
        # Image resolution is set to 84x84.

        if name in self._cache:
            return self._cache[name]
        with open(os.path.join(self.dir_path, name), 'rb') as in_file:
            img = Image.open(in_file).resize((84, 84)).convert('RGB')
            img = np.array(img).astype('float32') / 0xff
            img = np.rollaxis(img, 2, 0)
            self._cache[name] = torch.tensor(img)
            return self._read_image(name)
```

**Main code: run_learner.py**

```python
import random
import os
import sys
import numpy as np
import torch

import argparse
from img_data_process import read_dataset
from datetime import datetime
from copy import deepcopy

from src.models import NetworkModel
from src.eval_model import bulk_evaluate
from src.train_model import train
from src.learners import Learner

def argument_parser():
    # Get an argument parser for a training script.
```

```python
    parser = argparse.ArgumentParser(formatter_class=argparse.ArgumentDefaultsHelpFormatter)
    parser.add_argument('--dataset', help='name of dataset', default=None)
    parser.add_argument('--algorithm', help='name of algorithm', default=None)
    parser.add_argument('--seed', help='random seed', default=0, type=int)
    parser.add_argument('--order', help='order for MAML and variants.', default=None, type=int)
    parser.add_argument('--classes', help='number of classes per inner task', default=None, type=int)
    parser.add_argument('--shots', help='number of examples per class', default=None, type=int)
    parser.add_argument('--meta-shots', help='shots for meta update', default=None, type=int)
    parser.add_argument('--inner-iters', help='inner iterations', default=None, type=int)
    parser.add_argument('--learning-rate', help='inner loop learning rate', default=None, type=float)
    parser.add_argument('--meta-step', help='outer loop learning rate', default=None, type=float)
    parser.add_argument('--meta-batch', help='meta-training batch size', default=None, type=int)
    parser.add_argument('--meta-iters', help='meta-training iterations', default=None, type=int)
    parser.add_argument('--eval-iters', help='evaluation inner iterations', default=None, type=int)
    parser.add_argument('--eval-samples', help='evaluation samples', default=None, type=int)
    parser.add_argument('--eval-interval', help='evaluation interval during training', default=None, type=int)
    parser.add_argument('--eval-interval-sample', help='evaluation samples during training', default=None, type=int)
    parser.add_argument('--ibp-eps', help='IBP neighborhood size', default=0, type=float)
    parser.add_argument('--softmax-temp', help='softmax temperature', default=None, type=float)
    parser.add_argument('--only-evaluation', help='for only evaluation', action='store_true', default=False)
    parser.add_argument('--checkpoint', help='load saved checkpoint from path', default=None)
    parser.add_argument('--test-iters', help='number of evaluations', default=None, type=int)
    parser.add_argument('--beta-a', help='beta distrebution parameter a', default=None, type=float)
    parser.add_argument('--beta-b', help='beta distrebution parameter b', default=None, type=float)
    parser.add_argument('--mixup', help='set to use mixup task', action='store_true', default=False)
    parser.add_argument('--ibp-layers', help='number layer to perform IBP/IBI', default=None, type=int)
    return parser

def model_kwargs(parsed_args):

    # Parameters used for initializing the learner.

    return {
        'update_lr': parsed_args.learning_rate,
        'meta_step_size': parsed_args.meta_step,
        'beta_a': parsed_args.beta_a,
        'beta_b': parsed_args.beta_b,
        'softmax_temp': parsed_args.softmax_temp
    }

def train_kwargs(parsed_args):

    # Parameters used for training.

    return {
        'order': parsed_args.order,
        'num_classes': parsed_args.classes,
        'num_shots': parsed_args.shots,
        'meta_shots': parsed_args.meta_shots,
        'inner_iters': parsed_args.inner_iters,
        'meta_batch_size': parsed_args.meta_batch,
        'meta_iters': parsed_args.meta_iters,
        'eval_inner_iters': parsed_args.eval_iters,
        'eval_interval': parsed_args.eval_interval,
        'eval_interval_sample': parsed_args.eval_interval_sample,
        'ibp_epsilon': parsed_args.ibp_eps,
        'mixup': parsed_args.mixup,
        'ibp_layers': parsed_args.ibp_layers
    }

def evaluate_kwargs(parsed_args):

    # Parameters used for evaluation over multiple tasks.

    return {
        'num_classes': parsed_args.classes,
        'num_shots': parsed_args.shots,
        'eval_inner_iters': parsed_args.eval_iters,
        'num_samples': parsed_args.eval_samples
    }

def main():

    args = argument_parser().parse_args()

    torch.manual_seed(args.seed)
    random.seed(args.seed)
    np.random.seed(args.seed)

    # Edit here according to need.
    DATA_DIR = '/home/<userName>/<workingDirectory>/data/' + args.dataset

    # Create directory for storing results and initiate logging.
    if os.path.exists(os.path.join(DATA_DIR, 'val')):
        val_presence = True
        print("Validation set is present.")
    else:
        val_presence = False
        print("Validation set is not found. Exiting.")
        sys.exit()

    time_string = datetime.now().strftime("%m%d%Y_%H:%M:%S")
```

```python
        output_folder = args.dataset + '_' + args.algorithm + '_output_folder_' + time_string
        output_file = output_folder + '/' + 'log_' + time_string + '.txt'

        if not os.path.exists(output_folder):
            os.makedirs(output_folder)

        with open(output_file, 'a+') as fp:
            print('\n'.join(f'{k}={v}' for k, v in vars(args).items()), file=fp)

        device = torch.device('cuda')

        # Instantiate the dataset.
        train_set, val_set, test_set = read_dataset(DATA_DIR, val_presence)

        # Instantiate the learner
        model=NetworkModel(args.classes)

        learner = Learner(model, device, **model_kwargs(args))

        # Perform training or evaluation as per need.
        if args.only_evaluation is False:
            train(learner, train_set, val_set, output_file, output_folder, **train_kwargs(args))
        else:
            assert args.checkpoint is not None, 'For_evaluating_without_training_please_provide_a_checkpoint'
            print('Evaluating...')

            res_file = output_folder + '/' + 'test_performance_' + time_string + '_.txt'
            with open(res_file, 'a+') as fp:
                print('Evalulation_checkpoint:_' + args.checkpoint, file=fp)

            checkpoint_model = torch.load(args.checkpoint, map_location='cuda:0')
            learner.net.load_state_dict(checkpoint_model['model_state'])
            learner.meta_optim.load_state_dict(checkpoint_model['meta_optim_state'])

            train_accuracy, val_accuracy, test_accuracy = [], [], []
            train_cnf, val_cnf, test_cnf = [], [], []

            for ii in range(args.test_iters):

                train_acc, train_div = bulk_evaluate(learner, train_set, **evaluate_kwargs(args))
                val_acc, val_div = bulk_evaluate(learner, val_set, **evaluate_kwargs(args))
                test_acc, test_div = bulk_evaluate(learner, test_set, **evaluate_kwargs(args))

                train_accuracy.append(train_acc)
                val_accuracy.append(val_acc)
                test_accuracy.append(test_acc)

                train_cnf.append(train_div)
                val_cnf.append(val_div)
                test_cnf.append(test_div)

                with open(res_file, 'a+') as fp:
                    print('Test_iteration:_' + str(ii + 1), file=fp)
                    print('Train_accuracy:_' + str(train_accuracy[-1]) + '_+/-_' + str(train_cnf[-1]), file=fp)
                    print('Validation_accuracy:_' + str(val_accuracy[-1]) + '_+/-_' + str(val_cnf[-1]), file=fp)
                    print('Test_accuracy:_' + str(test_accuracy[-1]) + '_+/-_' + str(test_cnf[-1]) + '\n', file=fp)
            save_path = output_folder + '/' + 'results' + '.npz'
            train_accuracy = np.array(train_accuracy)
            val_accuracy = np.array(val_accuracy)
            test_accuracy = np.array(test_accuracy)

            train_cnf = np.array(train_cnf)
            val_cnf = np.array(val_cnf)
            test_cnf = np.array(test_cnf)

            np.savez(save_path, train_accuracy=train_accuracy, val_accuracy=val_accuracy,
                test_accuracy=test_accuracy, train_confidence=train_cnf, val_confidence=val_cnf,
                test_confidence=test_cnf)

if __name__ == '__main__':
    main()
```

## Model: "src/models.py"

```python
import torch
import torch.nn as nn
import torch.nn.functional as F
import numpy as np

# A regular 4-CONV network
class NetworkModel(nn.Module):

    def __init__(self, k_way):

        # Initialize the network layers.

        super(NetworkModel, self).__init__()

        self.conv1 = nn.Conv2d(3, 64, kernel_size=3, stride=1, padding=(1, 1))
        self.batch1 = nn.BatchNorm2d(64, track_running_stats=False)
```

```python
        self.conv2 = nn.Conv2d(64, 64, kernel_size=3, stride=1, padding=(1, 1))
        self.batch2 = nn.BatchNorm2d(64, track_running_stats=False)

        self.conv3 = nn.Conv2d(64, 64, kernel_size=3, stride=1, padding=(1, 1))
        self.batch3 = nn.BatchNorm2d(64, track_running_stats=False)

        self.conv4 = nn.Conv2d(64, 64, kernel_size=3, stride=1, padding=(1, 1))
        self.batch4 = nn.BatchNorm2d(64, track_running_stats=False)

        self.lin1 = nn.Linear(64*5*5, k_way)

    def forward(self, x):

        # A forward function only for reference.

        x = F.relu(F.max_pool2d(self.batch1(self.conv1(x)), 2))
        x = F.relu(F.max_pool2d(self.batch2(self.conv2(x)), 2))
        x = F.relu(F.max_pool2d(self.batch3(self.conv3(x)), 2))
        x = F.relu(F.max_pool2d(self.batch4(self.conv4(x)), 2))

        x = x.view(-1, 64*5*5)
        x = self.lin1(x)

        return x

    def functional_forward(self, x,
        weight_dict,
        layer_index=None,
        mixup_flag=None,
        k_way=None,
        beta_a=None,
        beta_b=None):

        # A functional forward that will actually be used for all requirements.
        # It only uses functionals thus explicitly needs the weights to be passes.
        # The functionals can use the regular layer function or their IBP form as required.

        robust = True
        if layer_index is None:
            y, robust = None, False

        # Block 1
        x = robust_conv_forward(
            x, weight_dict['conv1.weight'], weight_dict['conv1.bias'], stride=1, padding=(1, 1), robust=robust)
        x = robust_batch_norm_forward(
            x, weight_dict['batch1.weight'], weight_dict['batch1.bias'], robust=robust)
        x = F.max_pool2d(x, kernel_size=2, stride=2)
        x = F.relu(x)

        if layer_index == 1:
            y, x, robust = intra_class_mixup(x, mixup_flag, k_way, beta_a, beta_b)

        # Block 2
        x = robust_conv_forward(
            x, weight_dict['conv2.weight'], weight_dict['conv2.bias'], stride=1, padding=(1, 1), robust=robust)
        x = robust_batch_norm_forward(
            x, weight_dict['batch2.weight'], weight_dict['batch2.bias'], robust=robust)
        x = F.max_pool2d(x, kernel_size=2, stride=2)
        x = F.relu(x)

        if layer_index == 2:
            y, x, robust = intra_class_mixup(x, mixup_flag, k_way, beta_a, beta_b)

        # Block 3
        x = robust_conv_forward(
            x, weight_dict['conv3.weight'], weight_dict['conv3.bias'], stride=1, padding=(1, 1), robust=robust)
        x = robust_batch_norm_forward(
            x, weight_dict['batch3.weight'], weight_dict['batch3.bias'], robust=robust)
        x = F.max_pool2d(x, kernel_size=2, stride=2)
        x = F.relu(x)

        if layer_index == 3:
            y, x, robust = intra_class_mixup(x, mixup_flag, k_way, beta_a, beta_b)

        # Block 4
        x = robust_conv_forward(
            x, weight_dict['conv4.weight'], weight_dict['conv4.bias'], stride=1, padding=(1, 1), robust=robust)
        x = robust_batch_norm_forward(
            x, weight_dict['batch4.weight'], weight_dict['batch4.bias'], robust=robust)
        x = F.max_pool2d(x, kernel_size=2, stride=2)
        x = F.relu(x)

        if layer_index == 4:
            y, x, robust = intra_class_mixup(x, mixup_flag, k_way, beta_a, beta_b)

        # Map to number of classes.
        x = x.view(-1, 64*5*5)
        x = F.linear(
            x, weight=weight_dict['lin1.weight'], bias=weight_dict['lin1.bias'])

        return y, x
```

```
def robust_conv_forward(x, weight, bias, stride, padding, robust):

    # Convolution function that can propagate interval bounds.

    if robust is False:
        # Regular convolution
        x = F.conv2d(x, weight, bias, stride, padding)
        return x

    # Convolution propagating interval bounds.
    b_size = x.shape[0]//3

    input_p = x[:b_size]
    input_o = x[b_size:2*b_size]
    input_n = x[2*b_size:]

    u = (input_p + input_n)/2
    r = (input_p - input_n)/2

    out_u = F.conv2d(u, weight, bias, stride, padding)
    out_r = F.conv2d(r, torch.abs(weight), None, stride, padding)
    out_o = F.conv2d(input_o, weight, bias, stride, padding)

    return torch.cat([out_u + out_r, out_o, out_u - out_r], 0)

def robust_batch_norm_forward(x, weight, bias, robust):

    # Batch normalization function that can propagate interval bounds.

    if robust is False:
        # Regular batch normalization.
        x = F.batch_norm(x, running_mean=None, running_var=None,
            weight=weight, bias=bias, training=True)
        return x

    # Batch normalization propagating interval bounds.
    b_size = x.shape[0]//3
    eps = 1e-5

    input_p = x[:b_size]
    input_o = x[b_size:2*b_size]
    input_n = x[2*b_size:]

    # Equivalent to input_o.mean((0, 2, 3))
    mean = input_o.transpose(0, 1).contiguous().view(
        input_o.shape[1], -1).mean(1)
    var = input_o.transpose(0, 1).contiguous().view(
        input_o.shape[1], -1).var(1, unbiased=False)

    # Element-wise multiplier
    multiplier = torch.rsqrt(var + eps)
    multiplier = multiplier * weight

    offset = (-multiplier * mean) + bias

    multiplier = multiplier.unsqueeze(0).unsqueeze(2).unsqueeze(3)
    offset = offset.unsqueeze(0).unsqueeze(2).unsqueeze(3)

    # Because the scale might be negative, we need to apply a strategy similar to linear
    u = (input_p + input_n)/2
    r = (input_p - input_n)/2

    out_u = torch.mul(u, multiplier) + offset
    out_r = torch.mul(r, torch.abs(multiplier))
    out_o = torch.mul(input_o, multiplier) + offset

    return torch.cat([out_u + out_r, out_o, out_u - out_r], 0)

def intra_class_mixup(y, mixup_flag, k_way, beta_a, beta_b):

    # Perform interval bound interpolation

    robust = False
    b_size = y.shape[0]//3
    u = y[:b_size]
    l = y[2*b_size:]
    o = y[b_size:2*b_size]

    if mixup_flag is True:

        num_shots = b_size // k_way
        mixup_params = np.repeat(np.random.beta(beta_a, beta_b, k_way), num_shots)
        mixup_params = torch.tensor(mixup_params, dtype=torch.float).view(b_size, 1, 1, 1).to(y.device)

        rand_ext = np.repeat(np.random.randint(0, 2, k_way), num_shots)
        rand_ext = torch.tensor(rand_ext, dtype=torch.float).view(b_size, 1, 1, 1).to(y.device)

        mixup_params_c = 1 - mixup_params
        rand_ext_c = 1 - rand_ext

        ulo = (mixup_params_c*o +
```

```
            mixup_params*rand_ext*u +
            mixup_params*rand_ext_c*l)

        x = torch.cat([o, ulo], 0)
        return y, x, robust

    return y, o, robust
```

## Learner: "src/learners.py"

```python
import random

import torch
import torch.nn.functional as F
from torch import optim
import numpy as np

import torch.optim as optim
from copy import deepcopy
from collections import OrderedDict

class Learner:

    # Base Learner class for MAML and IBP/IBI variants.

    def __init__(self, model, device, update_lr, meta_step_size, beta_a, beta_b, softmax_temp):

        # Initialization.
        self.device = device
        self.net = model.to(self.device)
        self.meta_optim = optim.Adam(self.net.parameters(), lr=meta_step_size)
        self.update_lr = update_lr
        self.beta_a, self.beta_b = beta_a, beta_b
        self.softmax_temp = softmax_temp

    def train_step(self,
                dataset,
                order,
                num_classes,
                num_shots,
                meta_shots,
                inner_iters,
                meta_batch_size,
                ibp_epsilon,
                mixup,
                ibp_layers):

        # Training function for MAML, MAML+IBP, and MAML+IBI learners.

        # For record keeping
        upper_loss_rec, lower_loss_rec, task_loss_rec, total_loss_rec = 0, 0, 0, 0

        # Triigers FOMAML and variants if required.
        create_graph, retain_graph = True, True
        if order == 1:
            create_graph, retain_graph = False, False

        self.meta_optim.zero_grad()

        if mixup is True:
            random_task = np.random.rand(0, meta_batch_size)

        # Iterate over tasks in a meta-batch
        for task_ind in range(meta_batch_size):

            fast_weight = OrderedDict(self.net.named_parameters())

            train_set, test_set = _split_train_test(
                _sample_mini_dataset(dataset, num_classes, num_shots+meta_shots), test_shots=meta_shots)

            # Support set
            inputs, labels = zip(*train_set)
            inputs = torch.stack(inputs).to(self.device)
            labels = torch.tensor(labels).to(self.device)

            # Fix ordering
            labels, sort_index = torch.sort(labels)
            inputs = inputs[sort_index]

            inputs_cat = torch.cat([inputs+ibp_epsilon, inputs, inputs-ibp_epsilon], 0)

            # Fast adaptation steps
            for _ in range(inner_iters):

                if mixup is True and task_ind == random_task:
                    # For MAML+IBI
                    _, logits = self.net.functional_forward(inputs_cat, fast_weight,
                        ibp_layers, mixup, num_classes, self.beta_a, self.beta_b)

                    b_size = logits.shape[0]//2
                    logits_o = logits[:b_size]
```

```python
                logits_ulo = logits[b_size:]

                fast_loss = (F.cross_entropy(logits_o, labels) +
                    F.cross_entropy(logits_ulo, labels))/2
            else:
                # For MAML and MAML+IBP
                _, logits = self.net.functional_forward(inputs, fast_weight,
                    None, False, None, None, None)

                fast_loss = F.cross_entropy(logits, labels)

            fast_gradients = torch.autograd.grad(fast_loss, fast_weight.values(),
                create_graph=create_graph)

            fast_weight = OrderedDict(
                (name, param - self.update_lr * grad_param)
                for ((name, param), grad_param) in zip(fast_weight.items(), fast_gradients))

        # Query set
        inputs, labels = zip(*test_set)
        inputs = torch.stack(inputs).to(self.device)
        labels = torch.tensor(labels).to(self.device)

        # Fix ordering
        labels, sort_index = torch.sort(labels)
        inputs = inputs[sort_index]

        if ibp_layers is None:
            # Vanilla MAML
            _, logits = self.net.functional_forward(inputs, fast_weight,
                None, False, None, None, None)
            total_loss = F.cross_entropy(logits, labels)
            task_loss_rec = task_loss_rec + total_loss.item()
            total_loss_rec = total_loss_rec + total_loss.item()

        else:
            # For MAML+IBP and MAML+IBI
            inputs_cat = torch.cat([inputs+ibp_epsilon, inputs, inputs-ibp_epsilon], 0)

            if mixup is True and task_ind == random_task:
                # For MAML+IBI
                ibp_estimate, logits = self.net.functional_forward(
                    inputs_cat, fast_weight, ibp_layers, mixup, num_classes,
                    self.beta_a, self.beta_b)

                b_size = logits.shape[0]//2
                logits_o = logits[:b_size]
                logits_ulo = logits[b_size:]

                task_loss = (F.cross_entropy(logits_o, labels) +
                    F.cross_entropy(logits_ulo, labels))/2
            else:
                # For MAML+IBP
                ibp_estimate, logits = self.net.functional_forward(
                    inputs_cat, fast_weight, ibp_layers, False, None, None, None)

                task_loss = F.cross_entropy(logits, labels)

            # Find the propagated bounds
            b_size = ibp_estimate.shape[0]//3

            ibp_estimate_u = ibp_estimate[:b_size]
            ibp_estimate_o = ibp_estimate[b_size:2*b_size]
            ibp_estimate_l = ibp_estimate[2*b_size:]

            # Calculate $\mathcal{L}_{UB}$ and $\mathcal{L}_{LB}$.
            upper_loss = F.mse_loss(ibp_estimate_u, ibp_estimate_o)
            lower_loss = F.mse_loss(ibp_estimate_l, ibp_estimate_o)

            # Dynamic weighting of losses
            concat_loss = torch.cat([task_loss.unsqueeze(0),
                upper_loss.unsqueeze(0), lower_loss.unsqueeze(0)], 0)

            weights = F.softmax(concat_loss/self.softmax_temp, dim=0)
            total_loss = torch.sum(concat_loss * weights)

            # Record keeping
            upper_loss_rec = upper_loss_rec + upper_loss.item()
            lower_loss_rec = lower_loss_rec + lower_loss.item()
            task_loss_rec = task_loss_rec + task_loss.item()
            total_loss_rec = total_loss_rec + total_loss.item()

        total_loss.backward(retain_graph=retain_graph)

# Averaging the loss over meta batches
for params in self.net.parameters():
    params.grad = params.grad / meta_batch_size

# update the meta learner parameters
self.meta_optim.step()
self.meta_optim.zero_grad()
```

```python
            upper_loss_rec = upper_loss_rec / meta_batch_size
            lower_loss_rec = lower_loss_rec / meta_batch_size
            task_loss_rec = task_loss_rec / meta_batch_size
            total_loss_rec = total_loss_rec / meta_batch_size

            return upper_loss_rec, lower_loss_rec, task_loss_rec, total_loss_rec

    def evaluate(self,
                 dataset,
                 num_classes,
                 num_shots,
                 inner_iters):

        # Run a single evaluation of the model.

        # Preserve currently trained model.
        old_state = deepcopy(self.net.state_dict())
        fast_weight = OrderedDict(self.net.named_parameters())

        train_set, test_set = _split_train_test(_sample_mini_dataset(dataset, num_classes, num_shots+1))

        # Support set
        inputs, labels = zip(*train_set)
        inputs = (torch.stack(inputs)).to(self.device)
        labels = (torch.tensor(labels)).to(self.device)

        # Fast adaptation
        for _ in range(inner_iters):

            _, logits = self.net.functional_forward(inputs, fast_weight,
                None, False, None, None, None)
            fast_loss = F.cross_entropy(logits, labels)
            fast_gradients = torch.autograd.grad(fast_loss, fast_weight.values())

            fast_weight = OrderedDict(
                (name, param - self.update_lr * grad_param)
                for ((name, param), grad_param) in zip(fast_weight.items(), fast_gradients))

        # Query set
        inputs, labels = zip(*test_set)
        inputs = (torch.stack(inputs)).to(self.device)
        labels = (torch.tensor(labels)).to(self.device)

        # Inference
        _, logits = self.net.functional_forward(inputs, fast_weight,
            None, False, None, None, None)
        test_preds = (F.softmax(logits, dim=1)).argmax(dim=1)

        # Accuracy
        num_correct = torch.eq(test_preds, labels).sum()

        # Return network to original state for safety.
        self.net.load_state_dict(old_state)

        return num_correct.item()

def _sample_mini_dataset(dataset, num_classes, num_shots):

    # Sample a few shot task from a dataset.

    shuffled = list(dataset)
    random.shuffle(shuffled)
    for class_idx, class_obj in enumerate(shuffled[:num_classes]):
        for sample in class_obj.sample(num_shots):
            yield (sample, class_idx)

def _mini_batches(samples, batch_size, num_batches):

    # Generate mini-batches from some data.

    samples = list(samples)
    cur_batch = []
    batch_count = 0
    while True:
        random.shuffle(samples)
        for sample in samples:
            cur_batch.append(sample)
            if len(cur_batch) < batch_size:
                continue
            yield cur_batch
            cur_batch = []
            batch_count += 1
            if batch_count == num_batches:
                return

def _split_train_test(samples, test_shots=1):

    # Split a few-shot task into a train and a test set.

    train_set = list(samples)
    test_set = []
    labels = set(item[1] for item in train_set)
```

```
            for _ in range(test_shots):
                for label in labels:
                    for i, item in enumerate(train_set):
                        if item[1] == label:
                            del train_set[i]
                            test_set.append(item)
                            break
            if len(test_set) < len(labels) * test_shots:
                raise IndexError('not enough examples of each class for test set')
            return train_set, test_set
```

## Training function: "src/train_model.py"

```python
import os
import numpy as np

import torch
from .learners import Learner

def train(learner,
          train_set,
          val_set,
          model_output_file=None,
          model_save_path=None,
          order=None,
          num_classes=None,
          num_shots=None,
          meta_shots=None,
          inner_iters=None,
          meta_batch_size=None,
          meta_iters=None,
          eval_inner_iters=None,
          eval_interval=None,
          eval_interval_sample=None,
          ibp_epsilon=None,
          mixup=False,
          ibp_layers=None):

    # Train a model on a dataset.
    train_accuracy, val_accuracy = [], []
    upper_loss_store, lower_loss_store, task_loss_store, total_loss_store = [], [], [], []

    # Loop over the training steps.
    for i in range(meta_iters+1):

        # Find current value of interval coefficient.
        cur_ibp_epsilon = eps_scheduler(i, meta_iters, ibp_epsilon)

        # Train the learner for a step.
        upper_loss, lower_loss, task_loss, total_loss = learner.train_step(train_set, order=order,
                                                         num_classes=num_classes, num_shots=num_shots,
                                                         meta_shots=meta_shots,
                                                         inner_iters=inner_iters,
                                                         meta_batch_size=meta_batch_size,
                                                         ibp_epsilon=cur_ibp_epsilon,
                                                         mixup=mixup,
                                                         ibp_layers=ibp_layers)

        # Record losses
        upper_loss_store.append(upper_loss)
        lower_loss_store.append(lower_loss)
        task_loss_store.append(task_loss)
        total_loss_store.append(total_loss)

        if i % eval_interval == 0:

            # Perform intermediate evaluation.
            total_correct = 0
            for _ in range(eval_interval_sample):
                total_correct = total_correct + learner.evaluate(train_set,
                                                    num_classes=num_classes, num_shots=num_shots,
                                                    inner_iters=eval_inner_iters)

            train_accuracy.append(total_correct / (eval_interval_sample * num_classes))

            save_path = model_save_path + '/intermediate_' + str(i) + '_model.pt'

            torch.save({'model_state': learner.net.state_dict(),
                        'meta_optim_state': learner.meta_optim.state_dict()},
                       save_path)

            total_correct = 0
            for _ in range(eval_interval_sample):
                total_correct = total_correct + learner.evaluate(val_set,
                                                    num_classes=num_classes, num_shots=num_shots,
                                                    inner_iters=eval_inner_iters)
            val_accuracy.append(total_correct / (eval_interval_sample * num_classes))

            with open(model_output_file, 'a+') as fp:
                print('batch %d: train=%f val=%f' % (i,
                    train_accuracy[-1], val_accuracy[-1]), file=fp)
```

```python
    # Intermediate record keeping.
    res_save_path = model_save_path + '/' + 'intermediate_accuracies.npz'
    loss_save_path = model_save_path + '/' + 'intermediates_losses.npz'

    np.savez(res_save_path, train_accuracy=np.array(train_accuracy),
        val_accuracy=np.array(val_accuracy))

    np.savez(loss_save_path, upper_loss=np.array(upper_loss_store),
        lower_loss=np.array(lower_loss_store), task_loss=np.array(task_loss_store),
        total_loss=np.array(total_loss_store))

def eps_scheduler(i, meta_iters, ibp_epsilon):

    # Schedule the value of interval coefficient.

    if i < meta_iters*0.9:
        return (i / (meta_iters*0.9)) * ibp_epsilon

    return ibp_epsilon
```

## Inference function: "src/eval_model.py"

```python
"""
Helpers for evaluating models.
"""
import numpy as np
from .learners import Learner

def bulk_evaluate(learner,
            dataset,
            num_classes=5,
            num_shots=5,
            eval_inner_iters=10,
            num_samples=10000):

    # For evaluating the learner on a set of tasks.
    total_correct = []
    for _ in range(num_samples):
        total_correct.append(learner.evaluate(dataset,
                        num_classes=num_classes, num_shots=num_shots,
                        inner_iters=eval_inner_iters))

    total_accuracies = np.array(total_correct) / num_classes
    test_accuracy = total_accuracies.sum() / num_samples
    test_cnf = np.std(total_accuracies)

    # For confidence interval 0.95%
    # z_score = 1.96
    # test_cnf = z_score * (test_cnf / (np.sqrt(num_samples)))

    return test_accuracy, test_cnf
```

