# OpenReview forum: "Interval Bound Interpolation for Few-shot Learning with Few Tasks"
_ICLR.cc/2023/Conference — Submitted to ICLR 2023_

### Official Review · Reviewer_yeNC · 2022-10-17

**Confidence:** 3
**Clarity, Quality, Novelty And Reproducibility:** 1. The paper is well-written and easy…
**Correctness:** 3
**Technical Novelty And Significance:** 3
**Empirical Novelty And Significance:** 2
**Recommendation:** 6

**Strength And Weaknesses:**

1. Related work section is poor. It does not describe a few-shot learning method. The chapter on manifold learning is poor. There exist many flows, NKK methods, and VAE-based method, which was even mentioned.
2. Equation (4) is misleading. It is not trivial to understand what exactly mean I_i^s(\epsilon.)
3. The idea of keeping small interval bounds was described in the paper: https://www.esann.org/sites/default/files/proceedings/2020/ES2020-57.pdf and should be mentioned in the paper.
4. It is unclear why we add losses deducted to the center of the interval and its two borders $L_{LB}$, $ L_{UB}$. We can use the only distance between borders.
5. According to Tab 1. the authors claim, "We see that MAML+IBP outperforms vanilla MAML in terms of 5-way 1-shot classification accuracy on the miniImageNet and tieredImageNet (Ren et al., 2018) datasets."
It is too strong since MAML has a significant variance on miniImageNet, and MAML+IBP is very close. It looks like a slight difference. Maybe the authors should use some statistical comparison of the models.
6. It is possible to see worst-case accuracy instead of interpolation. Does it steal work well?
7. If we do not use worst-case accuracy, we do not want any guarantees. Why we work with relatively complex intervals, we can simply work with a probability distribution (for example, Gaussian ones). Is it possible to use Gaussians instead of intervals?
8. Why are experiments presented on miniImageNet-S instead of miniImageNet?

**Summary Of The Paper:**

The authors use interval-bound propagation to work with few-shot learning problems. Interval arithmetic was used to model the manifold of data devoted to task interpolation. Finally, the authors apply interval architecture to model-agnostic meta-learning and prototype-based metric-learning paradigms.

**Summary Of The Review:**

The paper is very interesting, but I see some problems in the motivation of using Interval arithmetic without worst-case classification loss. In such scenario, we can use simpler objects than intervals.

---

> ### Author Response · Authors · 2022-11-14
> **Response to Reviewer yeNC [1/3]**
>
> We thank you for your insightful suggestions that came of great help for further improving the quality and presentation of our work. We respond to your comments in the corresponding order and have also modified the manuscript accordingly.
>
> ***Q1***. Coverage of Related works
>
> ***A1***: Thank you for your suggestion. We have thoroughly revised the Related Works section in the updated manuscript to include background on few-shot learning and provable robust training of neural networks as well as additional details on manifold learning. We have also rephrased relevant parts of the manuscript to clarify the details on these topics. Specifically, for few-shot learning, we formally defined the problem, briefly discussed the research direction, and set the basis for our work. For provable robust training of neural networks, we traced the evolution of IBP and discussed its working. Moreover, we highlighted how IBP is repurposed in our work. Finally, as per your suggestion, we have added the recent neural networks based research directions to the discussion on manifold learning.
>
> We were unaware of [Morawiecki, et al. 2020] at the time of writing our article and we thank you for notifying us about this important prior work. We have cited the work in the Introduction section while first discussing IBP and have also experimentally found that our approach seems to perform better than that of this work, as reported in the relevant sections of the revised manuscript (also noted below in Q3).
>
> References:
>
> [Morawiecki, et al. 2020]: Morawiecki, Pawel, et al. "Fast and Stable Interval Bounds Propagation for Training Verifiably Robust Models." ESANN 2020 proceedings, European Symposium on Artificial Neural Networks, Computational Intelligence and Machine Learning. 2020.
>
> ***Q2***. Equation (4) is misleading
>
> ***A2***: Thank you for pointing this out. We agree that the erstwhile Equation (4) in the initial manuscript was confusing. We have rephrased the concerned section and removed the equation to enhance the readability of the revised manuscript, without any loss of information.
>
> ***Q3***. Comparison with single bound loss
>
> ***A3***: We have undertaken a comparison of our approach with a new baseline (MAML+ULBL) which directly minimizes the distance between the two interval bounds in the feature space, similar to [Morawiecki, et al. 2020], instead of separately minimizing the individual distances between the original tasks and their respective lower and upper bounds. We summarize the results in Table R5 and have also reported this in Table 1 of the revised manuscript.
>
> ***Table R5: Comparison with single bound loss***
>
> | Algorithm | miniImageNet | tieredImageNet |
> |-----------|--------------|----------------|
> | MAML |	48.70 $\pm$ 1.75%	|	51.67 $\pm$ 1.81% |
> | MAML+ULBL | 49.43 $\pm$ 0.90% | 51.67 $\pm$ 0.91% |
> | **MAML+IBP (ours)** | **50.76 $\pm$ 0.83%** | **54.36 $\pm$ 0.80%** |
>
> We observe that while MAML+ULBL is able to improve upon vanilla MAML, it still does not perform as well as our framework. We would like to highlight that our framework not only consists of the separate losses for the two bounds but also has the associated dynamic weighting scheme to effectively trade them off. This setup forces the original tasks to be "centered" (to the extent possible) within their corresponding neighborhoods in the feature space, a mechanism that the single bound loss baseline lacks. Since the original tasks start out at the center of the neighborhoods in the image space and our aim is to preserve these neighborhoods in the feature space, it makes sense that it would be advantageous to force the original task to be "centered" in the feature embedding as well. Thus, the worse performance of this new baseline is possibly due to its inability to force the original task to be "centered" in the feature space.
>
> References:
>
> [Morawiecki, et al. 2020]: Morawiecki, Pawel, et al. "Fast and Stable Interval Bounds Propagation for Training Verifiably Robust Models." ESANN 2020 proceedings, European Symposium on Artificial Neural Networks, Computational Intelligence and Machine Learning. 2020.

---

> > ### Author Response · Authors · 2022-11-14
> > **Response to Reviewer yeNC [2/3]**
> >
> > ***Q4***. Worst-case loss as a proxy for task-interpolation
> >
> > ***A4***: Thank you for the thoughtful and non-trivial suggestion to use the worst-case loss as a proxy for the loss of artificial tasks. The idea is to find the worst-case loss on the interval bound-based neighborhood of a task and combine it with the classification loss on the original task (MAML+WCL). Conceptually, the worst-case loss can serve as an upper bound on the classification loss for any augmented tasks within the neighborhood. We present the results in Table R6 and have also included the same in Table 2 of the revised manuscript (please find a more detailed ablation study in the revised Appendix).
> >
> > ***Table R6: Comparison with worst-case loss instead of task interpolation***
> >
> > | Algorithm	|	miniImageNet-S	|	ISIC	|	DermNet-S |
> > |-----------|-------------------|-----------|-------------|
> > | MAML	|	38.27 $\pm$ 0.74%	|	57.59 $\pm$ 0.79%	|	43.47 $\pm$ 0.83%	|
> > | MAML+WCL	|	41.56 $\pm$ 0.88%	|	66.83 $\pm$ 1.64%	|	48.20 $\pm$ 0.81% |
> > | MAML+ULBL+WCL	|	41.27 $\pm$ 0.84%	|	63.50 $\pm$ 1.48%	|	48.43 $\pm$ 0.80%	|
> > | MAML+IBP+WCL	|	41.56 $\pm$ 0.85%	|	64.75 $\pm$ 1.61%	|	48.90 $\pm$ 0.83%	|
> > | **MAML+IBI (Ours)**	|	**42.20 $\pm$ 0.82%**	|	**68.58 $\pm$ 0.93%**	|	**49.13 $\pm$ 0.80%** |
> >
> > We compare three versions of the worst-case loss-based method, viz. with IBP losses, without IBP losses, and with the single loss between the interval bounds from [Morawiecki, et al. 2020]. For all of these baselines, we tune the parameter $\kappa$ controlling the trade-off between the original task loss and the corresponding worst-case loss as per the recommendations in [Gowal, et al. 2018]. We observe that these baselines, overall, do not perform as well as IBI, with the difference in performance being most prominent for the ISIC dataset. Moreover, we would like to highlight that the worst-case loss method requires the interval bounds to be propagated through the entire network, all the way to the logits. This is especially costly for larger networks like ResNet-12. We found that a single training step for these baselines on ResNet-12 requires 3.080 seconds on average, compared to the 1.408 seconds for baseline MAML, 1.994 seconds for MAML+IBP, and 2.001 seconds for MAML+IBI, on the same GPU.
> >
> > References:
> >
> > [Morawiecki, et al. 2020]: Morawiecki, Pawel, et al. "Fast and Stable Interval Bounds Propagation for Training Verifiably Robust Models." ESANN 2020 proceedings, European Symposium on Artificial Neural Networks, Computational Intelligence and Machine Learning. 2020.
> >
> > [Gowal, et al. 2018]: Gowal, Sven, et al. "On the effectiveness of interval bound propagation for training verifiably robust models." arXiv preprint arXiv:1810.12715 (2018).
> >
> > **Q5**. Experiments on miniImageNet
> >
> > **A5**: Given our work is focused on few-task few-shot learning problems we had to rely on miniImageNet-S, DermNet-S, and ISIC for the evaluations involving IBI. These three datasets have fewer training tasks and thus are fitting candidates for understanding the impact of IBI. However, to demonstrate the efficacy of IBP in aiding us with learning a better task manifold we already used the standard miniImageNet and tieredImageNet datasets in Table 1 of the initial manuscript. Moreover, Table 16 in Appendix F.3.1 of the initial manuscript already provides a detailed performance comparison of MAML+IBP/IBI and ProtoNet+IBP/IBI with some well-known few-shot learners on standard miniImageNet and tieredImageNet datasets.

---

> > > ### Author Response · Authors · 2022-11-14
> > > **Response to Reviewer yeNC [3/3]**
> > >
> > > ***Q6***. Gaussian noise instead of intervals
> > >
> > > ***A6***: As per your advice, we have undertaken a comparison with a Gaussian noise-based regularization method applied to the "4-CONV" network. The interval bounds are a very convenient way to characterize the neighborhood of a task and enable us to explicitly constrain the learner to preserve these neighborhoods using only two additional forward passes through the feature embedding part of the network. Characterizing the neighborhoods using Gaussian distributions instead makes it non-trivial to explicitly preserve such neighborhoods and we must instead resort to minimizing the distance between the original task and a sample from its corresponding neighborhood. Therefore, for this new Gaussian noise baseline (MAML+GL), instead of minimizing the IBP losses, we minimize the distance between the original tasks and their perturbed counterparts formed by adding Gaussian noise with a standard deviation of $\frac{\epsilon}{2}$ to each of the samples in the task. The results are shown in Table R7 and have also been included in Table 1 in the revised manuscript (further details can be found in the revised Appendix).
> > >
> > > ***Table R7: Comparison of IBP with Gaussian noise-based regularization***
> > >
> > > | Algorithm | miniImageNet | tieredImageNet |
> > > |-----------|--------------|----------------|
> > > | MAML |	48.70 $\pm$ 1.75%	|	51.67 $\pm$ 1.81% |
> > > | MAML+GL | 48.70 $\pm$ 0.97% | 51.90 $\pm$ 0.98% |
> > > | **MAML+IBP (ours)** | **50.76 $\pm$ 0.83%** | **54.36 $\pm$ 0.80%** |
> > >
> > > The results demonstrate that the Gaussian noise-based regularization performs slightly better than baseline MAML and worse than IBP. A possible explanation for the inability of the Gaussian noise-based baseline to perform as well as the proposed IBP-based approach is that the former results in a weaker regularization as it only minimizes the distance between two samples in a neighborhood instead of attempting to conserve the entire neighborhood characterized by bounds.
> > >
> > > ***Q7***. Bold claim while comparing MAML and MAML+IBP
> > >
> > > ***A7***: Thanks for pointing this out. We have rephrased the line to avoid making such a bold claim.

---

> > > > ### Comment · Reviewer_yeNC · 2022-11-15
> > > > **I raise my score.**
> > > >
> > > > The authors answered all my questions and showed exciting experiments, which should be added to the final version of the paper. Therefore I raise my score.

---

> > > > > ### Author Response · Authors · 2022-11-18
> > > > > **To Reviewer yeNC**
> > > > >
> > > > > Thank you for your kind response. We have updated all the recommended changes in the revised manuscript that we have already uploaded.

---

### Official Review · Reviewer_Ww7U · 2022-10-23

**Confidence:** 4
**Correctness:** 3
**Technical Novelty And Significance:** 2
**Empirical Novelty And Significance:** 2
**Recommendation:** 5

**Clarity, Quality, Novelty And Reproducibility:**

Clarity is great and the paper is well structured.  The experimantal validation could be more related to why decisions in paper made
Novelty is interesting, but not a very new idea of using lower and upper bounds. Interpoloating in latent spaces and limiting the updates by regularization has been explored in previous work too.
Reproducibility: I prefer having the running code on an anonymous git rather than the python files in the form of a paper, and I suggest the authors to try this, but I think there is good amount of info that makes reproducibility possible, so I give it 8 / 10

**Strength And Weaknesses:**

Strength:
Well-structured
Nice idea of looking into intervals during training

Weakness
The experiments do not compare with baselines to show the strength of the method. For example, what if instead of using lower and upper bounds for intervals, I just add some noise to the feature representation? That would be an interesting baseline to test against.

**Summary Of The Paper:**

Authors propose a solution for few-task few-shot learning. Their solution is based on two steps:
First they take the task and use interval bounds to estimate two other losses (if the task was in the neighborhood manifold) one upper bound and one lower bound. Then during optimization, they look at the loss of the task and make sure it should be minimized plus the feature representation of the model for the task should be close to the feature representation of the interval lower and upper bound feature representation. They furthermore take the softmax of the three losses and minimize it which is an interesting idea.

Second, they create artificial tasks from the latent space of the model. Since the upper and lower bounds are in proximity of original task, we can interpolate the features of original task with any of these bounds to create artificial tasks.

**Summary Of The Review:**

Based on the above comments, I vote for borderline reject.

---

> ### Author Response · Authors · 2022-11-14
> **Response to Reviewer Ww7U [1/2]**
>
> Thank you for your helpful review. Your comments greatly helped us to improve the quality of our work.
> We respond to your comments in the corresponding order and have also modified the manuscript accordingly.
>
> Due to some issues with the Markdown, we are using $L$ instead of our regular notation \mathcal{L}.
>
> ***Q1***. Using noise instead of lower and upper bound losses
>
> ***A1***: As per your suggestion, we have conducted additional experiments using the "4-CONV" network by augmenting tasks within an $\epsilon$-ball centered at the original task by adding Gaussian noise with a standard deviation of $\sigma=\frac{\epsilon}{2}$ to the samples from the original task, either in the image space or in the feature space. We summarize the new results in Table R4 and have also included the same as part of the ablation presented in Table 2 of the revised manuscript (a detailed version of Table 2 is provided in the Appendix).
>
> ***Table R4: Comparison with Gaussian noise-based augmentation techniques***
>
> | Algorithm	|	miniImageNet-S	|	ISIC	|	DermNet-S |
> |-----------|-------------------|-----------|-------------|
> | MAML	|	38.27 $\pm$ 0.74%	|	57.59 $\pm$ 0.79%	|	43.47 $\pm$ 0.83%	|
> | MAML+GA (Image Space)	|	41.33 $\pm$ 0.85%	|	63.25 $\pm$ 1.68%	|	47.67 $\pm$ 0.86%	|
> | MAML+IBP+GA (Image Space)	|	41.83 $\pm$ 0.82%	|	62.67 $\pm$ 1.59%	|	48.83 $\pm$ 0.82%	|
> | MAML+IBP+GA (after $f_{\theta^S}$)	|	41.66 $\pm$ 0.84%	|	63.75 $\pm$ 1.63%	|	47.60 $\pm$ 0.82%	|
> | **MAML+IBI (Ours)**	|	**42.20 $\pm$ 0.82%**	|	**68.58 $\pm$ 0.93%**	|	**49.13 $\pm$ 0.80%** |
>
> As a naive baseline, we first augment the samples (MAML+GA) in the image space, without minimizing the IBP losses. We further enhance this baseline by introducing the IBP losses in MAML+IBP+GA. We also conduct the corresponding experiment by augmenting the samples (MAML+IBP+GA) in the $f_{\theta^S}$ feature space instead. For the image space augmentations, we schedule $\sigma$ similarly to the $\epsilon$ of IBP. In the case of augmentation in the feature space, we choose $\sigma$ to reflect the median distance between the original tasks and its bounds for the corresponding best-performing IBI settings, to ensure a fair comparison. In all cases, we observe the Gaussian noise-based methods to perform worse than IBI, while the introduction of the IBP loss seems to improve results for image space augmentation. The overall worse performance of the corresponding feature space augmentation may be due to the fact that it is non-trivial to tune $\sigma$ for Gaussian noise-based augmentation in the feature space.
>
> ***Q2***. Relation between methodological choices and experimental results
>
> ***A2***: The following is the list of the main components by which the proposed approach differs from the general few-shot learning setup, along with clarifications about how our experiments support the inclusion of each component:
>
> 1. IBP losses $L_{LB}$ and $L_{UB}$ - We demonstrate in Table 1 of the manuscript how IBP losses alone can improve the performance over vanilla few-shot learning by conserving the task neighborhoods in the feature embedding of the network. Moreover, we have further enriched this table with additional baselines in the revised manuscript, as per the reviewers' suggestions.
> 2.  IBI interpolation between original tasks and their corresponding interval bounds - We establish the effectiveness of IBI using the ablation experiments reported in Table 2. We show that IBI works better than established task interpolation methods like MLTI as well as vanilla task interpolation in both the image and feature spaces. We also show that IBP losses must be minimized for IBI to work well, thus demonstrating how these two components complement each other. We have further enriched this ablation study with additional experiments suggested by the reviewers. Moreover, we also empirically demonstrate the overall effectiveness of IBI across multiple datasets in Table 4.
> 3.  Dynamic weighing of the loss terms - Figure 3 shows the ability of the proposed dynamic weighting scheme to effectively trade-off the classification loss and the two IBP-bound losses. We also conduct an ablation in Table 3 to further motivate the use of the dynamic weighting scheme in our experiments by showing that it performs better than static weighting while dispelling the need to use costly manual tuning of loss weights.

---

> > ### Author Response · Authors · 2022-11-14
> > **Response to Reviewer Ww7U [2/2]**
> >
> > ***Q3***. Reproducibility
> >
> > ***A3***: Thank you for this suggestion. In keeping with your suggestion, we have uploaded the full code to the anonymous git repository at: https://anonymous.4open.science/r/maml-ibp-ibi-D072/, to further improve accessibility.

---

> ### Author Response · Authors · 2022-12-02
> **Please engage with us as the discussion period draws to a close**
>
> Dear reviewer Ww7U,
>
> Thanks again for taking the time to review our manuscript and for providing your valuable feedback! This is a friendly reminder that the final discussion ends soon. We have revised the manuscript as per the reviewers’ suggestions (please see the summary of changes [here](https://openreview.net/forum?id=gwTP_sA-aj-&noteId=bs1PfqqHD7)) and have also furnished a response addressing each of your specific concerns [here](https://openreview.net/forum?id=gwTP_sA-aj-&noteId=f8bLaI4AFw). It has since been slightly more than two weeks, but we have unfortunately not heard from you yet. It would be great if you could engage with us as the discussion period enters its last stages. We would appreciate at least an acknowledgement of our revisions and are also happy to answer any further questions.
>
> Best regards,
>
> The authors.

---

### Official Review · Reviewer_NpKQ · 2022-10-24

**Confidence:** 3
**Correctness:** 3
**Technical Novelty And Significance:** 2
**Empirical Novelty And Significance:** 2
**Recommendation:** 5

**Clarity, Quality, Novelty And Reproducibility:**

The overall presentation is clear. However, this is a gap between the concept of task manifold and data manifold.

In order to clarify the novelty, the authors are expected to discuss the contribution of this method on a task level, which should be something more than simply applying an established sample-wise operation to every sample in a task, and calling it a task-level operation.



**Strength And Weaknesses:**

**Strength**

The challenge of training a few-shot learning model with few tasks is indeed a very interesting direction as the small number of tasks leads to a poor approximation of the task distribution. And interpolating in the task space is clearly a promising way of addressing this challenge.

The idea and the implementation are clearly presented.

** Weaknesses**

1. Task manifold

My primary concern regarding this paper is the definition of 'task manifold'. While manifold is a well-studied concept, task manifold seems new to me, and the connection between tasks manifold and data manifold remains very much unclear to me after reading this paper.
Specifically,  according to the definition of $\epsilon$-neighborhood of a task, I do not see a clear difference between the interval bounds of a task and the interval bounds of a collection of samples. And based on the little difference, the 'task manifold' in this paper seems almost identical to data manifold to me. And this is also confirmed by the realization of the method, which is basically achieved by injecting **sample-wise** noise.
In this case, what is the contribution of this paper to the study of task-level few-shot learning?

While I appreciate the theoretical analysis provided by the authors, I believe the analysis is still very much based on samples instead of tasks.


2. Cost

The discussion of the limitation of the proposed method on cost is appreciated. However, I believe it is still necessary to provide in the experiment section comprehensive comparisons of the training cost between the proposed method and the other methods.

Based on the cost issue, it might be necessary to discuss other alternatives. For example, can adopting spectrum normalization achieve similar results by simply promoting Lipschitz continuity?


**Summary Of The Paper:**

In this paper, the authors propose to extend the idea of interval bounds from the provably robust training literature to few-shot classification and introduce task-level interval bounds based regularization for the training of few-shot classification models (MAML + protonets).
To further improve the model robustness under the challenge of insufficient training tasks, the authors introduce interval bounds interpolation to synthesize more plausible augmented tasks to improve the task diversity for training.

**Summary Of The Review:**

More discussions are expected to clarify the contribution of this paper on a task level. Please refer to the Strength And Weaknesses question for details.

---

> ### Author Response · Authors · 2022-11-14
> **Response to Reviewer NpKQ [1/2]**
>
> Thank you for your insightful review. Your comments and suggestions were extremely helpful for improving the quality of our work. We respond to your comments in order and also revise the manuscript to reflect the changes.
>
> Please note due to markdown issues we are using $T$ instead of our regular notation \mathcal{T}.
>
> ***Q1***. The distinction between task manifold and data manifold
>
> ***A1***: Few-shot learning problems deal with diverse tasks $T=(X,Y)$ consisting of subsets of data $X$ drawn from the same underlying data manifold along with associated labels $Y$. The probability distribution of $T$ (denoted as $p(T)$) which governs the sampling of such tasks is supported on the joint spaces of $X$ and $Y$ and is called the task distribution [Finn et al. 2017, Yao et al. 2022]. Clearly, there is a subtle distinction between the same and the support of $p(T)$. Our usage of the term "task manifold" is an attempt to make this difference clear to the reader. Consequently, the task manifold is also distinct from but closely-related to the data manifold which is only associated with the data distribution on $X$. We thank the reviewer for highlighting the confusion in the initial manuscript due to the interchangeable use of the two terms. We have clearly distinguished the two concepts at the beginning of the revised manuscript and have also explained how the two relate to one another.
>
> References:
>
> [Finn et al. 2017]: Finn, Chelsea, Pieter Abbeel, and Sergey Levine. "Model-agnostic meta-learning for fast adaptation of deep networks." International conference on machine learning. PMLR, 2017.
>
> [Yao et al. 2022]: Yao, Huaxiu, Linjun Zhang, and Chelsea Finn. "Meta-learning with fewer tasks through task interpolation." In International Conference on Learning Representations. 2021.
>
> ***Q2***. Contribution to task-level few-shot learning
>
> ***A2***: Since the unseen tasks are sampled from the same underlying manifold governing the task distribution, we should ideally learn a good representation of the task manifold by preserving the neighborhoods from the high-dimensional manifold in the lower-dimensional feature embedding, similar to the existing manifold learning methods. However, the labels associated with a task can define any arbitrary partitioning of the data. Therefore, we must attempt to preserve the neighborhood for a task by conserving the neighborhoods for the corresponding subset of the data manifold in the feature embedding. For a given set of samples for a particular task, we attempt to conserve the neighborhoods of each of the samples using the lower and upper bound losses obtained using IBP. Thus, you are right in observing that the interval bounds of a task are indeed the same as the collection of the interval bounds for the corresponding data samples. However, this facilitates effective generalization to new tasks using a limited amount of labeled data for task-level few-shot learning by only updating the classifier as the learned feature embedding would likely require very little adaptation.
>
> ***Q3***. Theoretical analysis of the algorithm at the task level.
>
> ***A3***: Thank you for suggesting that we extend our analysis to the task level, in keeping with the few-shot learning setup. We have extended our theoretical analysis of generalization bound to the task level in Theorem 1 of the revised manuscript.

---

> > ### Author Response · Authors · 2022-11-14
> > **Response to Reviewer NpKQ [2/2]**
> >
> > ***Q4***. Comparison of training cost
> >
> > ***A4***: As per your suggestion, we have undertaken a comparison of the proposed methods with the corresponding baselines in terms of the average time in seconds to execute a single training step of the algorithm on a RTX 3090 GPU. We present the results here in Table R2 and have also included the table under Remark 3 in Appendix E of the revised manuscript.
> >
> > ***Table R2: Actual computational cost in seconds for IBP and IBI variants of MAML and ProtoNet with "4-CONV" and ResNet-12 backbones***
> >
> > | Algorithm | 4-CONV 1-shot | 4-CONV 5-shot | ResNet-12 1-shot | ResNet-12 5-shot |
> > |-----------|--------|--------|-----------|-----------|
> > | MAML	|	0.244	|	0.432	|	1.408	|	3.742 |
> > | MAML + IBP (ours)	|	0.407	|	0.615	|	1.994	|	4.324 |
> > | MAML + IBI (ours)	|	0.412	|	0.616	|	2.001	|	4.326 |
> > | ProtoNet	|	0.067	|	0.073	|	0.075	|	0.091 |
> > | ProtoNet + IBP (ours)	|	0.129	|	0.144	|	0.196	|	0.221	|
> > | ProtoNet + IBI (ours)	|	0.133	|	0.156	|	0.202	|	0.233	|
> >
> > We can observe from Table R2 that, in the case of MAML, the IBI and IBP variants only takes about 40\%-70\% additional time for the "4-CONV" network. The difference in cost reduces further to about 20\%-40\% if ResNet-12 is used as the backbone. This is expected as we only need to apply IBP to the first few layers of ResNet-12 to gain its full advantage. Further, the increase in time is generally lower for the 5-shot experiments. For ProtoNet, the increment in computational cost for the proposed techniques is observed to be slightly more compared to that of MAML.
> >
> > ***Q5***. Comparison to lower-cost alternatives of IBP
> >
> > ***A5***: As per your advice that we should undertake a comparison with alternative baselines which have a lower computational cost, we have considered a comparison with spectral normalization [Miyato et al. 2018] and a Gaussian noise-based regularization method, applied to the "4-CONV" network. We consider two spectral normalization baselines (MAML+SN) which apply normalization respectively to the first three layers (corresponding to the feature embedding part of the best-performing IBP configuration) and all layers of the network. For the Gaussian noise baseline (MAML+GL), instead of minimizing the IBP losses, we minimize the distance between the original tasks and their perturbed counterparts formed by adding Gaussian noise with a standard deviation of $\frac{\epsilon}{2}$ to each of the image samples in the task. The results are shown in Table R3 and also included in Table 1 in the revised version of the manuscript (a detailed version of Table 1 can be found in the revised Appendix).
> >
> > ***Table R3: Comparison of IBP with other lower-cost alternatives***
> >
> > | Algorithm | miniImageNet | tieredImageNet |
> > |-----------|--------------|----------------|
> > | MAML |	48.70 $\pm$ 1.75%	|	51.67 $\pm$ 1.81% |
> > | MAML+SN on $f_{\theta^S}$ | 44.90 $\pm$ 1.12% | 45.26 $\pm$ 1.05% |
> > | MAML+SN on $f_{\theta}$ | 42.83 $\pm$ 0.94% | 43.06 $\pm$ 0.96% |
> > | MAML+GL | 48.70 $\pm$ 0.97% | 51.90 $\pm$ 0.98% |
> > | **MAML+IBP (ours)** | **50.76 $\pm$ 0.83%** | **54.36 $\pm$ 0.80%** |
> >
> > The results demonstrate that all the alternative methods perform worse than the corresponding IBP counterpart with the Gaussian noise-based regularization performing the best among these new baselines. While the spectral normalization does impose Lipschitz continuity on the network, we suspect that this alone may not be enough to effectively conserve the neighborhoods from the high-dimensional task/data manifold in the feature-embedding space. This is further supported by the relatively better performance of the Gaussian noise baseline which explicitly attempts to minimize the distance between neighbors in the feature space.
> >
> > References:
> >
> > [Miyato et al. 2018]: Miyato, Takeru, et al. "Spectral normalization for generative adversarial networks." arXiv preprint arXiv:1802.05957 (2018).

---

> > > ### Comment · Reviewer_Ww7U · 2022-12-05
> > > **Reply**
> > >
> > > Hello,
> > >
> > > Since the method works better compared to adding noise as a baseline, I will increase my score to 6. However, I have this assumption that maybe if you spend more time on adjusting the amount of noise it could achieve the same accuracy since the improvement is not that significant. So, I encourage you to look at this in the future.

---

> > > > ### Author Response · Authors · 2022-12-11
> > > > **Thoughts regarding the noise (MAML+GL) baseline**
> > > >
> > > > Dear Reviewer Ww7U,
> > > >
> > > > Thank you so much for engaging!
> > > >
> > > > As you have already observed, the MAML+GL baseline (which adds noise to create a neighbor for a given task and then minimizes the distance with it in the feature space) performs worse than MAML+IBP and is only able to slightly improve over vanilla MAML (same average Accuracy on miniImageNet and slightly higher for tieredImageNet dataset). We would like to highlight that we have chosen the level of noise to be (Gaussian noise with) a standard deviation of $\frac{\epsilon}{2}$ to ensure fair comparison with MAML+IBP.
> > > >
> > > > While tuning the level of noise further may slightly improve the performance for this baseline, we suspect that it may still not close the gap with MAML+IBP, which simply is a stronger regularizer for the same noise level/neighborhood size (since we regularize on the entire neighborhood rather than individually sampled neighbors like in MAML+GL). On the flip side, increasing the level of noise for MAML+GL may in fact harm the performance as the effective neighborhood size (and therefore the resulting regularizing effect) will become larger. We are, however, happy to investigate this in the future like you suggest.
> > > >
> > > > We are stoked that you have decided to increase the score based on the additional results in the revision! We would be grateful if you can kindly change the score in your review to reflect this.
> > > >
> > > > Sincerely,
> > > >
> > > > The authors.

---

> ### Author Response · Authors · 2022-12-02
> **Please engage with us as the discussion period draws to a close**
>
> Dear reviewer NpKQ,
>
> Thanks again for taking the time to review our manuscript and for providing your valuable feedback! This is a friendly reminder that the final discussion ends soon. We have revised the manuscript as per the reviewers’ suggestions (please see the summary of changes [here](https://openreview.net/forum?id=gwTP_sA-aj-&noteId=bs1PfqqHD7)) and have also furnished a response addressing each of your specific concerns [here](https://openreview.net/forum?id=gwTP_sA-aj-&noteId=B-BqGLXrzkW). It has since been slightly more than two weeks, but we have unfortunately not heard from you yet. It would be great if you could engage with us as the discussion period is entering its last stages. We would appreciate at least an acknowledgement of our revisions and are also happy to answer any further questions.
>
> Best regards,
>
> The authors.

---

> ### Author Response · Authors · 2022-12-11
> **Please provide us your valuable feedback**
>
> Dear reviewer NpKQ,
>
> All the other reviewers seem to now be of the opinion that the revised paper has mostly addressed their concerns. It'll be great if you can kindly have a look at the revisions as well and give us your valuable feedback.
>
> Looking forward to hearing back,
>
> The authors.

---

### Official Review · Reviewer_9giy · 2022-10-25

**Confidence:** 4
**Correctness:** 4
**Technical Novelty And Significance:** 3
**Empirical Novelty And Significance:** 3
**Recommendation:** 6

**Clarity, Quality, Novelty And Reproducibility:**

A reasonably good paper with clear motivations and well-presented methodology (except for the background on IBP). The work uses the already known IB and IBP concepts/methods, but how the concept is employed in few-shot learning provides significant insights.

It seems hard to reproduce the algorithm. Do the authors have the plan to share the code?

**Strength And Weaknesses:**

Strengths: The authors propose an interesting and effective solution to few-shot, few-task learning. A theoretical analysis is provided. Ample experimental results are given validating the proposed method. Figures 1 and 2 are very helpful in understanding the proposed ideas.

Weaknesses:

It is a bit difficult to fully understand the descriptions of interval bound propagation without any background on related topics. It would be better to have more detailed insights, explanations, discussions on this and add them to the Related Works or Preliminary section.

The theoretical analysis seems to say nothing about the algorithm in terms the efficacy of minimizing L_{LB} + L_{UB}; the analysis focuses only on CE loss by making the remaining terms arbitrarily small (also lacking any insights into the effectiveness of the proposed task interpolation). The generalization error with bounded loss function is somewhat obvious from stat theory point of view.

Not necessarily weaknesses but some questions:

1. In Table 4, why does +IBI perform worse than +IBP in some cases even with the augmented tasks?

2. If the authors are concerned about whether the new task would be on the task manifold or not, then why don’t they consider just interpolating between the task and the neighboring one, where the neighboring task lies within the epsilon ball centered from $f_{\theta^{S}}(x_{i,r}^s)$?

3. Applying IBP boosts the performance even without task interpolation, which I found intriguing. Could the authors provide more insights into how preserving the neighborhoods can assist the model to generalize well on unseen tasks?


**Summary Of The Paper:**

The notion of interval bounds is used to enhance few-shot learning, especially in the situation where only few tasks are in the training set.  Interval bound propagation (IBP) through network layers and the use of the modified training loss ensure that the model outputs stay close to the interval bounds in the embedding space of initial layers. This improves the chances for the artificially formed tasks to remain on the task manifold. A theorem is established that guarantees that artificially created (via IBP interpolation) support sets are essentially as good as those from the actual distribution. Experiments on both MAML and prototypical few-shot learning show improved performance over baselines.

**Summary Of The Review:**

A reasonably good paper overall. Although the IB and IBP method/concept is not new, the application to few-shot few-task learning is interesting and insightful. A theoretical analysis that reflects the effect of bound losses would have been better, as they are in the core of the proposed method.

---

> ### Author Response · Authors · 2022-11-14
> **Response to Reviewer 9giy [1/2]**
>
> Thank you for your helpful suggestions. Your comments greatly helped us to further improve the quality of our work. We respond to your comments in the corresponding order and have also modified the manuscript accordingly.
>
> Note that due to Markdown issues we are using $L$ instead of our regular notation \mathcal{L}.
>
> ***Q1***. Background on Interval Bound Propagation.
>
> ***A1***: As per your suggestion, we have included an additional Related Works subsection providing some crucial background on IBP, which was designed for provably robust training of neural networks. We have also rephrased the parts of the Introduction and the dedicated subsections in the Preliminaries as well as the Proposed Method sections, that describe IBP, to make the concept easier for the reader.
>
> ***Q2***. Theory on the efficacy of minimizing $L_{LB}$ and $L_{UB}$
>
> ***A2***: Solutions from the constrained formulation tend to become upper bounds of those obtained using the Lagrangian form [Boyd and Vandenberghe, 2004]. This has motivated us to follow the former approach for our analysis. In this setup, given margins of error $t_1$ and $t_2$, Lemma 1 shows that we can always choose $\varepsilon$ suitably such that both the constraints $L_{LB} \leq t_1$ and $L_{UB} \leq t_2$ are satisfied. Hence, we demonstrate the generalization bound on the remaining loss $L_{CE}$. We would however like to highlight that the generalization bound depends on $\tilde{\lambda}(\hat{\varepsilon}, \lambda)$. As a consequence, for a given $\varepsilon$ satisfying the error margins, further minimizing $L_{LB}$ and $L_{UB}$ makes the generalization bound tighter by making $\hat{\varepsilon}$ smaller. Moreover, we would like to mention in this connection that we have attempted to further enrich our theoretical analysis by extending it to the case of multiple tasks, in keeping with the few-shot learning setup.
>
> References:
>
> [Boyd and Vandenberghe, 2004]: Boyd, Stephen, Stephen P. Boyd, and Lieven Vandenberghe. Convex optimization. Cambridge university press, 2004.
>
> ***Q3***. Why do +IBI variants perform worse than +IBP variants in some cases?
>
> ***A3***: We think the slight loss in performance for IBI, compared to IBP, in a few cases is due to the use of the dynamic weighting which we employ to trade-off the three losses in our formulation. We speculate that manually tuning the weights of the losses may help IBI outperform IBP in these cases (or at least close the gap between the two). We, however, refrain from undertaking this comparison as that would not be a particularly tractable endeavor, due to the number of losses involved and their diverse scales. We would also like to observe that the consistently better performance of both IBP and IBI variants (in comparison to baselines) along with the cost savings (no additional tuning for the loss weights), owing to the dynamic weighting, far outweighs this possible drawback.
>
> ***Q4***. Insight into how IBP helps with generalization to new tasks
>
> ***A4***: Few-shot learning problems deal with diverse tasks consisting of subsets of data drawn from the same underlying data manifold along with associated labels. The joint distribution of data and corresponding labels which governs the sampling of such tasks is often called the task distribution while we refer to the support of this distribution as the task manifold. The task manifold is distinct from but closely related to the data manifold associated with the data distribution. Since the unseen tasks are sampled from the same underlying manifold governing the task distribution, we should ideally learn a good representation of the task manifold in the lower-dimensional feature embedding. However, the labels associated with a task can define any arbitrary partitioning of the data. Therefore, we must attempt to preserve the neighborhood for a task by conserving the neighborhoods for the corresponding subset of the data manifold in the feature embedding. For a given set of samples for a particular task, we attempt to conserve the neighborhoods of each of the samples using the lower and upper bound losses obtained using IBP. This facilitates effective generalization to new tasks using a limited amount of labeled data by only updating the classifier as the learned feature embedding would likely require very little adaptation.

---

> > ### Author Response · Authors · 2022-11-14
> > **Response to Reviewer 9giy [2/2]**
> >
> > ***Q5***. Interpolation within an $\epsilon$-ball centered at $f_{\theta^{S}}(x_{i,r})$
> >
> > ***A5***: As per your suggestion, we have conducted additional experiments using the "4-CONV" network by augmenting tasks within an $\epsilon$-ball centered at $f_{\theta^{S}}(x_{i,r})$. Since it is difficult in practice to always find another task within the $\epsilon$-ball of the original task, we can instead construct new tasks by adding Gaussian noise with a standard deviation of $\sigma=\frac{\epsilon}{2}$ to the samples from the original task, either in the image space or in the feature space. We summarize the results in Table R1 and have also included the same as part of the ablation presented in Table 2 of the revised manuscript (a detailed version of Table 2 is also provided in the Appendix).
> >
> > ***Table R1: Comparison with Gaussian augmentation within $\epsilon$-ball***
> >
> > | Algorithm	|	miniImageNet-S	|	ISIC	|	DermNet-S |
> > |-----------|-------------------|-----------|-------------|
> > | MAML	|	38.27 $\pm$ 0.74%	|	57.59 $\pm$ 0.79%	|	43.47 $\pm$ 0.83%	|
> > | MAML+GA (Image Space)	|	41.33 $\pm$ 0.85%	|	63.25 $\pm$ 1.68%	|	47.67 $\pm$ 0.86%	|
> > | MAML+IBP+GA (Image Space)	|	41.83 $\pm$ 0.82%	|	62.67 $\pm$ 1.59%	|	48.83 $\pm$ 0.82%	|
> > | MAML+IBP+GA (after $f_{\theta^S}$)	|	41.66 $\pm$ 0.84%	|	63.75 $\pm$ 1.63%	|	47.60 $\pm$ 0.82%	|
> > | **MAML+IBI (Ours)**	|	**42.20 $\pm$ 0.82%**	|	**68.58 $\pm$ 0.93%**	|	**49.13 $\pm$ 0.80%** |
> >
> > As a naive baseline, we first augment the samples (MAML+GA) in the image space, without minimizing the IBP losses. We further enhance this baseline by introducing the IBP losses in MAML+IBP+GA. We also conduct the corresponding experiment by augmenting the samples (MAML+IBP+GA) in the $f_{\theta^S}$ feature space instead. For the image space augmentations, we schedule $\sigma$ similarly to the $\epsilon$ of IBP. In the case of augmentation in the feature space, we choose $\sigma$ to reflect the median distance between the original tasks and its bounds for the corresponding best-performing IBI settings, to ensure a fair comparison. In all cases, we observe the Gaussian noise-based methods to perform worse than IBI, while the introduction of the IBP loss seems to improve results for image space augmentation. The overall worse performance of the corresponding feature space augmentation may be due to the fact that it is non-trivial to tune $\sigma$ for Gaussian noise-based augmentation in the feature space.
> >
> > ***Q6***: Reproducibility
> >
> > ***A6***: We had already included the python codes for our method in Appendix G of the initial manuscript. However, to further improve accessibility, we have now uploaded the full code to the anonymous git repository https://anonymous.4open.science/r/maml-ibp-ibi-D072/.

---

> > > ### Comment · Reviewer_9giy · 2022-11-27
> > > **thanks for the response**
> > >
> > > Thanks for your efforts on addressing my concerns and questions, particularly on extending the theorem to the case of multiple tasks and providing further comparison. But I’m not fully convinced by your response wrgds to Theorem 1. Regarding the response A2, you claim that one can always choose $\epsilon$ arbitrarily small (you say “suitably” but this may be referring to making it “small” as far as l understand) such that the desired constraints are met. But I feel this is a bit contradicting to what you have done in the paper; it is clear to me that making $\epsilon$ arbitrarily small means the effect of $L_{LB}$ and $L_{UB}$ is excluded in practice. In short, Theorem 1 does not take the effect of minimizing $ L_{LB}, L_{UB}$ (along with $L_{CE}$) into account. Nevertheless I remain positive overall; I maintain my original score.

---

> > > > ### Author Response · Authors · 2022-12-02
> > > > **additional clarification regarding Theorem 1**
> > > >
> > > > Thank you for taking the time to peruse and appreciate our revisions. We would like to take this opportunity to provide some additional clarification regarding Theorem 1. In the constrained formulation, the quantities $t_1$, $t_2$ act as tolerable error margins to realized losses; whereas in a Lagrangian setup, such regulations are done by $\lambda_1$, $\lambda_2$. Since there is a direct correspondence between the two formulations, one may always go back to the latter given $t_1$, $t_2$ [Boyd and Vandenberghe, 2004; Chapter 5]. From Lemma 1, we find that the two losses ($L_{LB}$, $L_{UB}$) turn out to be $\leq \varepsilon D$ (ignoring the square and sum over samples for simplicity), almost surely. As such, choosing $\varepsilon \leq \min( \frac{t_1}{D}, \frac{t_2}{D} )$ suffices to show that the constraints are met and that is all we need. Such a choice will also preserve the regularizing effect of the constraints. Thus, we emphasize the fact that our usage of the term ‘suitably’ does not generally mean ‘arbitrarily small’, rather small enough to satisfy this criterion.
> > > >
> > > > References: [Boyd and Vandenberghe, 2004]: Boyd, Stephen, Stephen P. Boyd, and Lieven Vandenberghe. Convex optimization. Cambridge university press, 2004.

---

### Author Response · Authors · 2022-11-15
**Summary of Paper Revisions**

We thank the reviewers for their time and effort spent reviewing our article. The insightful comments and suggestions made by the reviewers were extremely helpful for significantly improving the quality and presentation of our work. We have addressed each of the reviewers individually and here we present a summary of the revisions made in the revised manuscript. All the changes made in the revised manuscript are highlighted in blue. The revised paper that we have uploaded contains the following key changes:

1. We have updated the Introduction section to provide a better motivation for our proposed approach. In this context, we attempted to clarify the concept of task manifold and how Interval Bound Propagation (IBP) may be repurposed for learning better representation for few-shot learners along with providing a straightforward way to augment new tasks in a few-task scenario.
2. We have thoroughly revised the Related Works section to add more discussion on the manifold learning approaches. Further, we have added additional discussion on few-shot learning and provably robust neural networks to better put our contributions in context.
3. We have updated the Preliminaries and Proposed Method sections to better describe the IBP method and how we are using it for learning better representations.
4. In an attempt to demonstrate the efficacy of MAML+IBP we have revamped Table 1 to include new contenders using Spectral Normalization, Gaussian loss-based regularization, and a single loss measuring the distance between the two interval bounds. An extended version of Table 1 is also presented in the Appendix.
5. We have also significantly improved the ablation study of Table 2 to include several variants using Worst Case Loss and Gaussian augmentation at image and feature levels. An extended version of Table 2 is also available in the Appendix.
6. We have listed the actual training cost of our proposed method in the Appendix.
7. We thoroughly revised Theorem 1 to extend the analysis to the task level.
8. On top of the extensive discussion on hyperparameter setting and the actual python code that were already present in the Appendix of the initial manuscript, we have included a link to an anonymous code repository to further facilitate reproducibility.

---

### Author Response · Authors · 2022-11-17
**Revised Manuscript Available**

We would like to highlight that we have uploaded the revised version of the manuscript containing all the revisions listed in the "Summary of Paper Revisions". Since we are fast approaching the deadline for revising the manuscript, we request the reviewers to take some time to peruse the revisions and kindly let us know if they'd like any additional changes made before the Nov 18th deadline. We would however also be happy to make any necessary updates to the final version of the manuscript based on discussion subsequent to the deadline. We thank all the reviewers once again for their time and constructive feedback.

---

### Decision · Program_Chairs · 2023-01-20

**Decision:**

Reject

**Justification For Why Not Higher Score:**

See comments above.  There are enough lingering concerns in this paper that I think it could use another round of review and polish.

**Justification For Why Not Lower Score:**

N/A

**Metareview: Summary, Strengths And Weaknesses:**

Thanks for your submission to ICLR.

This was a borderline paper.  After rebuttal and discussion, the scores were 5, 6, 6, 6 (one reviewer noted that they wanted to change their score from 5 to 6, but could not, so that is accounted for in the above).  The reviewers were somewhat engaged in discussion, with some reviewers noting that the author replies did clarify many doubts.

Some of the strengths noted by the reviewers included an interesting and relevant problem, along with some theoretical analysis and a fairly solid empirical evaluation.  On the negative side, some reviewers noted that there were clarity issues with the paper (e.g., related work, discussion of some terms) and that the novelty was not super high.

Given that this was a borderline paper, I took a close look at the paper as well.  I actually like the idea a lot and think it's definitely worth publishing.  And I'm reasonably happy with the experimental results.  My main issue is that I still feel that this paper needs some polish, even after the changes made by the authors.  In particular, I actually agree with some of the criticisms raised by the reviewers that the intro, background, and related work sections are difficult to follow and lack details.  Many readers of this paper will be unfamiliar with interval bound propagation, and the discussion in the paper lacks some key details.  I also agree that some topics are imprecise; in general I just think this needs a bit more polish.  That's really my main criticism, but I think it's significant enough that it should go through another round of review.

**Summary Of Ac-Reviewer Meeting:**

We were unable to schedule a time that worked for everyone.